# Understanding Subpopulation Shifts through a Unified Lens of Separability

## Abstract

Subpopulation shifts have been a major challenge for deploying machine learning algorithms. The shift in subgroup proportions between training and test data always leads to a significant performance drop or suboptimal performance in certain groups, therefore limiting the broader or more reliable usage of machine learning methods. We present a unified theoretical framework to characterize a broad range of subpopulation shifts, including but not limited to well-studied shifts such as spurious correlation, under-representation, and class imbalance. Within this framework, we derive the performance of the Bayesian optimal classifier fitted on skewed data. The evaluation of thorough subpopulation shifts provides a quantitative tool to guide dataset collection. Our analysis further highlights the critical role of the feature separability assumption in our modeling, which explains the effectiveness of recent shift-mitigation methods and enabled principled comparison of encoders. Overall, this framework offers a unified perspective on evaluating subpopulation shifts and provides practical guidance on future work in both data collection and training strategies.

## 1 Introduction

Despite the wide application of machine learning in real-world scenarios (Dong et al., 2021), there have been various challenges in practice (Quiñonero-Candela et al., 2022; Koh et al., 2021; Amodei et al., 2016). For example, in ideal scenarios, we assume that the training and test data are independent and identically distributed (i.i.d.). However, the assumption does not always hold. The model performance often degrades due to distribution shifts — differences between training and test data distributions (Quiñonero-Candela et al., 2022). Among various challenges brought by the distribution shift, the subpopulation shift (Koh et al., 2021), where there is a disparity between the relative proportions in the training and test data, has been a common and critical issue. The subpopulation shift often lead to in the potential failure of model generalization on the subgroups that are underrepresented in training data. Therefore, addressing subpopulation shifts to ensure performance across subgroups is essential for both effective dataset curation and robust model generalization.

Several well-studied phenomena in distribution shift can fit into the definition of subpopulation shifts, however, they are often discussed or mitigated separately. For example, models trained on data with *spurious correlation* (SC) suffer from performance drops in subgroups where the correlation breaks or no longer exists (Arjovsky et al., 2019; Sagawa et al., 2020; Geirhos et al., 2020). Such spurious correlation refers to certain attributes associated with the target labels that may not consistently appear in the test data, such as the background (Beery et al., 2018; Geirhos et al., 2020; Xiao et al., 2020) or texture (Geirhos et al., 2018). Another case, *under-representation* (UR), is often discussed within the research of machine learning fairness (Kearns et al., 2018), where the data with one or more specific attributes has a significantly larger or smaller proportion than others (demographic groups), potentially leading to accuracy disparities among these groups. To ensure fairness for the application, achieving higher worst-group accuracy is often prioritized (Idrissi et al., 2022). Emphasizing under-represented groups in test data aligns closely with mitigating subpopulation shifts. Additionally, *class imbalance* (CI), with a few classes making up the majority of the data, often leads the models to prioritize the performance of these majority classes (He & Garcia, 2009; Johnson & Khoshgoftaar, 2019; Zhang et al., 2023). In binary settings, class imbalance can also pose similar challenges with performance degradation when one class significantly outweighs the other,

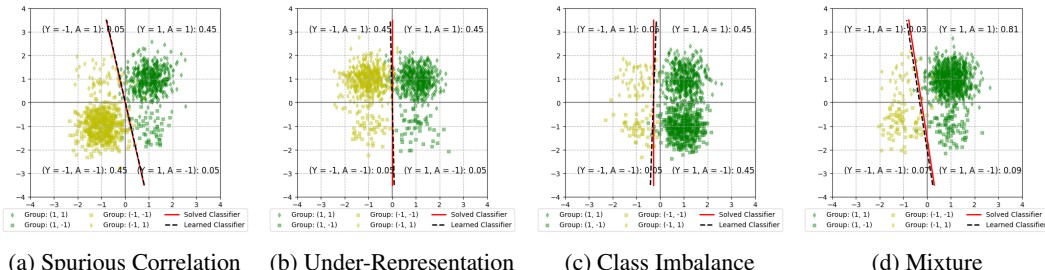

| (a) Spurious Correlation | (b) Under-Representation | (c) Class Imbalance | (d) Mixture |

Figure 1: **Visualization of Classifiers Solved/Learned on 2D Data.** Green and yellow denote two classes. Solid red lines denote theoretical solutions; dashed black lines denote empirical ones, which align closely, indicating the theoretical estimates reliably approximate empirical ones.

as seen in skin cancer diagnosis, where benign cases greatly predominate among all data (Codella et al., 2019; Wu et al., 2023).

Together, these scenarios — data with SC, UR, or CI — share common challenges or goals to ensure performance across subgroups. Yang et al. (2023b) introduced a benchmark for these subpopulation shifts, examining and categorizing them together from a simplified probabilistic perspective. Furthermore, building upon the assumption of feature separability (Wang & Wang, 2024), our work aims for a unified theoretical framework to model general subpopulation shifts, encompassing these shifts and mixtures, and study their quantitative consequences.

Specifically, we derive the theoretical performance of Bayesian optimal classifier that fits on skewed training data, modeled by subpopulation configurations and feature separability. Considering the effects of these factors in data modeling, our contributions are twofold from data and model perspectives. From the aspect of data, we evaluate the performance under comprehensive subpopulation configurations, from highly skewed to fully balanced. The evaluation provides a quantitative tool to examine the performance and guide future dataset collection to ensure robust model training on them. This is especially critical in high-stakes domains such as healthcare (Kukar, 2003) and autonomous driving (Huval et al., 2015), where failures on specific subgroups can lead to severe consequences. Additionally, we conduct analysis on these named shifts with varying intensities to reveal their effects. We found that as the dataset with CI/SC becomes more balanced, the performance gains become marginal. This highlights another benefit of our quantitative analysis: even moderately skewed datasets can yield good robustness, which is valuable in scenarios where data collection is challenging to balance the groups.

From the model perspective, we identify feature separability, especially of the invariant attribute, as a central factor for robustness under shifts. Our empirical analysis shows that higher separability is consistently associated with better performance. This principle provides a unifying explanation for the success of several recent methods of shift mitigation. Moreover, the similar idea has also been proven to be effective in the shift mitigation in foundation models such as CLIPs (Yang et al., 2023a; Saranrittichai et al., 2024). In addition, when synthetic datasets or data variants are available, our framework can estimate performance under fully balanced configurations, which serves as a performance upper bound and helps encoder comparison, guiding the selection of strong image encoders for deployment.

In summary, our contribution can be concluded as follows: (1) a theoretical framework to model random subpopulation shifts and examine the performance; (2) data aspect: a quantitative tool to evaluate the subpopulation configurations for dataset constructions; (3) model aspect: usage of feature separability as the key to robustness that provides explanation for recent success, and evaluation of encoders via theoretical achievable performance.

## 2 RELATED WORK

**Subpopulation Shift.** Following our discussion in Section 1, several research topics align well with the concept of subpopulation shifts. For spurious correlation, there has been a series of work on the mitigation of the spurious correlation from different aspects, including data augmentation (Zhang et al., 2018; Yao et al., 2022), resampling (sag, 2020; Kirichenko et al., 2023; Idrissi et al., 2022), reweighting (Nam et al., 2020; Liu et al., 2021; Idrissi et al., 2022), regularization (Wang & Wang,

2024), group robust learning (Hu et al., 2018; Sagawa et al., 2020) and invariant feature learning (Arjovsky et al., 2019; Sun & Saenko, 2016; Li et al., 2018). Additionally, under-representation, where certain demographic groups appear less frequently, is often characterized in machine learning fairness or debiasing. Fairness methods are typically categorized based on the stage of application—pre-processing (Wang et al., 2020; Jang et al., 2021; Zietlow et al., 2022), in-processing (Alghamdi et al., 2022; Shui et al., 2022; Wan et al., 2023), or post-processing (Jiang et al., 2020; Petersen et al., 2021; Xian et al., 2023). We focus on model training in our work, thus primarily using in-processing techniques (Deka & Sutherland, 2023; Lu et al., 2024) that apply regularization during model training. For class imbalance, there are three main categories: the data-level (Hensman & Masko, 2015; Lee et al., 2016; Pouyanfar et al., 2018; Buda et al., 2018), algorithm-level (Wang et al., 2016; Lin et al., 2017; Cao et al., 2019; 2021), and hybrid (Huang et al., 2016; Dong et al., 2018) approaches, according to their primary technical characteristics (Johnson & Khoshgoftaar, 2019). In addition to the above, several recent augmentation-based methods have been proposed specifically to improve robustness under subpopulation shifts. Umix (Yao et al., 2022) interpolates samples in an uncertainty-aware manner to improve importance weighting, and Selective Augmentation (Han et al., 2022) applies targeted data generation to improve the out-of-distribution robustness. These approaches can be viewed as modifying the effective training distribution.

While these methods provide effective algorithmic tools, they do not offer a unified theoretical understanding of why and when such distribution alterations improve robustness. Our work instead provides a principled analytical framework that characterizes subpopulation shifts. The work that is most similar to ours is (Yang et al., 2023b), where Yang *et al.* categorized shifts with a probabilistic decomposition framework and contributed an empirical benchmarking approach to evaluate algorithmic generalization. In contrast, our work focuses on unifying the shifts under a generalized theoretical framework to analyze the performance and underlying factors of subpopulation shifts.

**Gaussian Data Modeling.** Previous works have demonstrated the consistency between Gaussian assumptions and the more complicated non-Gaussian settings (Montanari et al., 2019; Mei & Montanari, 2021). Such Gaussian assumptions are widely used in the analysis of spurious correlation problems (Nagarajan et al., 2021; Liu et al., 2022; Yao et al., 2022; Idrissi et al., 2022; Ming et al., 2022; Wang & Wang, 2024). Specifically, Wang & Wang (2024) proposed an important assumption of feature separability for data modeling and demonstrated that the analysis generalizes well to complex models and real-world data. However, constrained by their oversimplified SC-only setting, the data have to be completely balanced regarding any attribute $A_n$, and this setup falls in a typical spurious correlation under our framework, without capturing broader forms of subpopulation shifts. In contrast, we adopt the assumption of feature separability and focus on more general cases of subpopulation shifts, greatly extending the scope and encompassing the prior finding.

## 3 BAYESIAN CLASSIFIER UNDER SUBPOPULATION SHIFTS

### 3.1 PROBLEM SETUP

We consider the binary classification data $\mathcal{D} = \mathcal{X} \times \mathcal{Y}$ with labels $\mathcal{Y} = \{\pm 1\}$. Each input $\boldsymbol{x} \in \mathcal{X}$ can be represented by $N$ features $\boldsymbol{z} = (\boldsymbol{z}_1, \ldots, \boldsymbol{z}_N) \in \mathcal{Z} \subset \mathbb{R}^d$ via a bijection $\Phi : \mathcal{Z} \to \mathcal{X}$. We model each feature $\boldsymbol{z}_n$ as Gaussian (Ming et al., 2022; Zhang et al., 2022; Wang & Wang, 2024), controlled by binary attributes $a_n$ with all attributes as $\boldsymbol{a} \in \mathcal{A} = \{\pm 1\}^N$. For example, the Waterbirds dataset (Sagawa et al., 2020) includes attributes of bird species (water & land birds) and the background type (water & land). Without loss of generality, let $a_1$ be the **invariant** attribute defining the label, i.e., $y \equiv a_1$. Specifically, conditioned on an attribute value, the feature follows $Z_n | A_n = a_n \sim \mathcal{N}(a_n \cdot \boldsymbol{\mu}_n, \Sigma_n)$ with mean $\boldsymbol{\mu}_n \in \mathbb{R}^{d_i}$ and covariance $\Sigma_n^{d_n} \in \mathbb{H}_+^{d_n \times d_n}$ which determine the geometric characteristics of the $n$-th feature. The feature dimensions $d_n$ satisify $\sum_{i=1}^{N} d_i = d$. Following Wang & Wang (2024), let $\boldsymbol{\mu_a} = [a_1 \boldsymbol{\mu}_1^T, \cdots, a_N \boldsymbol{\mu}_N^T]^T \in \mathbb{R}^d$ denote the group center determined by $\boldsymbol{a}$, and $\Sigma = \text{diag}(\Sigma_1, \cdots, \Sigma_N) \in \mathbb{H}_+^{d \times d}$ as the global covariance.

Thus, the data are modeled with Gaussian features parameterized by attributes, whose values also naturally categorize the data into subpopulations. We describe the configuration of subpopulation sizes by the parameters: $\beta = \mathbb{P}(Y = 1)$, the base rate of the positive class; $\alpha_{\text{sign}(y)}^{(i)} = \mathbb{P}(A_i = Y_i | Y_i = y)$, the correlation between attributes $A_i$ and the labels within each class. Together, $\boldsymbol{\alpha}$ and $\beta$ uniquely determine the statistical configuration of the subgroups; thus, the subpopulation shift can

be characterized by the shifts in $\boldsymbol{\alpha}$ and $\beta$ between training and test data. For simplicity, we focus on shifts from randomly biased training data to a balanced test set.

## 3.2 DERIVATION OF BAYESIAN OPTIMAL LINEAR CLASSIFIER

To study the effects of the subpopulation shifts, we first identify the classifier that fits the biased training distribution, then evaluate its test performance. We start by considering the expected performance of a classifier parameterized by $\boldsymbol{w}, b$, i.e., $\hat{y} = \boldsymbol{w}^T\boldsymbol{x} + b$. Under our data modeling, the linear classifier is a natural choice. The invariant features correspond to the invariant attribute that aligns with the labels. Therefore, the decision rule reduces to a shared-covariance Gaussian discrimination problem, which produces a linear boundary, similar to classical LDA.

**Lemma 1. (General Accuracy)** *Given the data $\mathcal{D}$ characterized by $(\boldsymbol{\mu}, \Sigma, \boldsymbol{\alpha}_{\pm}, \beta)$ and the model $(\boldsymbol{w}, b)$, the accuracy of the model over the data distribution is*

$$Acc(\boldsymbol{w}, b) = \frac{1}{2}\left\{1 + \sum_{\boldsymbol{a} \in \{\pm 1\}^N} \left[\kappa_+(\boldsymbol{a})\operatorname{erf}\left(\frac{\boldsymbol{\mu}_{\boldsymbol{a}}^T\boldsymbol{w} + b}{\sqrt{2\boldsymbol{w}^T\Sigma\boldsymbol{w}}}\right) + \kappa_-(\boldsymbol{a})\operatorname{erf}\left(\frac{\boldsymbol{\mu}_{\boldsymbol{a}}^T\boldsymbol{w} - b}{\sqrt{2\boldsymbol{w}^T\Sigma\boldsymbol{w}}}\right)\right]\right\}$$

*where $\kappa_+(\boldsymbol{a}) = \beta \cdot \prod_{n=1}^N (\alpha_+^{(n)})^{\mathbf{1}_{a_n=1}}(1 - \alpha_+^{(n)})^{\mathbf{1}_{a_n=-1}}$ and $\kappa_-(\boldsymbol{a}) = (1 - \beta) \cdot \prod_{n=1}^N (\alpha_-^{(n)})^{\mathbf{1}_{a_n=-1}}(1 - \alpha_-^{(n)})^{\mathbf{1}_{a_n=1}}$, with $\boldsymbol{a}$ encoding $N$ binary attributes for subgroups.*

The proof is delayed to Appendix A.1. Now that the closed form of $Acc(\boldsymbol{w}, b)$ of a given classifier and data distribution is available, we solve the Bayesian optimal classifier $(\boldsymbol{w}^*, b^*) = \arg\max_{\boldsymbol{w}, b} Acc(\boldsymbol{w}, b)$ which maximizes the performance for the training distribution. A key property of the Bayesian optimal classifier is the colinearity of its parameter, stated as follows:

**Lemma 2. (Colinearity)** *For the Bayesian optimal classifier, the $n$-th block of the weight vector is colinear with $\Sigma_n^{-1}\boldsymbol{\mu}_n$. Specifically, $\boldsymbol{w}_n^* = \eta_n \Sigma_n^{-1}\boldsymbol{\mu}_n$, for some scalar coefficient $\eta_n \in \mathbb{R}$.*

We thus prove the following results in Appendix A.2. This lemma highlights that the geometric property of the Bayesian classifier is highly dependent on the corresponding features. Moreover, it indicates that solving the Bayesian optimal classifier equals solving the colinear coefficients, which can be solved with numerical algorithms. We next analyze the test performance of the Bayesian optimal classifier, focusing on group accuracy for simplicity.

**Lemma 3. (Bayesian Group Accuracy)** *The Bayesian optimal classifier achieves group accuracy for $\boldsymbol{a} \in \mathcal{A}$ as*

$$Acc_{\boldsymbol{a}} = \frac{1}{2}\left(1 + \operatorname{erf}\left(\frac{\sum_{n=1}^N a_n \eta_n m_n + a_1 b}{\sqrt{2\sum_{n=1}^N \eta_n^2 m_n}}\right)\right) \tag{1}$$

*where $\eta_n$ are the coefficients such that $\boldsymbol{w}_n^* = \eta_n \Sigma_n^{-1}\boldsymbol{\mu}_n$, $b$ is the bias term in the classifier, $a_n$ denotes the value of $n$-th attribute for this subgroup, and $m_n = \boldsymbol{\mu}_n^T\Sigma_n^{-1}\boldsymbol{\mu}_n$ are the Mahalanobis distance of the $n$-th feature that determine the **feature separability**.*

This formulation emphasizes group-wise performance, whereas the global accuracy or other forms of accuracy metrics can be achieved by the weighted group-wise performance. This perspective offers a clearer and more flexible view of performance shifts, which subpopulation shift arises from changes in the subpopulation configuration, rather than the performance difference between individual training and test *samples*.

Lemmas 1–3 characterize the Bayes-optimal classifier and group-wise accuracies for an arbitrary number of attributes, establishing that our framework applies directly to the general $N$-attribute setting. Since all attributes are indirectly correlated to each other through their correlations with the label $Y$, it suffices to examine each attribute individually, and the insights generalize to others. As a result, we focus on the scenario of two attributes $N = 2$. With $a_1$ as the invariant attribute tied to the label, we have $\alpha_+^{(1)} = \alpha_-^{(1)} = \mathbb{P}(A_1 = Y) = 1$. By expressing the coefficients of two attributes with their ratios $\tau = \eta_2/\eta_1$, we establish the following theorem characterizing the Bayesian classifier.

**Theorem 1. (Bayesian Classifier)** *When $N = 2$, the Bayesian classifier is $\hat{y} = \boldsymbol{\mu}_1^T \Sigma_1^{-1} \boldsymbol{z}_1 + \tau \boldsymbol{\mu}_2^T \Sigma_2^{-1} \boldsymbol{z}_2 + b$ where $\tau, b \in \mathbb{R}$ are the solutions to the equation system:*

$$\begin{cases} \tau = \dfrac{\gamma_{1,1}(\tau, b) + \gamma_{-1,-1}(\tau, b) - \gamma_{1,-1}(\tau, b) - \gamma_{-1,1}(\tau, b)}{\gamma_{1,1}(\tau, b) + \gamma_{-1,-1}(\tau, b) + \gamma_{1,-1}(\tau, b) + \gamma_{-1,1}(\tau, b)} \\ \gamma_{1,1}(\tau, b) + \gamma_{1,-1}(\tau, b) = \gamma_{-1,-1}(\tau, b) + \gamma_{-1,1}(\tau, b) \end{cases}$$

Here $\gamma$s are functions of $\tau, b$ defined along with the proof in Appendix A.4. Such a system can be efficiently solved by nonlinear optimization once the data characteristic $(\boldsymbol{\mu}, \Sigma, \boldsymbol{\alpha}, \beta)$ is specified. Furthermore, the performance can be estimated with the solved $\tau, b$ according to Lemma 3. To this end, we are capable of estimating the performance under any subpopulation configurations.

## 4 ANALYSIS ON KEY FACTORS

We now incorporate the well-studied shifts in our general modeling and delve deep into the analysis of the models' compromised performance under subpopulation shifts. We focus on the effects of subpopulation configurations and feature separability, two important factors in our data modeling.

**Typical Shifts.** For two binary attributes with $N = 2$, we specify each subgroup by $(Y, A)$ where $Y$ the invariant attribute (the label) and $A$ is the other attribute for short. Since $\alpha_+^{(1)} = \alpha_-^{(1)} = 1$ for the invariant attribute, we eliminate them and simplify $\boldsymbol{\alpha}$ by writing $\alpha_\pm$ in place of $\alpha_+^{(2)}, \alpha_-^{(2)}$. In addition to our general configuration characterized by $\alpha_\pm, \beta$, our framework incorporates several well-studied categories of subpopulation shifts with the following criteria: (1) **Class Imbalance (CI)**: $\mathbb{P}(Y = 1) \neq 0.5$; (2) **Spurious Correlation (SC)**: $\mathbb{P}(A = Y) \neq 0.5$; (3) **Under-Representation (UR)**: $\mathbb{P}(A = 1) \neq 0.5$. For those configurations that satisfy only one of the criteria, we refer to them as *typical shifts*; otherwise, a mixture of shifts.

In fig. 1, we conduct a simple comparison between decision boundaries, obtained from either the estimated Bayesian classifier or the empirical classifier trained on 2D synthetic data. Across various typical shifts and even a mixture of shifts, the close alignment between their decision boundaries indicates that the empirically trained classifiers tend to converge toward the Bayes classifiers, thereby justifying the use of the theoretical estimations to analyze the behavior of empirical models. Further, existing work either studies a shift mixture with fixed group sizes (e.g. fig. 1(d)) or varying degrees of correlation with a single type (e.g. fig. 1(a)-(c)). As a result, we aim to provide a holistic understanding of the shifts and their consequences, which have been lacking in prior work.

To evaluate the performance under subpopulation shifts, we focus on the two common metrics: adjusted accuracy (AA) (Yang et al., 2023b), the accuracy averaged over subgroups, and worst-group accuracy (WGA), the minimum group accuracy across all subgroups. The AA also equals the accuracy on fully balanced test data. Both metrics are easily derived based on per-group accuracy $Acc_{\boldsymbol{a}}$, which, from Lemma 3 and Theorem 1, can be computed directly. As our framework allows an exhaustive examination across possible combinations, we further discuss the relationship between adjusted accuracy and worst-group accuracy in Appendix B. Our analysis shows cases in which the two metrics behave counterintuitively, which underscores the value of a theoretical framework in revealing behaviors that may be difficult to detect empirically. By taking $\alpha_+, \alpha_-, \beta \in \{i/M\}_{i=1}^{M-1}$, we examine the performance under a comprehensive enumeration of subpopulation configurations with $M = 50$, resulting in $49^3$ evaluated configurations.

### 4.1 DATA ASPECT: SUBPOPULATION CONFIGURATIONS

**Result visualization.** For the effective visualization across the continuous and comprehensive configurations, we map the $49^3$ data configuration $\alpha_\pm, \beta$ to the 3D coordinates in a 3-simplex and use gradient colors at these coordinates to denote the performance. The reason for adopting the 3-simplex visualization includes (1) the 3-simplex allows a continuous visualization of four groups bounded with a sum of 1 in 3D space; (2) the 3-simplex coordinate mapping ensures a symmetric view, especially convenient for comparison of symmetric group sizes.

Specifically, each coordinate within the 3-simplex represents a specific configuration characterized by $\alpha_\pm, \beta$, computed by a convex combination weighted by the group sizes. The four **vertices** rep-

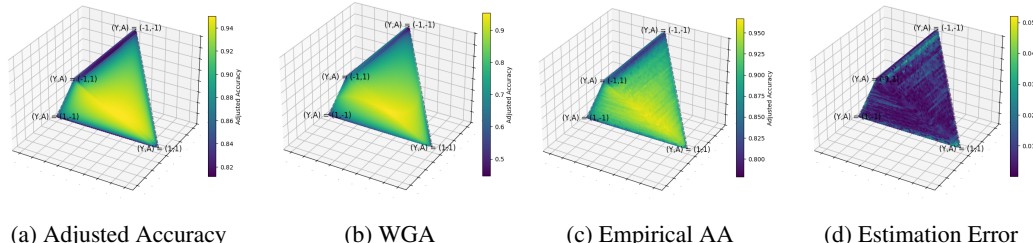

(a) Adjusted Accuracy      (b) WGA      (c) Empirical AA      (d) Estimation Error

Figure 2: **Theoretical and Empirical Results over All Subpopulation Configurations.** The configurations are mapped to 3-simplex coordinates, with color denoting performance. The results highlight consistency between adjusted accuracy (AA) and worst-group accuracy (WGA), and more importantly, between (a) theoretical AA and (b) empirical AA, with low estimation error in (d).

resent the extreme cases where all data fall in one group, and we denote the corresponding group in fig. 2 for the vertices with the values for $(Y, A)$. The closer one coordinate is to a vertex, the more dominant the corresponding group is in the configuration. Furthermore, each **edge** represents the cases where all data only fall into two groups, resulting in extreme shifts. For example, the edge connecting $(Y, A) = (-1, -1)$ and $(Y, A) = (1, -1)$, denoted as $E_{(-1,-1),(-1,1)}$, shows all data have $Y = -1$, and $A = -1$ or $A = 1$ with varying ratio, indicating strong CI standing out in a mixture of CI and UR. To summarize, the edges for extreme shifts are (1) CI: $E_{(-1,-1),(-1,1)}$ and $E_{(1,-1),(1,1)}$; (2) SC: $E_{(1,1),(-1,-1)}$ and $E_{(1,-1),(-1,1)}$; (3) UR: $E_{(1,1),(-1,1)}$ and $E_{(1,-1),(-1,-1)}$ (blocked). Due to the symmetry of the 3-simplex, we choose the view of the 3-simplex as shown in fig. 2, which provides a sufficient view of the unique edges and faces for our analysis. Further details of the 3-simplex mapping are included in the appendix.

**Theoretical and Empirical Comparison.** With the above setup of configurations, fig. 2 (a) and (b) visualize the adjusted accuracy and worst-group accuracy across comprehensive configurations. However, enumerating and evaluating all $49^3$ points empirically is infeasible for deep models and large datasets. So we test linear classifiers with a small synthetic dataset named MNIST-concat (Wang & Wang, 2024), where images of digit 3 and digit 5 are concatenated vertically, where the top block represents the core attribute and the bottom block represents the other attribute. More details of the dataset are provided in the appendix. The empirical results are shown in a 3-simplex in fig. 2 (c), which shows the empirical adjusted accuracy of classifiers trained on data with diverse subpopulation configurations and evaluated on balanced test data. It follows a similar pattern to fig. 2 (a), thus indicating our framework offers a comprehensive view to examine the performance changes as the shifts vary.

Further, section 3 provides the theoretical performance if the subpopulation configuration and feature separability are available, and we aim to validate it with empirical results. While the configuration is explicitly defined with $\alpha_{\pm}, \beta$, the feature separability $m_1, m_2$, defined with the Mahalanobis distance between Gaussian-modeled features, is generally infeasible to compute directly for non-Gaussian datasets. The feature separability measures how easily an attribute can be distinguished between possible values, and higher feature separability indicates an easier attribute to classify. Thus, considering the feature separability as unknowns, Wang & Wang (2024) estimated $\hat{m}_1$ and $\hat{m}_2$ using two sets of empirical accuracies — correlation ratio pairs, resulting in an equation system for numerical solution to the separability. We adopt this estimation under our framework for broader subpopulation shifts and provide the equation system of our settings in the appendix. While in principle any two sets of empirical results can be used to estimate separability, we typically select two sets where the subpopulation configuration is the major difference between them and the resulting performance disparity is sufficiently large compared to their standard deviation, to exclude the effects of empirical factors. The estimated $\hat{m}_1$ and $\hat{m}_2$ allow us to estimate the performance under new subpopulation configurations according to Lemma 3 and Theorem 1.

Following the above discussion, we estimate the separability and accuracy for MNIST-concat and display the estimation error in fig. 2 (d). We can observe that the estimations are very accurate. Quantitatively, 96.3% of the estimates show absolute deviations within 2%. Despite requiring data variants for separability estimation, the thorough evaluation can greatly facilitate the dataset collection, as the data variants are accessible and the configuration matters under the scenario.

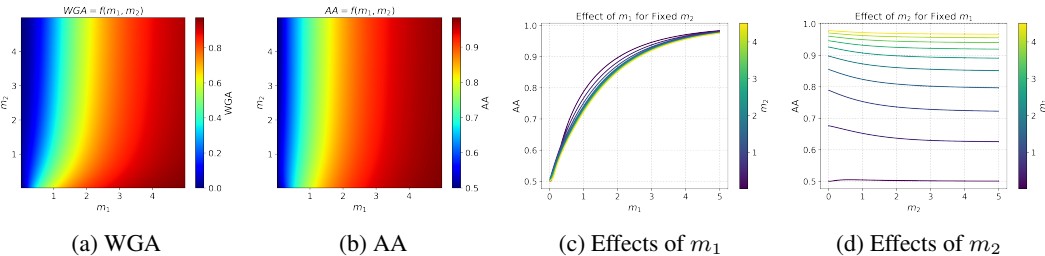

(a) WGA         (b) AA         (c) Effects of $m_1$         (d) Effects of $m_2$

Figure 3: **The Influence of Separability on Performance.** (a)–(b) show worst-group and adjusted accuracy as feature separability $m_1$ (invariant) and $m_2$ (the other) vary under a fixed subpopulation configuration. (c) visualizes the effects of $m_1$ by fixing $m_2$ for lines, and (d) shows the effects of $m_2$ similarly. Performance improves with a larger $m_1$ or a smaller $m_2$, with $m_1$ being more effective.

## 4.2 MODEL ASPECT: FEATURE SEPARABILITY

In addition to the above analysis with varying data configuration, we are also interested in the effects of data property, which is modeled as feature separability. We study the estimated performance as the feature separability $m_1, m_2$ varies, and visualize the performance in fig. 3 (a) and (b). Additionally, fig. 3(c) shows adjusted accuracy as $m_1$ (of invariant attribute) varies with fixed $m_2$, while (d) shows the reverse, varying $m_2$ with fixed $m_1$, with the line color denoting the value of the fixed separability. Our results show that increasing $m_1$ or decreasing $m_2$ improves the performance with adjusting $m_1$ being most effective. That is, making the core features stand out more clearly from the other helps the model stay reliable against subpopulation shift.

However, we note that although feature separability is feasible to estimate, as discussed earlier, it requires data variants that may still be inaccessible in standard datasets. Alternatively, we consider a more general **empirical separability** for standard datasets that with only one data variant. Following the theoretical setup, we examine the features the encoder extracts from balanced test images. Categorizing the features according to the values of a specified attribute, we compute the Euclidean distance between the centers of the feature clusters and consider it as an indicator of the separability. Although sharing the same idea of evaluating how difficult one attribute can be classified, the theoretical feature separability describes an ideal property of the data itself, while the alternative empirical feature separability can be influenced by several factors: the inherent theoretical feature separability, the configurations of subpopulations in the training data, the encoders, or even the training strategies. The theoretical feature separability is crucial yet difficult to alter, while other factors can be partially addressed by adjusting subgroup configurations in datasets, employing better encoders, and better training strategies. Therefore, in addition to evaluating the theoretical performance for data curation, our framework also guides shift mitigation from this aspect, such as applying strong encoders or improving training strategies for better separability.

## 5 EXPERIMENTS

### 5.1 EXPERIMENT SETUP

**Datasets.** Our datasets can be categorized into two parts: the synthetic datasets with flexible subpopulation configurations, and the standard datasets with fixed subpopulation configurations.

Similar to the evaluation in section 4, we use synthetic datasets to verify the accuracy estimation. However, unlike section 4, it is infeasible to exhaustively iterate over these configurations due to time and computational costs. Instead, we focus on the three typical shifts with varying configurations, ranging from highly skewed to balanced. We use two synthetic datasets based on the original Waterbirds (Sagawa et al., 2020) and CelebA (Liu et al., 2015). Both datasets contain two binary attributes, and we construct them as a fixed number of samples across shift configurations per dataset. The setup of typical shifts allows us to simplify the shift configurations from $\alpha_{\pm}, \beta$ to a single parameter $\zeta \in [0.5, 1)$, which refers to the percentage of the two majority groups in the training data. We refer to the synthetic datasets as Waterbirds-$\zeta$ and CelebA-$\zeta$ to denote their shift configurations, and use fully balanced test data for the empirical performance evaluation. The Waterbirds-$\zeta$ is synthesized by combining the segmented birds from the Caltech-UCSD Birds-200-

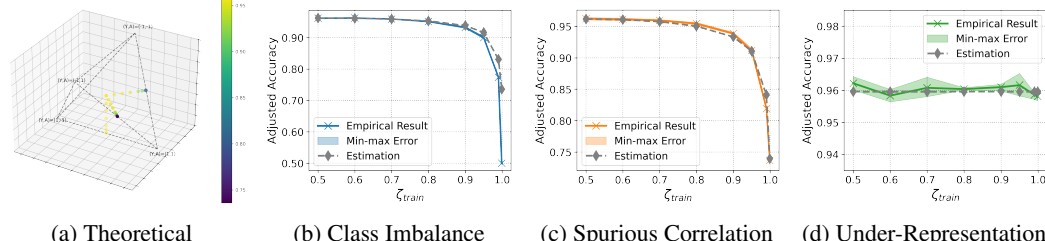

(a) Theoretical      (b) Class Imbalance      (c) Spurious Correlation      (d) Under-Representation

Figure 4: **Theoretical and Empirical Performance with Varying $\zeta_{\text{train}}$ on Waterbirds-$\zeta$ with Typical Shifts.** (a) visualizes the theoretical performance along with the configurations of typical shifts in the 3-simplex and (b-d) show the empirical adjusted accuracy for each typical shift, which aligns with the estimated accuracy.

Table 1: **Performance of ResNet50 with Various Methods on Waterbirds-$\zeta$.** Results are color-coded by shift type with lighter or darker shades indicating lower/higher accuracy within all results with each shift. The best result per $\zeta$ and shift is in **bold**. The performance gain diminishes as the data with CI/SC becomes more balanced.

| | Class Imbalance | | | Spurious Correlation | | | Under-Representation | | |
|---|---|---|---|---|---|---|---|---|---|
| Method | $\zeta = 0.9$ | $\zeta = 0.7$ | $\zeta = 0.5$ | $\zeta = 0.9$ | $\zeta = 0.7$ | $\zeta = 0.5$ | $\zeta = 0.9$ | $\zeta = 0.7$ | $\zeta = 0.5$ |
| ERM | 0.907±0.002 | 0.948±0.002 | 0.954±0.001 | 0.925±0.002 | 0.951±0.002 | 0.955±0.003 | 0.954±0.003 | 0.955±0.002 | 0.956±0.002 |
| IRM | 0.913±0.003 | 0.952±0.001 | **0.958**±0.001 | 0.932±0.002 | 0.954±0.003 | **0.958**±0.003 | **0.958**±0.001 | **0.959**±0.003 | **0.960**±0.002 |
| CORAL | 0.909±0.002 | 0.948±0.002 | 0.954±0.001 | 0.935±0.002 | 0.954±0.002 | 0.955±0.003 | 0.955±0.002 | 0.955±0.002 | 0.955±0.002 |
| MMD | 0.913±0.002 | 0.950±0.002 | 0.955±0.001 | 0.933±0.003 | 0.946±0.002 | 0.956±0.003 | 0.956±0.002 | 0.957±0.002 | 0.957±0.002 |
| MMD_b_fair | 0.893±0.011 | 0.946±0.002 | 0.953±0.002 | 0.930±0.003 | 0.951±0.002 | 0.954±0.002 | 0.950±0.002 | 0.954±0.001 | 0.955±0.001 |
| Group DRO | 0.915±0.003 | 0.949±0.002 | 0.954±0.001 | 0.937±0.002 | 0.952±0.002 | 0.955±0.003 | 0.955±0.002 | 0.955±0.002 | 0.955±0.003 |
| Reg | 0.931±0.002 | 0.952±0.002 | 0.955±0.002 | 0.945±0.001 | 0.952±0.003 | 0.953±0.002 | 0.951±0.001 | 0.954±0.003 | 0.954±0.002 |
| NC | **0.941**±0.002 | **0.953**±0.002 | 0.957±0.003 | **0.947**±0.002 | **0.955**±0.002 | 0.957±0.002 | 0.956±0.002 | 0.957±0.003 | 0.958±0.002 |

2011 (CUB) dataset (Wah et al., 2011) and backgrounds from the Places dataset (Zhou et al., 2017), while CelebA-$\zeta$ relies on a smaller CelebA subset and applies data resampling to adjust the configuration with images from CelebA. More details on the dataset construction can be found in section 5.1 and the appendix.

Regarding our experiments on data with fixed configurations, we utilize the datasets from a benchmark for subpopulation shift (Yang et al., 2023b), including MetaShift (Liang & Zou, 2022), Waterbirds (Sagawa et al., 2020), and CelebA (Liu et al., 2015), whose configurations exhibit mixtures of shifts. The details of the dataset statistics can be found in the 4 in the appendix. All datasets use two binary attributes. Note that the test data in the datasets provided by the benchmark is neither as shifted as the training data nor fully balanced. To keep consistency in our analyses, we primarily report the *adjusted accuracy* (AA) (Yang et al., 2023b), which averages accuracy across four groups and aligns with test accuracy when the group sizes are equal. In addition, *worst-group accuracy* (WGA), the lowest accuracy across four groups on test data, is provided in the appendix.

**Models and Methods.** We employ a model structure comprised of a pretrained image encoder (ResNet18 (He et al., 2016), ResNet50 (He et al., 2016), and Vision Transformer (ViT) (Dosovitskiy et al., 2021)) and a linear classifier afterwards. The whole network is finetuned on the datasets. In addition to ERM (Vapnik, 1999), we selected several methods that attempt to mitigate shifts in terms of model training, and categorized them by the original focus in their work: (1) spurious correlation: IRM (Arjovsky et al., 2019), Group DRO (Sagawa et al., 2020), and Regularization (Reg) (Wang & Wang, 2024); (2) fairness: MMD_b_fair (Deka & Sutherland, 2023), Neural-collapsed-inspired method (NC) (Lu et al., 2024); (3) domain-invariant learning: MMD (Li et al., 2018), CORAL (Sun & Saenko, 2016). These methods share in commonality that most of them apply regularization or penalty to mitigate the shifts in terms of model training, which aligns with our aim to analyze from the model aspect. Full training details are listed in the appendix.

## 5.2 DATA ASPECT: INSIGHT VALIDATION

We first evaluate the effectiveness of accuracy estimation on the synthetic datasets. Following our discussion in section 4, we apply the performance under two data variants, $\zeta = 0.9$ and $\zeta = 0.999$, to estimate the feature separability, and further estimate the accuracy for the remaining configurations. The configurations with typical shifts and varying $\zeta$ are visualized in the 3-simplex, shown in

Table 2: **Performance on Standard Datasets for Various Methods and Networks.** Gradient colors indicate value trends of empirical separability for the invariant attribute in each column with `lighter` or `darker` shades for lower or higher values. Best-performing methods per column are in **bold**. Reg, Group DRO, and NC are top-performing methods across datasets and networks.

| Method | MetaShift | | | Waterbirds | | | CelebA | | |
|---|---|---|---|---|---|---|---|---|---|
| | ResNet18 | ResNet50 | ViT | ResNet18 | ResNet50 | ViT | ResNet18 | ResNet50 | ViT |
| ERM | 0.853±0.005 | 0.872±0.001 | 0.844±0.009 | 0.769±0.005 | 0.822±0.003 | 0.800±0.023 | 0.796±0.004 | 0.794±0.005 | 0.808±0.014 |
| IRM | 0.850±0.009 | 0.867±0.006 | 0.848±0.008 | 0.782±0.003 | 0.839±0.002 | 0.823±0.030 | 0.792±0.044 | 0.806±0.015 | 0.777±0.051 |
| CORAL | 0.853±0.004 | 0.872±0.003 | 0.870±0.003 | 0.778±0.005 | 0.847±0.004 | 0.841±0.005 | 0.793±0.007 | 0.788±0.005 | 0.807±0.009 |
| MMD | 0.853±0.006 | 0.872±0.003 | 0.859±0.010 | 0.780±0.004 | 0.830±0.004 | 0.890±0.011 | 0.791±0.006 | 0.796±0.005 | 0.810±0.024 |
| MMD_b_fair | 0.857±0.004 | 0.870±0.006 | 0.864±0.007 | 0.791±0.005 | 0.842±0.011 | 0.868±0.011 | 0.793±0.018 | 0.809±0.006 | 0.804±0.011 |
| Group DRO | 0.857±0.004 | 0.877±0.005 | **0.886**±0.004 | 0.826±0.004 | 0.875±0.004 | **0.909**±0.003 | 0.805±0.003 | 0.805±0.005 | 0.826±0.010 |
| Reg | 0.852±0.006 | 0.882±0.001 | 0.844±0.003 | 0.850±0.005 | 0.899±0.001 | 0.871±0.005 | 0.790±0.007 | 0.812±0.019 | 0.816±0.007 |
| NC | **0.860**±0.001 | **0.882**±0.001 | 0.869±0.004 | **0.871**±0.001 | **0.910**±0.000 | 0.891±0.002 | **0.836**±0.005 | **0.832**±0.010 | **0.840**±0.007 |

fig. 4 (a), where the configuration covers the extreme cases on the edges to the balance at the center. The theoretical and empirical performance of these configurations is also displayed in fig. 4 (b-d) for each typical shift, with ResNet50 as the encoder on Waterbirds-$\zeta$. The close alignment indicates the effectiveness of the estimation using our framework, even with the complex image datasets. The skewed data of CI/SC, such as $\zeta = 0.9$, can still achieve a strong performance of 0.907/0.925, compared to the performance of 0.954/0.955 achieved on balanced data. This observation highlights another practical benefit, that when balancing subpopulations is particularly challenging, our framework quantifies the performance and provides an "early stopping" of subpopulation configuration optimization. Thus, our framework provides a quantitative evaluation of performance under subpopulation shifts.

Moreover, table 1 shows the results obtained on datasets with typical shifts and varying shift intensities, which provides prospects to examine the performance trend across methods and shows the framework's potential in terms of characterizing the performance changes. Table 1 shows a similar trend across methods that matches the theoretical characterization. Additionally, we included the performance estimation for competitive mitigation methods, including Group DRO and NC, in Appendix E. The involvement of mitigation methods can amplify the gap between the theoretical and empirical setups, whereas our framework remains effective.

In addition, Table 1 also shows the framework for guiding the usage of the mitigation methods, which where the performance is close to the best achievable performance, the improvement from the mitigation methods becomes marginal. Additional results with other encoders on Waterbirds-$\zeta$ and on CelebA-$\zeta$ are provided in the appendix, showing similar alignment and validating the insights across encoders and datasets.

### 5.3 MODEL ASPECT: INSIGHT VALIDATION

We quantitatively evaluate the empirical separability of the invariant attribute, following section 4, and plot the results against the performance obtained with ResNet50 on various methods in fig. 5. The fitted trend line suggests that methods with higher separability can yield better performance. We note that NC not only regularizes features but also adjusts the linear classifier, which likely explains its deviations from the fitted line.

Further, we evaluate different image encoders and methods described in section 5.1 on the standard datasets. table 2 reports the adjusted accuracy (averaged over 5 seeds) for models trained with different encoders and methods, with the cell colored according to the empirical separability of the invariant attribute. It illustrates the role of empirical invariant separability in the top performance. The results of WGA are provided in the appendix. Across datasets, Reg, Group DRO, and NC achieve the strongest performance. While Group DRO serves as a strong baseline, Reg and NC share a similar principle that by encouraging subgroup separation, which aligns with

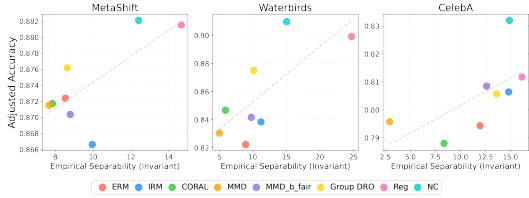

Figure 5: **Empirical Separability (Invariant) vs. Adjust Accuracy of Various Methods.** Higher separability of the invariant attribute correlates with better performance.

the insights of increasing the empirical separability of the invariant attribute, and likely contributes to their effectiveness.

Finally, our framework offers another advantage from the modeling perspective: when synthetic datasets or data variants are available to estimate separability, it allows us to accurately estimate performance under a fully balanced configuration, as shown in fig. 4. This provides a practical upper bound on performance, helping to assess different encoders and guide the selection of strong image encoders for deployment.

## 6 DISCUSSION

We provide a unified perspective on understanding subpopulation shifts in one framework, including class imbalance (CI), spurious correlation (SC), under-representation (UR), or more general cases of mixed shifts. Our analysis of evaluating theoretical performance offers a practical tool for dataset design. Moreover, the theoretical analysis also highlights the key role of feature separability in robustness, which unifies the recent progress in shift mitigation and enables encoder performance comparison. Together, our work evaluates the subpopulation shifts in a unified theoretical framework, providing practical insights for future work on both data and model aspects.

**Broader Impacts.** Our framework is designed as a quantitative tool for dataset design by enabling the estimation of performance for new configurations for the training data. This allows an informed and efficient decision on whether further efforts are needed for the dataset collection. Although separability estimation needs the data variants, it is far more accessible to assess during dataset collection, making our framework practical for guiding the process.

Our theoretical and empirical analysis focuses on the joint training of image encoders and linear classifiers, while examining the separability of the encoded features. Related work has directly transformed features for invariance (Li et al., 2025), or fine-tuned CLIP to prioritize invariant attributes (Yang et al., 2023a), suggesting that extending model-aspect insights more broadly is valuable. Other methods, such as removing non-invariant attributes during zero-shot inference (Saranrittichai et al., 2024), can also be viewed within our framework: they emphasize the invariant attribute over others. We hope that categorizing and explaining such mechanisms will inform future work on unified shift mitigation.

**Limitations and Future Work.** Despite our problem setup covering the cases of multiple attributes, the lack of well-defined shifts and datasets limits broader analysis. Exploring the multi-attribute scenarios is a promising direction for future work. In addition, we model the invariant attribute as binary to enable a closed-form accuracy derivation using the $\text{sign}(\mathbf{w}^T\mathbf{z} + b)$ decision boundary. Extending this derivation to attributes with multiple discrete values, which would require integrating over multi-way decision boundaries, is left for future work.

Additionally, our theoretical and empirical analysis primarily focuses on the model architecture of image encoders and linear classifiers, while examining the separability of the encoded features. Related work (Yang et al., 2023a) has investigated similar directions for finetuning CLIP models, suggesting that extending the insights of model training to multi-modal models or zero-shot methods could be valuable.

## 7 STATEMENTS

As recommended, we include the ethics and reproducibility statements of our work as follows, and a statement of LLM usage is provided in the appendix.

**Ethics Statement.** This work does not raise explicit ethical concerns. We use publicly available datasets, including MNIST, MetaShift, Waterbirds, and CelebA. Our study focuses on understanding the model performance against subpopulation shifts, which helps improve the robustness of the machine learning deployment. However, we acknowledge that the usage of machine learning in the real world should proceed with caution and careful evaluation to avoid any negative consequences.

**Reproducibility Statement.** We are committed to ensuring the reproducibility of our work. All datasets and methods in our work are publicly available, and we include the details of synthetic dataset construction and model training in the appendix. We also include several scripts to demonstrate functionality: (1) accuracy estimation for each shift type in `estimate.py`; (2) preparation for Waterbirds-$\zeta$ in `waterbirds_zeta.py`; (3) preparation for CelebA-$\zeta$ in `celeba_zeta.py`. The full code will be released upon publication.

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

# A   PROOF OF THEOREMS

## A.1   PROOF OF LEMMA 1

**Lemma 1. (Accuracy).**   Given the setting $(\boldsymbol{\mu}, \Sigma, \boldsymbol{\alpha}_\pm, \beta)$ and the model $(\boldsymbol{w}, b)$, the accuracy of the model over the data distribution $\mathcal{D}$ is

$$Acc(\boldsymbol{w}, b) = \frac{1}{2}\left\{1 + \sum_{\boldsymbol{a} \in \{\pm 1\}^N} \left[\kappa_+(\boldsymbol{a}) \operatorname{erf}\left(\frac{\boldsymbol{\mu}_{\boldsymbol{a}}^T \boldsymbol{w} + b}{\sqrt{2\boldsymbol{w}^T \Sigma \boldsymbol{w}}}\right) + \kappa_-(\boldsymbol{a}) \operatorname{erf}\left(\frac{\boldsymbol{\mu}_{\boldsymbol{a}}^T \boldsymbol{w} - b}{\sqrt{2\boldsymbol{w}^T \Sigma \boldsymbol{w}}}\right)\right]\right\} \quad (2)$$

where $\kappa_+(\boldsymbol{a}) = \beta \cdot \prod_{n=1}^N (\alpha_+^{(n)})^{\mathbf{1}_{a_n=1}} (1 - \alpha_+^{(n)})^{\mathbf{1}_{a_n=-1}}$ and $\kappa_-(\boldsymbol{a}) = (1 - \beta) \cdot \prod_{n=1}^N (\alpha_-^{(n)})^{\mathbf{1}_{a_n=-1}} (1 - \alpha_-^{(n)})^{\mathbf{1}_{a_n=1}}$

*Proof.* Given $N$ attributes $a_1, \cdots, a_N$, there are totally $2^N$ groups. Note that the base rate is $\mathbb{P}(Y = 1) = \beta$, then for group $\boldsymbol{a} \in \{\pm 1\}^N$, we have the corresponding group ratio:

$$r_{y,\boldsymbol{a}} = \mathbb{P}(A = \boldsymbol{a}, Y = y) = \mathbb{P}(A_1 = a_1, \cdots, A_N = a_N, Y = y) \quad (3)$$

$$= \left(\prod_{n=1}^N \mathbb{P}(A_i = a_i | Y = y)\right) \mathbb{P}(Y = y) \quad (4)$$

$$= \beta^{\mathbf{1}_{y=1}} (1 - \beta)^{\mathbf{1}_{y=-1}} \prod_{n=1}^N (\alpha^{(n)})^{\mathbf{1}_{a_n=1}} (1 - \alpha^{(n)})^{\mathbf{1}_{a_n=-1}} \quad (5)$$

Conditioned on the class $Y = y$, the distribution of the $n$-th feature $Z_n$ can be written as

$$\begin{cases} Z_n | Y = 1 \sim \alpha_+^{(n)} \mathcal{N}(\boldsymbol{\mu}_n, \Sigma_n) + (1 - \alpha_+^{(n)}) \mathcal{N}(-\boldsymbol{\mu}_n, \Sigma_n) \\ Z_n | Y = -1 \sim (1 - \alpha_-^{(n)}) \mathcal{N}(\boldsymbol{\mu}_n, \Sigma_n) + \alpha_-^{(n)} \mathcal{N}(-\boldsymbol{\mu}_n, \Sigma_n) \end{cases} \quad (6)$$

Hence the corresponding PDFs are:

$$\begin{cases} p(\boldsymbol{z}_n | Y = 1) = \dfrac{\alpha_+^{(n)}}{\sqrt{(2\pi)^{d_n} |\Sigma_a|}} \exp\left(-\frac{1}{2}(\boldsymbol{z}_n - \boldsymbol{\mu}_n)^T \Sigma_n^{-1}(\boldsymbol{z}_n - \boldsymbol{\mu}_n)\right) \\ \qquad\qquad + \dfrac{1 - \alpha_+^{(n)}}{\sqrt{(2\pi)^{d_n} |\Sigma_n|}} \exp\left(-\frac{1}{2}(\boldsymbol{z}_n + \boldsymbol{\mu}_n)^T \Sigma_n^{-1}(\boldsymbol{z}_n + \boldsymbol{\mu}_n)\right) \\ p(\boldsymbol{z}_n | Y = -1) = \dfrac{1 - \alpha_-^{(n)}}{\sqrt{(2\pi)^{d_n} |\Sigma_n|}} \exp\left(-\frac{1}{2}(\boldsymbol{z}_n - \boldsymbol{\mu}_n)^T \Sigma_n^{-1}(\boldsymbol{z}_n - \boldsymbol{\mu}_n)\right) \\ \qquad\qquad + \dfrac{\alpha_-^{(n)}}{\sqrt{(2\pi)^{d_n} |\Sigma_n|}} \exp\left(-\frac{1}{2}(\boldsymbol{z}_n + \boldsymbol{\mu}_n)^T \Sigma_n^{-1}(\boldsymbol{z}_n + \boldsymbol{\mu}_n)\right) \end{cases} \quad (7)$$

And therefore joint distribution of all $N$ features are written as

$$p(\boldsymbol{z}|Y=1) = p(\boldsymbol{z}_1, \cdots, \boldsymbol{z}_N | Y = 1) = \prod_{n=1}^{N} p(\boldsymbol{z}_n | Y = 1) \tag{8}$$

$$= \frac{1}{\sqrt{(2\pi)^{\sum_{n=1}^{N} d_i} \prod_{n=1}^{N} |\Sigma_n|}} \sum_{\boldsymbol{a} \in \{\pm 1\}^N} \left\{ \left[ \prod_{n=1}^{N} \texttt{coef}_+^{(n)}(\boldsymbol{a}) \right] \exp\left( -\frac{1}{2} \texttt{index}_+^{(n)}(\boldsymbol{a}) \right) \right\} \tag{9}$$

$$= \frac{1}{\sqrt{(2\pi)^d |\Sigma|}} \sum_{\boldsymbol{a} \in \{\pm 1\}^N} \left\{ \left[ \prod_{n=1}^{N} \texttt{coef}_+^{(n)}(\boldsymbol{a}) \right] \exp\left( -\frac{1}{2} \texttt{index}_+^{(n)}(\boldsymbol{a}) \right) \right\} \tag{10}$$

$$p(\boldsymbol{z}|Y=-1) = p(\boldsymbol{z}_1, \cdots, \boldsymbol{z}_N | Y = -1) = \prod_{n=1}^{N} p(\boldsymbol{z}_n | Y = 1) \tag{11}$$

$$= \frac{1}{\sqrt{(2\pi)^{\sum_{n=1}^{N} d_i} \prod_{n=1}^{N} |\Sigma_n|}} \sum_{\boldsymbol{a} \in \{\pm 1\}^N} \left\{ \left[ \prod_{n=1}^{N} \texttt{coef}_-^{(n)}(\boldsymbol{a}) \right] \exp\left( -\frac{1}{2} \texttt{index}_-(\boldsymbol{a}) \right) \right\} \tag{12}$$

$$\tag{13}$$

$$= \frac{1}{\sqrt{(2\pi)^d |\Sigma|}} \sum_{\boldsymbol{a} \in \{\pm 1\}^N} \left\{ \left[ \prod_{n=1}^{N} \texttt{coef}_-^{(n)}(\boldsymbol{a}) \right] \exp\left( -\frac{1}{2} \texttt{index}_-(\boldsymbol{a}) \right) \right\} \tag{14}$$

$$\tag{15}$$

where $\texttt{coef}_{\pm}^{(n)}(\boldsymbol{a})$ and $\texttt{index}_{\pm}(\boldsymbol{a})$ are the composed coefficients and indexes of the exponential terms. In detail, given $\boldsymbol{a} \in \mathcal{A}$, the index terms are determined as follows:

$$\begin{cases} \texttt{index}_+(\boldsymbol{a}) = \sum_{n=1}^{N} (\boldsymbol{z}_n - a_n \boldsymbol{\mu}_n)^T \Sigma_n^{-1} (\boldsymbol{z}_n - a_n \boldsymbol{\mu}_n) \\ \texttt{index}_-(\boldsymbol{a}) = \sum_{n=1}^{N} (\boldsymbol{z}_n - a_n \boldsymbol{\mu}_n)^T \Sigma_n^{-1} (\boldsymbol{z}_n - a_n \boldsymbol{\mu}_n) \end{cases} \tag{16}$$

which are identical across different classes. This is because the geometric features are assumed to be irrelevant to the class. Leveraging the compact notations $\boldsymbol{\mu_a}$ and $\Sigma$, this can be further simplified to:

$$\texttt{index}_+(\boldsymbol{a}) = \texttt{index}_-(\boldsymbol{a}) = (\boldsymbol{z} - \boldsymbol{\mu_a})^T \Sigma^{-1} (\boldsymbol{z} - \boldsymbol{\mu_a}) \tag{17}$$

On the other hand, the coefficient terms depend on the classes $y$. Given the attribute $\boldsymbol{a}$, when $y = 1$, if $a_n = 1$, then the $n$-th term's coefficient is $\alpha_+^{(n)}$, otherwise it is $1 - \alpha_+^{(n)}$. Differently, when $y = -1$, if $a_n = 1$, then the $n$-th term's coefficient is $1 - \alpha_-^{(n)}$ and $\alpha_-^{(n)}$ otherwise. This is because $\alpha$s denote the conditional probabilities of "attributes being identical to the label". As a result, the group correlations are detailed as follows:

$$\begin{cases} \texttt{coef}_+^{(n)}(\boldsymbol{a}) = (\alpha_+^{(n)})^{\mathbf{1}_{a_n=1}} (1 - \alpha_+^{(n)})^{\mathbf{1}_{a_n=-1}} \\ \texttt{coef}_-^{(n)}(\boldsymbol{a}) = (\alpha_-^{(n)})^{\mathbf{1}_{a_n=-1}} (1 - \alpha_-^{(n)})^{\mathbf{1}_{a_n=1}} \end{cases} \tag{18}$$

The overall accuracy of $(\boldsymbol{w}, b)$ can be decomposed as follows:

$$Acc(\boldsymbol{w}, b) = \mathbb{P}(Y = 1, \boldsymbol{w}^T \boldsymbol{z} + b > 0) + P(Y = -1, \boldsymbol{w}^T \boldsymbol{z} + b < 0) \tag{19}$$

$$= \mathbb{P}(\boldsymbol{w}^T \boldsymbol{z} + b > 0 | Y = 1)\mathbb{P}(Y = 1) + \mathbb{P}(\boldsymbol{w}^T \boldsymbol{z} + b < 0 | Y = -1)\mathbb{P}(Y = -1) \tag{20}$$

$$= \beta \mathbb{P}(\boldsymbol{w}^T \boldsymbol{z} + b > 0 | Y = 1) + (1 - \beta)\mathbb{P}(\boldsymbol{w}^T \boldsymbol{z} + b < 0 | Y = -1) \tag{21}$$

Now we consider the True Positive Rate of model $(\boldsymbol{w}, b)$, which is

$$Acc_{y=1} = \mathbb{P}(\boldsymbol{w}^T Z + b > 0 | Y = 1) \tag{22}$$

This is essentially the integral of $p(\boldsymbol{z}|Y=1)$ over the half space $\{\boldsymbol{z}|\boldsymbol{w}^T\boldsymbol{z}+b \geq 0\}$:

$$Acc_{y=1} = \int_{\{\boldsymbol{z}|\boldsymbol{w}^T\boldsymbol{z}+b>0\}} p(\boldsymbol{z}|Y=1)\mathrm{d}\boldsymbol{z} \tag{23}$$

$$= \int_{\{\boldsymbol{z}|\boldsymbol{w}^T\boldsymbol{z}+b>0\}} \frac{1}{\sqrt{(2\pi)^d|\Sigma|}} \sum_{\boldsymbol{a}\in\{\pm 1\}^N} \left\{ \left[ \prod_{n=1}^{N} \mathrm{coef}_+^{(n)}(\boldsymbol{a}) \right] \exp\left( -\frac{1}{2}(\boldsymbol{z}-\boldsymbol{\mu_a})^T\Sigma^{-1}(\boldsymbol{z}-\boldsymbol{\mu_a}) \right) \right\} \mathrm{d}\boldsymbol{z} \tag{24}$$

$$= \frac{1}{\sqrt{(2\pi)^d|\Sigma|}} \sum_{\boldsymbol{a}\in\{\pm 1\}^N} \left\{ \left[ \prod_{n=1}^{N} \mathrm{coef}_+^{(n)}(\boldsymbol{a}) \right] \int_{\{\boldsymbol{z}|\boldsymbol{w}^T\boldsymbol{z}+b>0\}} \exp\left( -\frac{1}{2}(\boldsymbol{z}-\boldsymbol{\mu_a})^T\Sigma^{-1}(\boldsymbol{z}-\boldsymbol{\mu_a}) \right)\mathrm{d}\boldsymbol{z} \right\} \tag{25}$$

It suffices to compute

$$\int_{\{\boldsymbol{z}|\boldsymbol{w}^T\boldsymbol{z}+b>0\}} \exp\left( -\frac{1}{2}(\boldsymbol{z}-\boldsymbol{\mu_a})^T\Sigma^{-1}(\boldsymbol{z}-\boldsymbol{\mu_a}) \right)\mathrm{d}\boldsymbol{z} \tag{26}$$

$$= \int_{\{\boldsymbol{z}|\boldsymbol{w}^T\boldsymbol{z}+b>0\}} \exp\left( -\frac{1}{2}(\boldsymbol{z}-\boldsymbol{\mu_a})^T(U^T D^{-1} U)(\boldsymbol{z}-\boldsymbol{\mu_a}) \right)\mathrm{d}\boldsymbol{z} \tag{27}$$

$$\overset{\boldsymbol{n}=U(\boldsymbol{z}-\boldsymbol{\mu_a})}{=\!=\!=\!=\!=\!=\!=} \int_{\{\boldsymbol{n}|\boldsymbol{w}^T(U^T\boldsymbol{n}+\boldsymbol{\mu_a})+b>0\}} \exp\left( -\frac{1}{2}\boldsymbol{n}^T D^{-1}\boldsymbol{n} \right)\mathrm{d}(U^T\boldsymbol{n}+\boldsymbol{\mu_a}) \tag{28}$$

$$= \int_{\{\boldsymbol{n}|\boldsymbol{w}^T(U^T\boldsymbol{n}+\boldsymbol{\mu_a})+b>0\}} \exp\left( -\frac{1}{2}\boldsymbol{n}^T D^{-1}\boldsymbol{n} \right)|U^T|\mathrm{d}\boldsymbol{n} \tag{29}$$

$$= \int_{\{\boldsymbol{n}|\boldsymbol{w}^T(U^T\boldsymbol{n}+\boldsymbol{\mu_a})+b>0\}} \exp\left( -\frac{1}{2}\boldsymbol{n}^T (\sqrt{D^{-1}})^T \sqrt{D^{-1}}\boldsymbol{n} \right)\mathrm{d}\boldsymbol{n} \tag{30}$$

$$\overset{\boldsymbol{m}=\sqrt{D^{-1}}\boldsymbol{n}}{=\!=\!=\!=\!=\!=} \int_{\{\boldsymbol{m}|\boldsymbol{w}^T(U^T\sqrt{D}\boldsymbol{m}+\boldsymbol{\mu_a})+b>0\}} \exp\left( -\frac{1}{2}\boldsymbol{m}^T\boldsymbol{m} \right)|\sqrt{D}|\mathrm{d}\boldsymbol{m} \tag{31}$$

Let $\boldsymbol{u} = O\boldsymbol{m}$ for a special orthogonal matrix such that

$$O(\sqrt{D}U\boldsymbol{w}) = \|\sqrt{D}U\boldsymbol{w}\|\boldsymbol{e}_1 \tag{32}$$

Then

$$\{\boldsymbol{m}|\boldsymbol{w}^T(U^T\sqrt{D}\boldsymbol{m}+\boldsymbol{\mu_a})+b>0\} = \{\boldsymbol{m}|\boldsymbol{m}^T(\sqrt{D}U\boldsymbol{w})+\boldsymbol{\mu_a}^T\boldsymbol{w}+b>0\} \tag{33}$$

$$= \{\boldsymbol{u}|(O^T\boldsymbol{u})^T(\sqrt{D}U\boldsymbol{w})+\boldsymbol{\mu_a}^T\boldsymbol{w}+b>0\} \tag{34}$$

$$= \{\boldsymbol{u}|\boldsymbol{u}^T O(\sqrt{D}U\boldsymbol{w})+\boldsymbol{\mu_a}^T\boldsymbol{w}+b>0\} \tag{35}$$

$$= \{\boldsymbol{u}|\boldsymbol{u}^T\|\sqrt{D}U\boldsymbol{w}\|\boldsymbol{e}_1+\boldsymbol{\mu_a}^T\boldsymbol{w}+b>0\} \tag{36}$$

$$= \{\boldsymbol{u}|\|\sqrt{D}U\boldsymbol{w}\|u_1+\boldsymbol{\mu_a}^T\boldsymbol{w}+b>0\} \tag{37}$$

$$= \{\boldsymbol{u}|u_1 > -\frac{\boldsymbol{\mu_a}^T\boldsymbol{w}+b}{\|\sqrt{D}U\boldsymbol{w}\|}\} \tag{38}$$

$$= (-\frac{\boldsymbol{\mu_a}^T\boldsymbol{w}+b}{\|\sqrt{D}U\boldsymbol{w}\|}, \infty] \times \mathbb{R}^{d-1} \tag{39}$$

$$= (-\frac{\boldsymbol{\mu_a}^T\boldsymbol{w}+b}{\sqrt{\boldsymbol{w}^T\Sigma\boldsymbol{w}}}, \infty] \times \mathbb{R}^{d-1} \tag{40}$$

As a result, we have

$$\int_{\{z|\boldsymbol{w}^T\boldsymbol{z}+b>0\}} \exp\Big(-\frac{1}{2}(\boldsymbol{z}-\boldsymbol{\mu_a})^T\Sigma^{-1}(\boldsymbol{z}-\boldsymbol{\mu_a})\Big)\mathrm{d}\boldsymbol{z} \tag{41}$$

$$=\int_{\{\boldsymbol{m}|\boldsymbol{w}^T(U^T\sqrt{D}\boldsymbol{m}+\boldsymbol{\mu_a})+b>0\}} \exp\Big(-\frac{1}{2}\boldsymbol{m}^T\boldsymbol{m}\Big)|\sqrt{D}|\mathrm{d}\boldsymbol{m} \tag{42}$$

$$=\sqrt{|\Sigma|}\int_{(-\frac{\boldsymbol{\mu_a}^T\boldsymbol{w}+b}{\sqrt{\boldsymbol{w}^T\Sigma\boldsymbol{w}}},\infty]\times\mathbb{R}^{d-1}} \exp(-\frac{1}{2}\boldsymbol{u}^T\boldsymbol{u})|O^T|\mathrm{d}\boldsymbol{u} \tag{43}$$

$$=\sqrt{|\Sigma|}\sqrt{(2\pi)^{d-1}}\int_{-\frac{\boldsymbol{\mu_a}^T\boldsymbol{w}+b}{\sqrt{\boldsymbol{w}^T\Sigma\boldsymbol{w}}}}^{\infty} \exp(-\frac{1}{2}u^2)\mathrm{d}u \tag{44}$$

$$=\sqrt{|\Sigma|}\sqrt{(2\pi)^{d-1}}\sqrt{\frac{\pi}{2}}\Big(1-\mathrm{erf}(-\frac{\boldsymbol{\mu_a}^T\boldsymbol{w}+b}{\sqrt{2\boldsymbol{w}^T\Sigma\boldsymbol{w}}})\Big) \tag{45}$$

Substitute eq. (45) back to eq. (25), we have

$$Acc_{y=1}=\frac{1}{\sqrt{(2\pi)^d|\Sigma|}}\sum_{\boldsymbol{a}\in\{\pm1\}^N}\Big\{\Big[\prod_{n=1}^{N}\mathtt{coef}_+^{(n)}(\boldsymbol{a})\Big]\sqrt{|\Sigma|}\sqrt{(2\pi)^{d-1}}\sqrt{\frac{\pi}{2}}\Big(1-\mathrm{erf}(-\frac{\boldsymbol{\mu_a}^T\boldsymbol{w}+b}{\sqrt{2\boldsymbol{w}^T\Sigma\boldsymbol{w}}})\Big)\Big\}$$

$$\tag{46}$$

$$=\frac{\sqrt{|\Sigma|}\sqrt{(2\pi)^{d-1}}}{\sqrt{(2\pi)^d|\Sigma|}}\sqrt{\frac{\pi}{2}}\sum_{\boldsymbol{a}\in\{\pm1\}^N}\Big\{\Big[\prod_{n=1}^{N}\mathtt{coef}_+^{(n)}(\boldsymbol{a})\Big]\Big(1-\mathrm{erf}(-\frac{\boldsymbol{\mu_a}^T\boldsymbol{w}+b}{\sqrt{2\boldsymbol{w}^T\Sigma\boldsymbol{w}}})\Big)\Big\} \tag{47}$$

$$=\frac{1}{2}\sum_{\boldsymbol{a}\in\{\pm1\}^N}\Big\{\Big[\prod_{n=1}^{N}\mathtt{coef}_+^{(n)}(\boldsymbol{a})\Big]\Big(1-\mathrm{erf}(-\frac{\boldsymbol{\mu_a}^T\boldsymbol{w}+b}{\sqrt{2\boldsymbol{w}^T\Sigma\boldsymbol{w}}})\Big)\Big\} \tag{48}$$

$$=\frac{1}{2}\Big\{1+\sum_{\boldsymbol{a}\in\{\pm1\}^N}\Big[\big(\prod_{n=1}^{N}\mathtt{coef}_+^{(n)}(\boldsymbol{a})\big)\,\mathrm{erf}\,\big(\frac{\boldsymbol{\mu_a}^T\boldsymbol{w}+b}{\sqrt{2\boldsymbol{w}^T\Sigma\boldsymbol{w}}}\big)\Big]\Big\} \tag{49}$$

Similarly, the True Negative Rate can be written as

$$Acc_{y=-1}=\int_{\{z|\boldsymbol{w}^T\boldsymbol{z}+b<0\}} \exp\Big(-\frac{1}{2}(\boldsymbol{z}-\boldsymbol{\mu_a})^T\Sigma^{-1}(\boldsymbol{z}-\boldsymbol{\mu_a})\Big)\mathrm{d}\boldsymbol{z} \tag{50}$$

$$=\frac{1}{2}\Big\{1+\sum_{\boldsymbol{a}\in\{\pm1\}^N}\Big[\big(\prod_{n=1}^{N}\mathtt{coef}_-^{(n)}(\boldsymbol{a})\big)\,\mathrm{erf}\,\big(\frac{\boldsymbol{\mu_a}^T\boldsymbol{w}-b}{\sqrt{2\boldsymbol{w}^T\Sigma\boldsymbol{w}}}\big)\Big]\Big\} \tag{51}$$

Finally, substitute eqs. (49) and (51) back to eq. (21), we have

$$Acc(\boldsymbol{w},b)=\frac{\beta}{2}\Big\{1+\sum_{\boldsymbol{a}\in\{\pm1\}^N}\Big[\big(\prod_{n=1}^{N}\mathtt{coef}_+^{(n)}(\boldsymbol{a})\big)\,\mathrm{erf}\,\big(\frac{\boldsymbol{\mu_a}^T\boldsymbol{w}+b}{\sqrt{2\boldsymbol{w}^T\Sigma\boldsymbol{w}}}\big)\Big]\Big\} \tag{52}$$

$$+\frac{1-\beta}{2}\Big\{1+\sum_{\boldsymbol{a}\in\{\pm1\}^N}\Big[\big(\prod_{n=1}^{N}\mathtt{coef}_-^{(n)}(\boldsymbol{a})\big)\,\mathrm{erf}\,\big(\frac{\boldsymbol{\mu_a}^T\boldsymbol{w}-b}{\sqrt{2\boldsymbol{w}^T\Sigma\boldsymbol{w}}}\big)\Big]\Big\} \tag{53}$$

$$=\frac{1}{2}\Big\{1+\sum_{\boldsymbol{a}\in\{\pm1\}^N}\Big[\kappa_+(\boldsymbol{a})\,\mathrm{erf}\,\big(\frac{\boldsymbol{\mu_a}^T\boldsymbol{w}+b}{\sqrt{2\boldsymbol{w}^T\Sigma\boldsymbol{w}}}\big)+\kappa_-(\boldsymbol{a})\,\mathrm{erf}\,\big(\frac{\boldsymbol{\mu_a}^T\boldsymbol{w}-b}{\sqrt{2\boldsymbol{w}^T\Sigma\boldsymbol{w}}}\big)\Big]\Big\} \tag{54}$$

where we denote

$$\begin{cases} \kappa_+(\boldsymbol{a})=\beta\cdot\prod_{n=1}^{N}\mathtt{coef}_+^{(n)}(\boldsymbol{a}) \\[2em] \kappa_-(\boldsymbol{a})=(1-\beta)\cdot\prod_{n=1}^{N}\mathtt{coef}_-^{(n)}(\boldsymbol{a}) \end{cases} \tag{55}$$

for brevity. And hence the lemma is proved.

$\square$

## A.2 PROOF OF LEMMA 2

**Lemma 2. (Colinearity)** The segment $\boldsymbol{w}_n^*$ of the Bayesian optimal classifier for the $n$-th feature $\boldsymbol{z}_n$ is colinear with $\Sigma_n^{-1}\boldsymbol{\mu}_n$.

*Proof.* In order to find the stationary point, we start by letting $\frac{\partial Acc(\boldsymbol{w},b)}{\partial b} = 0$. The partial derivative can be written as

$$\frac{\partial Acc(\boldsymbol{w},b)}{\partial b} = \frac{1}{2}\frac{\partial}{\partial b}\left\{ \sum_{\boldsymbol{a}\in\{\pm 1\}^N}\left[\kappa_+(\boldsymbol{a})\operatorname{erf}\left(\frac{\boldsymbol{\mu}_{\boldsymbol{a}}^T\boldsymbol{w}+b}{\sqrt{2\boldsymbol{w}^T\Sigma\boldsymbol{w}}}\right) + \kappa_-(\boldsymbol{a})\operatorname{erf}\left(\frac{\boldsymbol{\mu}_{\boldsymbol{a}}^T\boldsymbol{w}-b}{\sqrt{2\boldsymbol{w}^T\Sigma\boldsymbol{w}}}\right)\right]\right\} \tag{56}$$

$$= \frac{1}{2}\sum_{\boldsymbol{a}\in\{\pm 1\}^N}\left[\kappa_+(\boldsymbol{a})\exp\left(-\frac{(\boldsymbol{\mu}_{\boldsymbol{a}}^T\boldsymbol{w}+b)^2}{2\boldsymbol{w}^T\Sigma\boldsymbol{w}}\right) - \kappa_-(\boldsymbol{a})\exp\left(-\frac{(\boldsymbol{\mu}_{\boldsymbol{a}}^T\boldsymbol{w}-b)^2}{2\boldsymbol{w}^T\Sigma\boldsymbol{w}}\right)\right] \tag{57}$$

Thus $\frac{\partial Acc(\boldsymbol{w},b)}{\partial b} = 0$ is equivalent to

$$\sum_{\boldsymbol{a}\in\mathcal{A}}\left[\kappa_+(\boldsymbol{a})\exp\left(-\frac{(\boldsymbol{\mu}_{\boldsymbol{a}}^T\boldsymbol{w}+b)^2}{2\boldsymbol{w}^T\Sigma\boldsymbol{w}}\right)\right] = \sum_{\boldsymbol{a}\in\mathcal{A}}\left[\kappa_-(\boldsymbol{a})\exp\left(-\frac{(\boldsymbol{\mu}_{\boldsymbol{a}}^T\boldsymbol{w}-b)^2}{2\boldsymbol{w}^T\Sigma\boldsymbol{w}}\right)\right] \tag{58}$$

As for the weight $\boldsymbol{w}$, we also have $\frac{\partial Acc}{\partial \boldsymbol{w}} = \boldsymbol{0}$. This is equivalent to

$$\boldsymbol{0} = \frac{\partial}{\partial \boldsymbol{w}}\left\{ \sum_{\boldsymbol{a}\in\{\pm 1\}^N}\left[\kappa_+(\boldsymbol{a})\operatorname{erf}\left(\frac{\boldsymbol{\mu}_{\boldsymbol{a}}^T\boldsymbol{w}+b}{\sqrt{2\boldsymbol{w}^T\Sigma\boldsymbol{w}}}\right) + \kappa_-(\boldsymbol{a})\operatorname{erf}\left(\frac{\boldsymbol{\mu}_{\boldsymbol{a}}^T\boldsymbol{w}-b}{\sqrt{2\boldsymbol{w}^T\Sigma\boldsymbol{w}}}\right)\right]\right\} \tag{59}$$

$$= \sum_{\boldsymbol{a}\in\{\pm 1\}^N}\left[\kappa_+(\boldsymbol{a})\exp\left(-\frac{(\boldsymbol{\mu}_{\boldsymbol{a}}^T\boldsymbol{w}+b)^2}{2\boldsymbol{w}^T\Sigma\boldsymbol{w}}\right)\frac{(\Sigma\boldsymbol{w}\boldsymbol{w}^T - \boldsymbol{w}^T\Sigma\boldsymbol{w}I)\boldsymbol{\mu}_{\boldsymbol{a}} + \Sigma\boldsymbol{w}b}{(\boldsymbol{w}^T\Sigma\boldsymbol{w})^{3/2}}\right. \tag{60}$$

$$\left. + \kappa_-(\boldsymbol{a})\exp\left(-\frac{(\boldsymbol{\mu}_{\boldsymbol{a}}^T\boldsymbol{w}-b)^2}{2\boldsymbol{w}^T\Sigma\boldsymbol{w}}\right)\frac{(\Sigma\boldsymbol{w}\boldsymbol{w}^T - \boldsymbol{w}^T\Sigma\boldsymbol{w}I)\boldsymbol{\mu}_{\boldsymbol{a}} - \Sigma\boldsymbol{w}b}{(\boldsymbol{w}^T\Sigma\boldsymbol{w})^{3/2}}\right] \tag{61}$$

$$= \sum_{\boldsymbol{a}\in\{\pm 1\}^N}\left[\kappa_+(\boldsymbol{a})\exp\left(-\frac{(\boldsymbol{\mu}_{\boldsymbol{a}}^T\boldsymbol{w}+b)^2}{2\boldsymbol{w}^T\Sigma\boldsymbol{w}}\right)\frac{(\Sigma\boldsymbol{w}\boldsymbol{w}^T - \boldsymbol{w}^T\Sigma\boldsymbol{w}I)\boldsymbol{\mu}_{\boldsymbol{a}}}{(\boldsymbol{w}^T\Sigma\boldsymbol{w})^{3/2}}\right. \tag{62}$$

$$\left. + \kappa_-(\boldsymbol{a})\exp\left(-\frac{(\boldsymbol{\mu}_{\boldsymbol{a}}^T\boldsymbol{w}-b)^2}{2\boldsymbol{w}^T\Sigma\boldsymbol{w}}\right)\frac{(\Sigma\boldsymbol{w}\boldsymbol{w}^T - \boldsymbol{w}^T\Sigma\boldsymbol{w}I)\boldsymbol{\mu}_{\boldsymbol{a}}}{(\boldsymbol{w}^T\Sigma\boldsymbol{w})^{3/2}}\right] + \frac{\partial Acc}{\partial b}\cdot\Sigma\boldsymbol{w} \tag{63}$$

$$\overset{\partial Acc/\partial b=0}{=\!=\!=\!=\!=}(\Sigma\boldsymbol{w}\boldsymbol{w}^T - \boldsymbol{w}^T\Sigma\boldsymbol{w}I)\left\{\sum_{\boldsymbol{a}\in\{\pm 1\}^N}\left[\kappa_+(\boldsymbol{a})\exp\left(-\frac{(\boldsymbol{\mu}_{\boldsymbol{a}}^T\boldsymbol{w}+b)^2}{2\boldsymbol{w}^T\Sigma\boldsymbol{w}}\right)\right.\right. \tag{64}$$

$$\left.\left. + \kappa_-(\boldsymbol{a})\exp\left(-\frac{(\boldsymbol{\mu}_{\boldsymbol{a}}^T\boldsymbol{w}-b)^2}{2\boldsymbol{w}^T\Sigma\boldsymbol{w}}\right)\right]\boldsymbol{\mu}_{\boldsymbol{a}}\right\} \tag{65}$$

$$\overset{\{\cdot\}=\boldsymbol{q}}{=\!=\!=}\Sigma\boldsymbol{w}\boldsymbol{w}^T\boldsymbol{q} - (\boldsymbol{w}^T\Sigma\boldsymbol{w})\boldsymbol{q} \tag{66}$$

Thus it suffices to solve

$$(\Sigma\boldsymbol{w}\boldsymbol{w}^T)\boldsymbol{q} = (\boldsymbol{w}^T\Sigma\boldsymbol{w})\boldsymbol{q} \tag{67}$$

On the other hand, note that $\Sigma\boldsymbol{w}\boldsymbol{w}^T$ is a 1-rank matrix, and

$$(\Sigma\boldsymbol{w}\boldsymbol{w}^T)(\Sigma\boldsymbol{w}) = (\Sigma\boldsymbol{w})(\boldsymbol{w}^T\Sigma\boldsymbol{w}) = (\boldsymbol{w}^T\Sigma\boldsymbol{w})\Sigma\boldsymbol{w} \tag{68}$$

This means that $\Sigma\boldsymbol{w}$ is colinear with $\boldsymbol{q}$. Therefore, $\exists\eta\in\mathbb{R}$ s.t. $\Sigma\boldsymbol{w} = \eta\boldsymbol{q}$:

$$\Sigma\boldsymbol{w} = \eta\sum_{\boldsymbol{a}\in\{\pm 1\}^N}\left[\kappa_+(\boldsymbol{a})\exp\left(-\frac{(\boldsymbol{\mu}_{\boldsymbol{a}}^T\boldsymbol{w}+b)^2}{2\boldsymbol{w}^T\Sigma\boldsymbol{w}}\right) + \kappa_-(\boldsymbol{a})\exp\left(-\frac{(\boldsymbol{\mu}_{\boldsymbol{a}}^T\boldsymbol{w}-b)^2}{2\boldsymbol{w}^T\Sigma\boldsymbol{w}}\right)\right]\boldsymbol{\mu}_{\boldsymbol{a}} \tag{69}$$

Taking the $n$-th block, we have

$$\boldsymbol{w}_n = \eta\left\{ \sum_{\boldsymbol{a}\in\{\pm 1\}^N} \left[ \kappa_+(\boldsymbol{a})\exp\left(-\frac{(\boldsymbol{\mu}_{\boldsymbol{a}}^T\boldsymbol{w}+b)^2}{2\boldsymbol{w}^T\Sigma\boldsymbol{w}}\right) + \kappa_-(\boldsymbol{a})\exp\left(-\frac{(\boldsymbol{\mu}_{\boldsymbol{a}}^T\boldsymbol{w}-b)^2}{2\boldsymbol{w}^T\Sigma\boldsymbol{w}}\right) \right] a_n \right\} \Sigma_n^{-1}\boldsymbol{\mu}_n \tag{70}$$

which means that the $n$-th feature's classifier is colinear with $\Sigma_n^{-1}\boldsymbol{\mu}_n$.

$\square$

### A.3 PROOF TO LEMMA 3

**Lemma 3. (Bayesian Group Accuracy)** *The Bayesian optimal classifier achieves group accuracy for $\boldsymbol{a} \in \mathcal{A}$ as*

$$Acc_{\boldsymbol{a}} = \frac{1}{2}\left(1 + \mathrm{erf}\left(\frac{\sum_{n=1}^N a_n\eta_n m_n + y_{\boldsymbol{a}}b}{\sqrt{2\sum_{n=1}^N \eta_n^2 m_n}}\right)\right) \tag{71}$$

*where $\eta_n$ are the coefficients such that $\boldsymbol{w}_n^* = \eta_n\Sigma_n^{-1}\boldsymbol{\mu}_n$, and $m_n = \boldsymbol{\mu}_n^T\Sigma_n^{-1}\boldsymbol{\mu}_n$ are the Mahalanobis distance of the $n$-th feature that determine the **feature separability**.*

*Proof.* Consider the group defined by the attribute $A = \boldsymbol{a} \in \mathcal{A}$. Then the distribution of the group is determined by

$$Z|A = \boldsymbol{a} \sim \mathcal{N}(\boldsymbol{\mu}_{\boldsymbol{a}}, \Sigma) \tag{72}$$

where

$$\boldsymbol{\mu}_{\boldsymbol{a}} = \begin{bmatrix} a_1\boldsymbol{\mu}_1 \\ \vdots \\ a_N\boldsymbol{\mu}_N \end{bmatrix}, \Sigma = \begin{bmatrix} \Sigma_1 & & \\ & \ddots & \\ & & \Sigma_N \end{bmatrix} \tag{73}$$

Then the group accuracy of the group determined by $\boldsymbol{a}$ can be written as

$$Acc_{\boldsymbol{a}} = \mathbb{P}(Y = 1, \boldsymbol{w}^T\boldsymbol{z} + b > 0|A = \boldsymbol{a}) + \mathbb{P}(Y = -1, \boldsymbol{w}^T\boldsymbol{z} + b < 0|A = \boldsymbol{a}) \tag{74}$$

Note that a group $\boldsymbol{a}$ uniquely determines the class $Y = y_{\boldsymbol{a}}$. WLOG, we let $y_{\boldsymbol{a}} = 1$. Thus

$$Acc_{\boldsymbol{a}} = \mathbb{P}(Y = 1, \boldsymbol{w}^T\boldsymbol{z} + b > 0|A = \boldsymbol{a}) \tag{75}$$

$$= \int_{\{\boldsymbol{z}|\boldsymbol{w}^T\boldsymbol{z}+b>0\}} p(\boldsymbol{z}|A = \boldsymbol{a})\boldsymbol{z} \tag{76}$$

$$= \int_{\{\boldsymbol{z}|\boldsymbol{w}^T\boldsymbol{z}+b>0\}} \frac{1}{\sqrt{(2\pi)^d|\Sigma|}} \exp\left(-\frac{1}{2}(\boldsymbol{z} - \boldsymbol{\mu}_{\boldsymbol{a}})^T\Sigma^{-1}(\boldsymbol{z} - \boldsymbol{\mu}_{\boldsymbol{a}})\right) \tag{77}$$

$$= \frac{1}{2}\left(1 + \mathrm{erf}\left(\frac{\boldsymbol{\mu}_{\boldsymbol{a}}^T\boldsymbol{w} + b}{\sqrt{2\boldsymbol{w}^T\Sigma\boldsymbol{w}}}\right)\right) \tag{78}$$

From Lemma 2, the segment $\boldsymbol{w}_n^*$ of the Bayesian optimal classifier for the $n$-th feature $\boldsymbol{z}_n$ is colinear with $\Sigma_n^{-1}\boldsymbol{\mu}_n$. Without loss of generality, let $\eta_n \in \mathbb{R}$ be the coefficients such that $\boldsymbol{w}_n^* = \eta_n\Sigma_n^{-1}\boldsymbol{\mu}_n$. Therefore, the classifier is

$$\boldsymbol{w}^* = \begin{bmatrix} \boldsymbol{w}_1^* \\ \vdots \\ \boldsymbol{w}_N^* \end{bmatrix} = \begin{bmatrix} \eta_1\Sigma_1^{-1}\boldsymbol{\mu}_1 \\ \vdots \\ \eta_N\Sigma_N^{-1}\boldsymbol{\mu}_N \end{bmatrix} \tag{79}$$

Substitute this back to eq. (78), we have

$$Acc_{\boldsymbol{a}} = \frac{1}{2}\left(1 + \mathrm{erf}\left(\frac{\sum_{i=1}^N a_i\boldsymbol{\mu}_i^T\eta_i\Sigma_i^{-1}\boldsymbol{\mu}_i + b}{\sqrt{2(\sum_{i=1}^N \eta_i^2\boldsymbol{\mu}_i^T\Sigma_i^T\boldsymbol{\mu}_i)}}\right)\right) \tag{80}$$

$$= \frac{1}{2}\left(1 + \mathrm{erf}\left(\frac{\sum_{n=1}^N a_n\eta_n m_n + y_{\boldsymbol{a}}b}{\sqrt{2\Sigma_{n=1}^N \eta_n^2 m_n}}\right)\right) \tag{81}$$

Thus the lemma is proved. $\square$

A.4 PROOF OF THEOREM 1

**Theorem 1. (Bayesian Classifier)** When $N = 2$, the Bayesian classifier is $\hat{y} = \boldsymbol{\mu}_1^T \Sigma_1^{-1} \boldsymbol{z}_1 + \tau \boldsymbol{\mu}_2^T \Sigma_2^{-1} \boldsymbol{z}_2 + b$ where $\tau, b \in \mathbb{R}$ are the solutions to the equation system:

$$
\begin{cases}
\tau = \dfrac{\gamma_{1,1}(\tau, b) + \gamma_{-1,-1}(\tau, b) - \gamma_{1,-1}(\tau, b) - \gamma_{-1,1}(\tau, b)}{\gamma_{1,1}(\tau, b) + \gamma_{-1,-1}(\tau, b) + \gamma_{1,-1}(\tau, b) + \gamma_{-1,1}(\tau, b)} \\
\gamma_{1,1}(\tau, b) + \gamma_{1,-1}(\tau, b) = \gamma_{-1,-1}(\tau, b) + \gamma_{-1,1}(\tau, b)
\end{cases}
\tag{82}
$$

Here $\gamma$s are functions of $\tau, b$:

$$
\begin{cases}
\gamma_{1,1}(\tau, b) = \alpha_+^{(2)} \beta \exp\left( - \dfrac{(m_1 + \tau m_2 + b)^2}{2(m_1 + \tau^2 m_2)} \right) \\[2mm]
\gamma_{-1,-1}(\tau, b) = \alpha_-^{(2)}(1 - \beta) \exp\left( - \dfrac{(m_1 + \tau m_2 - b)^2}{2(m_1 + \tau^2 m_2)} \right) \\[2mm]
\gamma_{1,-1}(\tau, b) = (1 - \alpha_+^{(2)}) \beta \exp\left( - \dfrac{(m_1 - \tau m_2 + b)^2}{2(m_1 + \tau^2 m_2)} \right) \\[2mm]
\gamma_{-1,1}(\tau, b) = (1 - \alpha_-^{(2)})(1 - \beta) \exp\left( - \dfrac{(m_1 - \tau m_2 - b)^2}{2(m_1 + \tau^2 m_2)} \right)
\end{cases}
\tag{83}
$$

*Proof.* From section A.2, we already have

$$
\boldsymbol{w}_n = \left\{ \sum_{\boldsymbol{a} \in \{\pm 1\}^N} \left[ \kappa_+(\boldsymbol{a}) \exp\left( - \frac{(\boldsymbol{\mu}_a^T \boldsymbol{w} + b)^2}{2\boldsymbol{w}^T \Sigma \boldsymbol{w}} \right) + \kappa_-(\boldsymbol{a}) \exp\left( - \frac{(\boldsymbol{\mu}_a^T \boldsymbol{w} - b)^2}{2\boldsymbol{w}^T \Sigma \boldsymbol{w}} \right) \right] a_n \right\} \Sigma_n^{-1} \boldsymbol{\mu}_n
\tag{84}
$$

Now that $N = 2$, $\boldsymbol{a} \in \mathcal{A} = \{(1,1), (1,-1), (-1,-1), (-1,1)\}$. Then the coefficients $\kappa$s are simplified as

$$
y = 1 : \begin{cases}
\kappa_+(1, 1) = \beta \alpha_+^{(1)} \alpha_+^{(2)} = \beta \alpha_+^{(2)} \\
\kappa_+(1, -1) = \beta \alpha_+^{(1)}(1 - \alpha_+^{(2)}) = \beta(1 - \alpha_+^{(2)}) \\
\kappa_+(-1, -1) = \beta(1 - \alpha_+^{(1)})(1 - \alpha_+^{(2)}) = 0 \\
\kappa_+(-1, 1) = \beta(1 - \alpha_+^{(1)}) \alpha_+^{(2)} = 0
\end{cases}
\tag{85}
$$

$$
y = -1 : \begin{cases}
\kappa_-(1, 1) = (1 - \beta)(1 - \alpha_-^{(1)})(1 - \alpha_-^{(2)}) = 0 \\
\kappa_-(1, -1) = (1 - \beta)(1 - \alpha_-^{(1)}) \alpha_-^{(2)} = 0 \\
\kappa_-(-1, -1) = (1 - \beta) \alpha_-^{(1)} \alpha_-^{(2)} = (1 - \beta) \alpha_-^{(2)} \\
\kappa_-(-1, 1) = (1 - \beta) \alpha_+^{(1)}(1 - \alpha_+^{(2)}) = (1 - \beta)(1 - \alpha_-^{(2)})
\end{cases}
\tag{86}
$$

Substitute these coefficients back to eq. (84), we have

$$
\begin{bmatrix} \boldsymbol{w}_1 \\ \boldsymbol{w}_2 \end{bmatrix} = \eta \begin{bmatrix} \tau_1 \Sigma_1^{-1} \boldsymbol{\mu}_1 \\ \tau_2 \Sigma_2^{-1} \boldsymbol{\mu}_2 \end{bmatrix}
\tag{87}
$$

where

$$
\begin{cases}
\tau_1 = \eta \left[ \alpha_+^{(2)} \beta \exp\left( - \dfrac{(\boldsymbol{\mu}_1^T \boldsymbol{w}_1 + \boldsymbol{\mu}_2^T \boldsymbol{w}_2 + b)^2}{2\boldsymbol{w}^T \Sigma \boldsymbol{w}} \right) + \alpha_-^{(2)}(1 - \beta) \exp\left( - \dfrac{(\boldsymbol{\mu}_1^T \boldsymbol{w}_1 + \boldsymbol{\mu}_2^T \boldsymbol{w}_2 - b)^2}{2\boldsymbol{w}^T \Sigma \boldsymbol{w}} \right) \right] \\[2mm]
\quad + \left[ (1 - \alpha_+^{(2)}) \beta \exp\left( - \dfrac{(\boldsymbol{\mu}_1^T \boldsymbol{w}_1 - \boldsymbol{\mu}_2^T \boldsymbol{w}_2 + b)^2}{2\boldsymbol{w}^T \Sigma \boldsymbol{w}} \right) + (1 - \alpha_-^{(2)})(1 - \beta) \exp\left( - \dfrac{(\boldsymbol{\mu}_1^T \boldsymbol{w}_1 - \boldsymbol{\mu}_2^T \boldsymbol{w}_2 - b)^2}{2\boldsymbol{w}^T \Sigma \boldsymbol{w}} \right) \right] \\[2mm]
\tau_2 = \eta \left[ \alpha_+^{(2)} \beta \exp\left( - \dfrac{(\boldsymbol{\mu}_1^T \boldsymbol{w}_1 + \boldsymbol{\mu}_2^T \boldsymbol{w}_2 + b)^2}{2\boldsymbol{w}^T \Sigma \boldsymbol{w}} \right) + \alpha_-^{(2)}(1 - \beta) \exp\left( - \dfrac{(\boldsymbol{\mu}_1^T \boldsymbol{w}_1 + \boldsymbol{\mu}_2^T \boldsymbol{w}_2 - b)^2}{2\boldsymbol{w}^T \Sigma \boldsymbol{w}} \right) \right] \\[2mm]
\quad - \left[ (1 - \alpha_+^{(2)}) \beta \exp\left( - \dfrac{(\boldsymbol{\mu}_1^T \boldsymbol{w}_1 - \boldsymbol{\mu}_2^T \boldsymbol{w}_2 + b)^2}{2\boldsymbol{w}^T \Sigma \boldsymbol{w}} \right) + (1 - \alpha_-^{(2)})(1 - \beta) \exp\left( - \dfrac{(\boldsymbol{\mu}_1^T \boldsymbol{w}_1 - \boldsymbol{\mu}_2^T \boldsymbol{w}_2 - b)^2}{2\boldsymbol{w}^T \Sigma \boldsymbol{w}} \right) \right]
\end{cases}
\tag{88}
$$

WLOG, let $\eta = 1/\tau_1$ and $\tau = \tau_2/\tau_1$, then the solution can be simplified to:

$$\begin{bmatrix} \boldsymbol{w}_1 \\ \boldsymbol{w}_2 \end{bmatrix} = \begin{bmatrix} \Sigma_1^{-1}\boldsymbol{\mu}_1 \\ \tau\Sigma_2^{-1}\boldsymbol{\mu}_2 \end{bmatrix} \tag{89}$$

It suffices to solve for

$$\tau = \frac{\tau_2}{\tau_1} = \frac{\gamma_{1,1} + \gamma_{-1,-1} - \gamma_{1,-1} - \gamma_{-1,1}}{\gamma_{1,1} + \gamma_{-1,-1} + \gamma_{1,-1} + \gamma_{-1,1}} \tag{90}$$

where

$$\begin{cases} \gamma_{1,1} = \alpha_1\beta\exp\big(-\dfrac{(\boldsymbol{\mu}_1^T\boldsymbol{w}_1 + \boldsymbol{\mu}_2^T\boldsymbol{w}_2 + b)^2}{2\boldsymbol{w}^T\Sigma\boldsymbol{w}}\big) \\[2mm] \gamma_{-1,-1} = \alpha_0(1-\beta)\exp\big(-\dfrac{(\boldsymbol{\mu}_1^T\boldsymbol{w}_1 + \boldsymbol{\mu}_2^T\boldsymbol{w}_2 - b)^2}{2\boldsymbol{w}^T\Sigma\boldsymbol{w}}\big) \\[2mm] \gamma_{1,-1} = (1-\alpha_1)\beta\exp\big(-\dfrac{(\boldsymbol{\mu}_1^T\boldsymbol{w}_1 - \boldsymbol{\mu}_2^T\boldsymbol{w}_2 + b)^2}{2\boldsymbol{w}^T\Sigma\boldsymbol{w}}\big) \\[2mm] \gamma_{-1,1} = (1-\alpha_0)(1-\beta)\exp\big(-\dfrac{(\boldsymbol{\mu}_1^T\boldsymbol{w}_1 - \boldsymbol{\mu}_2^T\boldsymbol{w}_2 - b)^2}{2\boldsymbol{w}^T\Sigma\boldsymbol{w}}\big) \end{cases} \tag{91}$$

Substitute eq. (89) back, we have

$$\begin{cases} \gamma_{11} = \alpha_1\beta\exp\big(-\dfrac{(m_1 + \tau m_2 + b)^2}{2(m_1 + \tau^2 m_2)}\big) \\[2mm] \gamma_{00} = \alpha_0(1-\beta)\exp\big(-\dfrac{(m_1 + \tau m_2 - b)^2}{2(m_1 + \tau^2 m_2)}\big) \\[2mm] \gamma_{10} = (1-\alpha_1)\beta\exp\big(-\dfrac{(m_1 - \tau m_2 + b)^2}{2(m_1 + \tau^2 m_2)}\big) \\[2mm] \gamma_{01} = (1-\alpha_0)(1-\beta)\exp\big(-\dfrac{(m_1 - \tau m_2 - b)^2}{2(m_1 + \tau^2 m_2)}\big) \end{cases} \tag{92}$$

Thus the classifiers are

$$\hat{y} = \boldsymbol{w}_1^T\boldsymbol{z}_1 + \boldsymbol{w}_2^T\boldsymbol{z}_2 + b \tag{93}$$
$$= \boldsymbol{\mu}_1^T\Sigma^{-1}\boldsymbol{z}_1 + \tau\boldsymbol{\mu}_2^T\Sigma_2^{-1}\boldsymbol{z}_2 + b \tag{94}$$

This completes the proof.

$\square$

## A.5 ESTIMATING FEATURE SEPARABILITY FROM TWO SETS OF DATA VARIANTS

Estimating feature separability requires two sets of empirical results with different subpopulation configurations and solves the separability $\hat{m}_1, \hat{m}_2$ as unknown variables from the following equation system, derived from Lemma 3 and Theorem 1. Denoting the accuracy as $acc_1, acc_2$, the parameters for the Bayesian classifiers as $(\tau_1, b_1), (\tau_2, b_2)$,

$$\begin{cases} F_1 = \tau_1 - \dfrac{\gamma_{1,1}(\tau_1, b_1) + \gamma_{-1,-1}(\tau_1, b_1) - \gamma_{1,-1}(\tau_1, b_1) - \gamma_{-1,1}(\tau_1, b_1)}{\gamma_{1,1}(\tau_1, b_1) + \gamma_{-1,-1}(\tau_1, b_1) + \gamma_{1,-1}(\tau_1, b_1) + \gamma_{-1,1}(\tau_1, b_1)} = 0 \\[3mm] F_2 = \gamma_{1,1}(\tau_1, b_1) + \gamma_{1,-1}(\tau_1, b_1) - \gamma_{-1,-1}(\tau_1, b_1) - \gamma_{-1,1}(\tau_1, b_1) = 0 \\[3mm] F_3 = \tau_2 - \dfrac{\gamma_{1,1}(\tau_2, b_2) + \gamma_{-1,-1}(\tau_2, b_2) - \gamma_{1,-1}(\tau_2, b_2) - \gamma_{-1,1}(\tau_2, b_2)}{\gamma_{1,1}(\tau_2, b_2) + \gamma_{-1,-1}(\tau_2, b_2) + \gamma_{1,-1}(\tau_2, b_2) + \gamma_{-1,1}(\tau_2, b_2)} = 0 \\[3mm] F_4 = \gamma_{1,1}(\tau_2, b_2) + \gamma_{1,-1}(\tau_2, b_2) - \gamma_{-1,-1}(\tau_2, b_2) - \gamma_{-1,1}(\tau_2, b_2) = 0 \\[3mm] F_5 = acc_1 - \displaystyle\sum_{\boldsymbol{a}\in\{\pm 1\}^2} \frac{1}{2}\big(1 + \mathrm{erf}\big(\frac{a_1 m_1 + \tau_1 a_2 m_2 + a_1 b_1}{\sqrt{2(m_1^2 + \tau_1^2 m_2^2)}}\big)\big) = 0 \\[3mm] F_6 = acc_2 - \displaystyle\sum_{\boldsymbol{a}\in\{\pm 1\}^2} \frac{1}{2}\big(1 + \mathrm{erf}\big(\frac{a_1 m_1 + \tau_2 a_2 m_2 + a_1 b_2}{\sqrt{2(m_1^2 + \tau_2^2 m_2^2)}}\big)\big) = 0 \end{cases}$$

where we note that the $\gamma(\cdot)$ for the two sets of experiments uses their respective subpopulation configurations.

### A.6 THREE-ATTRIBUTE EXTENSION OF THEOREM 1

Our framework is capable of handling more than two attributes, as Lemmas 1–3 establish the general structure of the Bayesian optimal classifier and group-wise accuracies. Our Theorem 1 focuses on two attributes to make the analysis and follow-up visualization clear and consistent with the structure of most benchmark datasets, which contain two primary attributes. Extending Theorem 1 to $N$ attributes introduces $2^N$ $\gamma$-functions and $N$ equations, but requires no new conceptual framework and remains computationally feasible. Moreover, the redundant attributes with multiple discrete values can be equivalently represented as several binary attributes via one-hot encoding, and thus fit directly into our formulation. For clarity, we present the three-attribute extension of Theorem 1 in Theorem 2 as follows.

**Theorem 2. (Bayesian Classifier) When $N = 3$, the Bayesian classifier is $\hat{y} = \boldsymbol{\mu}_1^T \Sigma_1^{-1} \boldsymbol{z}_1 + \tau_1 \boldsymbol{\mu}_2^T \Sigma_2^{-1} \boldsymbol{z}_2 + \tau_2 \boldsymbol{\mu}_3^T \Sigma_3^{-1} \boldsymbol{z}_3 + b$ where $\tau_1, \tau_2, b \in \mathbb{R}$ are the solutions to the equation system:**

$$\begin{cases} \tau_1 = \dfrac{\gamma_{1,1,1} + \gamma_{1,1,-1} + \gamma_{-1,-1,-1} + \gamma_{-1,-1,1} - \gamma_{1,-1,1} - \gamma_{1,-1,-1} - \gamma_{-1,1,-1} - \gamma_{-1,1,1}}{\gamma_{1,1,1} + \gamma_{1,1,-1} + \gamma_{1,-1,1} + \gamma_{1,-1,-1} + \gamma_{-1,-1,-1} + \gamma_{-1,1,-1} + \gamma_{-1,-1,1} + \gamma_{-1,1,1}} \\ \tau_2 = \dfrac{\gamma_{1,1,1} + \gamma_{-1,-1,-1} + \gamma_{-1,1,-1} + \gamma_{1,-1,1} - \gamma_{1,1,-1} - \gamma_{1,-1,-1} - \gamma_{-1,-1,1} - \gamma_{-1,1,1}}{\gamma_{1,1,1} + \gamma_{1,1,-1} + \gamma_{1,-1,1} + \gamma_{1,-1,-1} + \gamma_{-1,-1,-1} + \gamma_{-1,1,-1} + \gamma_{-1,-1,1} + \gamma_{-1,1,1}} \\ \gamma_{1,1,1} + \gamma_{1,1,-1} + \gamma_{1,-1,1} + \gamma_{1,-1,-1} = \gamma_{-1,-1,-1} + \gamma_{-1,1,-1} + \gamma_{-1,-1,1} + \gamma_{-1,1,1} \end{cases}$$

(95)

where $\gamma$s are functions of $\tau_1, \tau_2, b$.

We first write the $\kappa_{\pm}(\boldsymbol{a})$, the group sizes specified by attributes $\boldsymbol{a}$ in Lemma 1 as follows:

$$y = 1 : \begin{cases} \kappa_+(1,1,1) = \beta \alpha_+^{(2)} \alpha_+^{(3)} \\ \kappa_+(1,-1,1) = \beta(1 - \alpha_+^{(2)}) \alpha_+^{(3)} \\ \kappa_+(1,1,-1) = \beta \alpha_+^{(2)} (1 - \alpha_+^{(3)}) \\ \kappa_+(1,-1,-1) = \beta(1 - \alpha_+^{(2)})(1 - \alpha_+^{(3)}) \end{cases}$$

(96)

$$y = -1 : \begin{cases} \kappa_-(-1,-1,-1) = (1-\beta)\alpha_-^{(2)} \alpha_-^{(3)} \\ \kappa_-(-1,1,-1) = (1-\beta)(1 - \alpha_-^{(2)}) \alpha_-^{(3)} \\ \kappa_-(-1,-1,1) = (1-\beta)\alpha_-^{(2)} (1 - \alpha_-^{(3)}) \\ \kappa_-(-1,1,1) = (1-\beta)(1 - \alpha_-^{(2)})(1 - \alpha_-^{(3)}) \end{cases}$$

(97)

According to Lemma 2, we have the general form of the Bayesian optimal classifier as

$$\boldsymbol{w}_n = \eta \left\{ \sum_{\boldsymbol{a} \in \{\pm 1\}^N} \left[ \kappa_+(\boldsymbol{a}) \exp\big(-\frac{(\boldsymbol{\mu}_{\boldsymbol{a}}^T \boldsymbol{w} + b)^2}{2\boldsymbol{w}^T \Sigma \boldsymbol{w}}\big) + \kappa_-(\boldsymbol{a}) \exp\big(-\frac{(\boldsymbol{\mu}_{\boldsymbol{a}}^T \boldsymbol{w} - b)^2}{2\boldsymbol{w}^T \Sigma \boldsymbol{w}}\big) \right] a_n \right\} \Sigma_n^{-1} \boldsymbol{\mu}_n$$

(98)

and we denote the coefficients for the weight block $\mathbf{w}_n$ as $\eta_n$:

$$\eta_n = \eta \left\{ \sum_{\boldsymbol{a} \in \{\pm 1\}^N} \left[ \kappa_+(\boldsymbol{a}) \exp\big(-\frac{(\boldsymbol{\mu}_{\boldsymbol{a}}^T \boldsymbol{w} + b)^2}{2\boldsymbol{w}^T \Sigma \boldsymbol{w}}\big) + \kappa_-(\boldsymbol{a}) \exp\big(-\frac{(\boldsymbol{\mu}_{\boldsymbol{a}}^T \boldsymbol{w} - b)^2}{2\boldsymbol{w}^T \Sigma \boldsymbol{w}}\big) \right] a_n \right\}$$

(99)

Denoting the following $\gamma$ functions to simplify the notations in the Bayesian optimal classifier:

$$
\begin{cases}
\gamma_{1,1,1} = \kappa_+(1,1,1) \exp\left(-\dfrac{(\boldsymbol{\mu}_1^T \boldsymbol{w}_1 + \boldsymbol{\mu}_2^T \boldsymbol{w}_2 + \boldsymbol{\mu}_3^T \boldsymbol{w}_3 + b)^2}{2\boldsymbol{w}^T \Sigma \boldsymbol{w}}\right) \\[2mm]
\gamma_{1,-1,1} = \kappa_+(1,-1,1) \exp\left(-\dfrac{(\boldsymbol{\mu}_1^T \boldsymbol{w}_1 - \boldsymbol{\mu}_2^T \boldsymbol{w}_2 + \boldsymbol{\mu}_3^T \boldsymbol{w}_3 + b)^2}{2\boldsymbol{w}^T \Sigma \boldsymbol{w}}\right) \\[2mm]
\gamma_{1,1,-1} = \kappa_+(1,1,-1) \exp\left(-\dfrac{(\boldsymbol{\mu}_1^T \boldsymbol{w}_1 + \boldsymbol{\mu}_2^T \boldsymbol{w}_2 - \boldsymbol{\mu}_3^T \boldsymbol{w}_3 + b)^2}{2\boldsymbol{w}^T \Sigma \boldsymbol{w}}\right) \\[2mm]
\gamma_{1,-1,-1} = \kappa_+(1,-1,-1) \exp\left(-\dfrac{(\boldsymbol{\mu}_1^T \boldsymbol{w}_1 - \boldsymbol{\mu}_2^T \boldsymbol{w}_2 - \boldsymbol{\mu}_3^T \boldsymbol{w}_3 + b)^2}{2\boldsymbol{w}^T \Sigma \boldsymbol{w}}\right) \\[2mm]
\gamma_{-1,-1,-1} = \kappa_-(1,-1,-1) \exp\left(-\dfrac{(\boldsymbol{\mu}_1^T \boldsymbol{w}_1 + \boldsymbol{\mu}_2^T \boldsymbol{w}_2 + \boldsymbol{\mu}_3^T \boldsymbol{w}_3 - b)^2}{2\boldsymbol{w}^T \Sigma \boldsymbol{w}}\right) \\[2mm]
\gamma_{-1,1,-1} = \kappa_-(-1,1,-1) \exp\left(-\dfrac{(\boldsymbol{\mu}_1^T \boldsymbol{w}_1 - \boldsymbol{\mu}_2^T \boldsymbol{w}_2 + \boldsymbol{\mu}_3^T \boldsymbol{w}_3 - b)^2}{2\boldsymbol{w}^T \Sigma \boldsymbol{w}}\right) \\[2mm]
\gamma_{-1,-1,1} = \kappa_-(1,-1,1) \exp\left(-\dfrac{(\boldsymbol{\mu}_1^T \boldsymbol{w}_1 + \boldsymbol{\mu}_2^T \boldsymbol{w}_2 - \boldsymbol{\mu}_3^T \boldsymbol{w}_3 - b)^2}{2\boldsymbol{w}^T \Sigma \boldsymbol{w}}\right) \\[2mm]
\gamma_{-1,1,1} = \kappa_-(1,-1,-1) \exp\left(-\dfrac{(\boldsymbol{\mu}_1^T \boldsymbol{w}_1 - \boldsymbol{\mu}_2^T \boldsymbol{w}_2 - \boldsymbol{\mu}_3^T \boldsymbol{w}_3 - b)^2}{2\boldsymbol{w}^T \Sigma \boldsymbol{w}}\right)
\end{cases}
\tag{100}
$$

Considering the colinearity, we use $\tau_1, \tau_2$ to denote the coefficients of weight blocks thus simplify the $\gamma$ functions as follows:

$$
\begin{cases}
\gamma_{1,1,1} = \kappa_+(1,1,1) \exp\left(-\dfrac{(m_1 + \tau_1 m_2 + \tau_2 m_3 + b)^2}{2(m_1 + \tau_1^2 m_2 + \tau_2^2 m_3)}\right) \\[2mm]
\gamma_{1,1,-1} = \kappa_+(1,1,1) \exp\left(-\dfrac{(m_1 + \tau_1 m_2 - \tau_2 m_3 + b)^2}{2(m_1 + \tau_1^2 m_2 + \tau_2^2 m_3)}\right) \\[2mm]
\gamma_{1,-1,1} = \kappa_+(1,1,1) \exp\left(-\dfrac{(m_1 - \tau_1 m_2 + \tau_2 m_3 + b)^2}{2(m_1 + \tau_1^2 m_2 + \tau_2^2 m_3)}\right) \\[2mm]
\gamma_{1,-1,-1} = \kappa_+(1,1,1) \exp\left(-\dfrac{(m_1 - \tau_1 m_2 - \tau_2 m_3 + b)^2}{2(m_1 + \tau_1^2 m_2 + \tau_2^2 m_3)}\right) \\[2mm]
\gamma_{-1,-1,-1} = \kappa_-(-1,-1,-1) \exp\left(-\dfrac{(m_1 + \tau_1 m_2 + \tau_2 m_3 - b)^2}{2(m_1 + \tau_1^2 m_2 + \tau_2^2 m_3)}\right) \\[2mm]
\gamma_{-1,1,-1} = \kappa_-(-1,1,-1) \exp\left(-\dfrac{(m_1 - \tau_1 m_2 + \tau_2 m_3 - b)^2}{2(m_1 + \tau_1^2 m_2 + \tau_2^2 m_3)}\right) \\[2mm]
\gamma_{-1,-1,1} = \kappa_-(-1,-1,1) \exp\left(-\dfrac{(m_1 + \tau_1 m_2 - \tau_2 m_3 - b)^2}{2(m_1 + \tau_1^2 m_2 + \tau_2^2 m_3)}\right) \\[2mm]
\gamma_{-1,1,1} = \kappa_-(-1,1,1) \exp\left(-\dfrac{(m_1 - \tau_1 m_2 - \tau_2 m_3 - b)^2}{2(m_1 + \tau_1^2 m_2 + \tau_2^2 m_3)}\right)
\end{cases}
\tag{101}
$$

With the notation of $\gamma$ functions, we have the equation system as follows, where the first two equations represent the coefficients $\tau_1, \tau_2$, and the last one is derived from the stationary point of performance with respect to $b$.

$$
\begin{cases}
\tau_1 = \dfrac{\eta_2}{\eta_1} = \dfrac{\gamma_{1,1,1} + \gamma_{1,1,-1} + \gamma_{-1,-1,-1} + \gamma_{-1,-1,1} - \gamma_{1,-1,1} - \gamma_{1,-1,-1} - \gamma_{-1,1,-1} - \gamma_{-1,1,1}}{\gamma_{1,1,1} + \gamma_{1,1,-1} + \gamma_{1,-1,1} + \gamma_{1,-1,-1} + \gamma_{-1,-1,-1} + \gamma_{-1,1,-1} + \gamma_{-1,-1,1} + \gamma_{-1,1,1}} \\[2mm]
\tau_2 = \dfrac{\eta_3}{\eta_1} = \dfrac{\gamma_{1,1,1} + \gamma_{-1,-1,-1} + \gamma_{-1,1,-1} + \gamma_{1,-1,1} - \gamma_{1,1,-1} - \gamma_{1,-1,-1} - \gamma_{-1,-1,1} - \gamma_{-1,1,1}}{\gamma_{1,1,1} + \gamma_{1,1,-1} + \gamma_{1,-1,1} + \gamma_{1,-1,-1} + \gamma_{-1,-1,-1} + \gamma_{-1,1,-1} + \gamma_{-1,-1,1} + \gamma_{-1,1,1}} \\[2mm]
\gamma_{1,1,1} + \gamma_{1,1,-1} + \gamma_{1,-1,1} + \gamma_{1,-1,-1} = \gamma_{-1,-1,-1} + \gamma_{-1,1,-1} + \gamma_{-1,-1,1} + \gamma_{-1,1,1}
\end{cases}
\tag{102}
$$

## A.7 Estimating Feature Separability from a Single Data Variant

We have provided the estimation of feature separability in Appendix A.4, when data variants are available. This section introduces an alternative approach to estimate feature separability with the

group accuracies from only one set of data, which makes it applicable to standard datasets that lack another data variant.

Denote the group accuracy $(acc_{1,1}, acc_{1,-1}, acc_{-1,1}, acc_{-1,-1})$ obtained from the given dataset, we solve the parameters for the Bayesian classifiers as $(\tau, b)$ along with the separability for both attributes $m_1, m_2$ from the following equation system:

$$
\begin{cases}
F_1 = \tau - \dfrac{\gamma_{1,1}(\tau,b) + \gamma_{-1,-1}(\tau,b) - \gamma_{1,-1}(\tau,b) - \gamma_{-1,1}(\tau,b)}{\gamma_{1,1}(\tau,b) + \gamma_{-1,-1}(\tau,b) + \gamma_{1,-1}(\tau,b) + \gamma_{-1,1}(\tau,b)} = 0 \\[2ex]
F_2 = \gamma_{1,1}(\tau,b) + \gamma_{1,-1}(\tau,b) - \gamma_{-1,-1}(\tau,b) - \gamma_{-1,1}(\tau,b) = 0 \\[2ex]
F_3 = acc_{1,1} - \dfrac{1}{2}\Big(1 + \mathrm{erf}\,\big(\dfrac{m_1 + \tau_1 m_2 + b_1}{\sqrt{2(m_1^2 + \tau_1^2 m_2^2)}}\big)\Big) = 0 \\[2ex]
F_4 = acc_{1,-1} - \dfrac{1}{2}\Big(1 + \mathrm{erf}\,\big(\dfrac{m_1 - \tau_1 m_2 + b_1}{\sqrt{2(m_1^2 + \tau_1^2 m_2^2)}}\big)\Big) = 0 \\[2ex]
F_5 = acc_{-1,1} - \dfrac{1}{2}\Big(1 + \mathrm{erf}\,\big(\dfrac{m_1 - \tau_1 m_2 - b_1}{\sqrt{2(m_1^2 + \tau_1^2 m_2^2)}}\big)\Big) = 0 \\[2ex]
F_6 = acc_{-1,-1} - \dfrac{1}{2}\Big(1 + \mathrm{erf}\,\big(\dfrac{m_1 + \tau_1 m_2 - b_1}{\sqrt{2(m_1^2 + \tau_1^2 m_2^2)}}\big)\Big) = 0
\end{cases}
$$

## B  ADJUSTED ACCURACY VS. WGA

As our theoretical framework allows a comprehensive evaluation across subpopulation configurations, we identify that the adjusted accuracy is not always positively correlated with worst-group accuracy (WGA). Specifically, we observed that for certain cases, the changes in non-invariant feature separability can lead to opposite trends between adjusted accuracy and WGA, underscoring the advantage of the theoretical framework in terms of comprehensive evaluation.

In this section, we conduct a more comprehensive study of the relation between adjusted accuracy and WGA. Our theoretical results enable estimating the adjusted accuracy and WGA for any given correlation types $(\beta, \alpha_+, \alpha_-)$. To search all feasible distribution shifts *uniformally*, we enumerate all group ratios as $r_{1,1}, r_{1,0}, r_{0,1}, r_{0,1}$ such that $r_{1,1}, r_{1,0}, r_{0,1}, r_{0,1} \geq 0, r_{1,1} + r_{1,0} + r_{0,1} + r_{0,1} = 1$. The step size is set to 0.01, resulting in $\binom{99}{3} = 156849$ data points. The separability is set to $m_1 = m_2 = 2$. They can be transformed to $(\beta, \alpha_+, \alpha_-)$ as

$$
\beta = r_{1,1} + r_{1,0}, \quad
\begin{cases}
\alpha_+ = \dfrac{r_{1,1}}{r_{1,1} + r_{1,0}} \\[2ex]
\alpha_- = \dfrac{r_{0,0}}{r_{0,0} + r_{0,1}}
\end{cases}
$$

The adjusted accuracy vs. WGA results are visualized in fig. 6. It can be observed that the variance of adjusted accuracy decreases as WGA increases, resulting in a cone-shaped region. This global view is the root of the prevalent understanding that they are always positively correlated. Especially, from the square dots, it is concluded that when there is only **one** type of bias, the correlation is **always positive**. And spurious correlation always leads to the lowest adjusted accuracy given the WGA value. This also suggests that WGA is the most distant to second-worst-group accuracy.

However, it can also be observed from the top left - bottom right strips that as the correlation ratio shifts continuously, adjusted accuracy and WGA can become negatively correlated.

## C  DATASET INFORMATION

Our dataset usage is categorized as follows: (1) synthetic datasets with varying subpopulation configurations, used in section 3 and section 5.2; (2) standard datasets from benchmarks with a fixed subpopulation configuration of shift mixture, used in section 5.3. We provide detailed information for in section C.1 and section C.2 respectively.

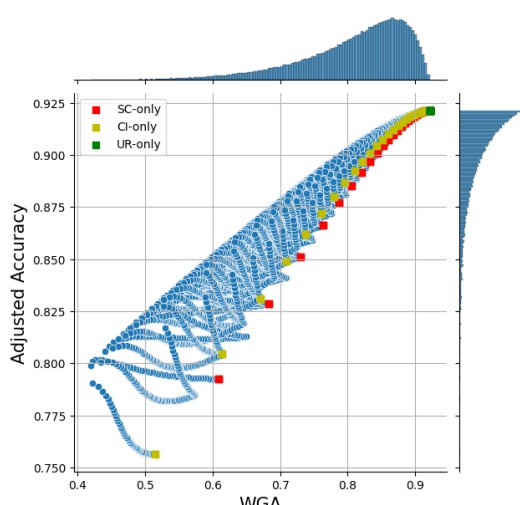

Figure 6: **The visualization of the relation between adjusted accuracy and WGA**. Each blue dot represents an individual distribution shift $r_{1,1}, r_{1,0}, r_{0,1}, r_{0,1}$. The square dots represent the scenario where only one type of bias is present.

### C.1 DATASETS WITH FLEXIBLE SUBPOPULATION CONFIGURATIONS

To represent general shifts in the binary case, we use a set of probabilities $\alpha$ and $\beta$, where $\beta = \mathbb{P}(Y = 1)$ denotes the probability of the ground truth labels (or invariant attribute), and $\alpha_+ = \mathbb{P}(A = 1|Y = 1)$ and $\alpha_- = \mathbb{P}(A = -1|Y = -1)$ represent the conditional probabilities of the other attribute $A$ given the ground truth labels $Y$.

We follow Wang & Wang (2024) to construct the MNIST-concat dataset. As the classification of digits is easy to learn, even for linear classifiers, we choose the images containing digits 3 and 5, a challenging digit pair in MNIST, as the binary attribute. Each image of digits has a shape of $28 \times 28$, and we concatenate two images vertically to form an image of $28 \times 56$. Considering the digit in the top image as the invariant attribute, the digit in the bottom image as the other attribute, MNIST-concat is thus constructed as a synthetic dataset with two binary attributes. We model the data with varying subpopulation configurations by adjusting the group sizes for the comprehensive evaluation in section 4.

To simplify the notation using $\alpha$ and $\beta$ for data with one shift, we only constrain the condition for the given shift type. Recall our definition for the shift types: (1) class imbalance: $\mathbb{P}(Y = 1) \neq 0.5$; (2) spurious correlation: $\mathbb{P}(A = Y) \neq 0.5$; (3) under-representation: $\mathbb{P}(A = 1) \neq 0.5$. Take class imbalance as an example, we require that $\mathbb{P}(Y = 1) \neq 0.5$ and simplify other conditions as $\mathbb{P}(A = 0) = \mathbb{P}(A = 1) = 0.5, \mathbb{P}(A = Y) = 0.5$. Note that we use $+1, -1$ as binary labels for convenience in theoretical derivation and follow the practice to use $+1, 0$ in experiments. In this way, data with any given shift type has four groups with two groups sharing the same larger size and the other two sharing the same smaller size, thus allowing us to simplify the notation with only one variable. We refer to the two groups with larger group sizes as *majority groups*, the other two groups as *minority groups*, and define $\zeta$ as the ratio of the two majority groups to the total dataset size. For reference, we list the corresponding values for $\alpha, \beta$ and $\zeta$ for each shift type in Table 3.

To construct datasets with single-shift types for one binary invariant and one redundant attribute, we follow the above discussion and keep the group size equal within majority or minority groups. We constructed Waterbirds-$\zeta$ and CelebA-$\zeta$ for each shift type with a varying shift degree denoted with $\zeta$. The Waterbirds dataset (Sagawa et al., 2020) contains images concatenated with the birds from Caltech-UCSD Birds-200-2011 (CUB) dataset (Wah et al., 2011) and backgrounds from the Places dataset (Zhou et al., 2017). Similarly, we concatenated the images per the needs of each group to construct the Waterbirds variants with each shift type. We used the water/land bird species and water/land backgrounds following (Sagawa et al., 2020) and created Waterbirds-$\zeta$ with $\zeta$ ranging from 0.5 to 0.999. We created Waterbirds-$\zeta$ with a total number of 20k images for each $\zeta$ in the experi-

Table 3: **The Correspondence for Majority Ratio $\zeta$ and General Configuration $\alpha_\pm, \beta$ in Single-shift Data.** We use a single variable $\zeta$ for the ratio of majority groups among all data to simplify the notation for data with single shift types. $\alpha_\pm, \beta$ are used to describe the general shift with $\beta = \mathbb{P}(Y = 1), \alpha_+ = \mathbb{P}(A = 1|Y = 1)$ and $\alpha_- = \mathbb{P}(A = -1|Y = -1)$. This table shows the correspondence between $\zeta$ and $\alpha_\pm, \beta$ to show the rationale of using $\zeta$ for the simplified notation.

| Shift Type | $r_{1,1}$ | $r_{1,0}$ | $r_{0,1}$ | $r_{0,0}$ | $\alpha_+$ | $\alpha_-$ | $\beta$ |
|---|---|---|---|---|---|---|---|
| Class Imbalance | $\frac{\zeta}{2}$ | $\frac{\zeta}{2}$ | $0.5 - \frac{\zeta}{2}$ | $0.5 - \frac{\zeta}{2}$ | 0.5 | 0.5 | $\zeta$ |
| Spurious Correlation | $\frac{\zeta}{2}$ | $0.5 - \frac{\zeta}{2}$ | $0.5 - \frac{\zeta}{2}$ | $\frac{\zeta}{2}$ | $\zeta$ | $\zeta$ | 0.5 |
| Under-Representation | $\frac{\zeta}{2}$ | $0.5 - \frac{\zeta}{2}$ | $\frac{\zeta}{2}$ | $0.5 - \frac{\zeta}{2}$ | $\zeta$ | $1 - \zeta$ | 0.5 |

Table 4: **Information and Statistics of the Standard Datasets.** We list the information of the binary attributes, along with the shift types, for the standard datasets. Additionally, we report the percentages of each subpopulation groups and the number of training/test images. Note that we use adjusted accuracy, which did not utilize the actual percentages of each groups in test data.

| Dataset | Invariant Attribute | | Redundant Attribute | | Dominant Shift | | | Trainset Statistics | | | | | Testset Statistics | | | | |
|---|---|---|---|---|---|---|---|---|---|---|---|---|---|---|---|---|---|
| | $Y = 1$ | $Y = 0$ | $A = 1$ | $A = 0$ | SC | UR | CI | $r_{1,1}$ | $r_{1,0}$ | $r_{0,1}$ | $r_{0,0}$ | # Images | $r_{1,1}$ | $r_{1,0}$ | $r_{0,1}$ | $r_{0,0}$ | # Images |
| MetaShift | Cat | Dog | Indoor | Outdoor | ✓ | | | 34.7% | 8.6% | 22.3% | 34.4% | 2,276 | 39.5% | 7.4% | 21.9% | 31.2% | 874 |
| Waterbirds | Waterbird | Landbird | Water BG | Land BG | ✓ | ✓ | ✓ | 22.0% | 1.2% | 3.8% | 73.0% | 4,795 | 11.1% | 11.1% | 38.9% | 38.9% | 5,794 |
| CelebA | Blond hair | Non-Blond hair | Male | Female | ✓ | | ✓ | 0.9% | 14.1% | 41.1% | 44.0% | 162,770 | 0.9% | 12.4% | 37.7% | 48.9% | 19,962 |

ments of shift comparison (Fig. 6), as the $\zeta$ values can be extreme to 0.999. A total number of 9,410 is used for the experiments of studying data shifts and mitigation methods in Tab. 2 for efficiency. Regarding the accuracy estimation, we deployed the test accuracy obtained from Waterbirds-0.9 and Waterbirds-0.999. We estimated the accuracy for $\zeta \geq 0.9$ in fig. 4 (b-d) to illustrate the impact on test accuracy if the data is improved. As for CelebA-$\zeta$, we chose images with blonde or black hair as the invariant attribute and male or female as the redundant attribute from the CelebA (Liu et al., 2015), and upsampled/downsampled images for specific groups to create the variants with $\zeta$ from 0.5 to 0.999 with 10k images for the training data, and $\zeta$ as 0.5 with 800 images for the test data. Similarly, we showed the trend for all CelebA-$\zeta$ and estimated the accuracy for $\zeta \geq 0.9$ in section F.

## C.2 STANDARD DATASETS

We use the datasets relating to the 3 typical shift types from the subpopulation shift benchmark (Yang et al., 2023b), including MetaShift (Liang & Zou, 2022), Waterbirds (Sagawa et al., 2020), and CelebA (Liu et al., 2015). We provide the information for these datasets in table 4. Note that as we calculate the adjusted accuracy by averaging the accuracy across four groups, the group size of the test data is only for reference and was not utilized.

## D EXPERIMENT DETAILS

Regarding the model structure, we chose the pretrained ResNet18 (He et al., 2016), ResNet50 (He et al., 2016), or Vision Transformer (ViT) (Dosovitskiy et al., 2021), each with a linear layer to map representation vectors to probabilities for two classes. This simplified structure facilitates our analysis of the representation vectors from the aspects of the classifier (linear layer). We applied a series of methods for model training, including ERM (Vapnik, 1999), IRM (Arjovsky et al., 2019), CORAL (Sun & Saenko, 2016), MMD (Li et al., 2018), MMD_b_fair (Deka & Sutherland, 2023), Group DRO (Sagawa et al., 2020), Regularization (Reg) (Wang & Wang, 2024), and Neural-collapsed-inspired method (NC) (Lu et al., 2024). The SGD optimizer was used for ResNet-based models, while the AdamW optimizer was applied for ViT-based models. The SGD optimizer used a learning rate of 5e-4 except for 1e-3 for MMD, a momentum of 0.9, and a weight decay of 5e-5. The AdamW optimizer used a learning rate of 1e-5. The batch size was set to 256 for ResNet-based training and 128 for ViT-based training. We trained all models for 50 epochs, with 5 random seeds for experiments with ERM on Waterbirds-$\zeta$ or CelebA-$\zeta$ and all methods on standard benchmark datasets, and 3 random seeds for experiments with varying methods on Waterbirds-$\zeta$/CelebA-$\zeta$. The experiments were conducted on a single GPU (NVIDIA RTX A5000 or Quadro RTX 6000) with a memory of 24 GB. The number of workers is set to 4.

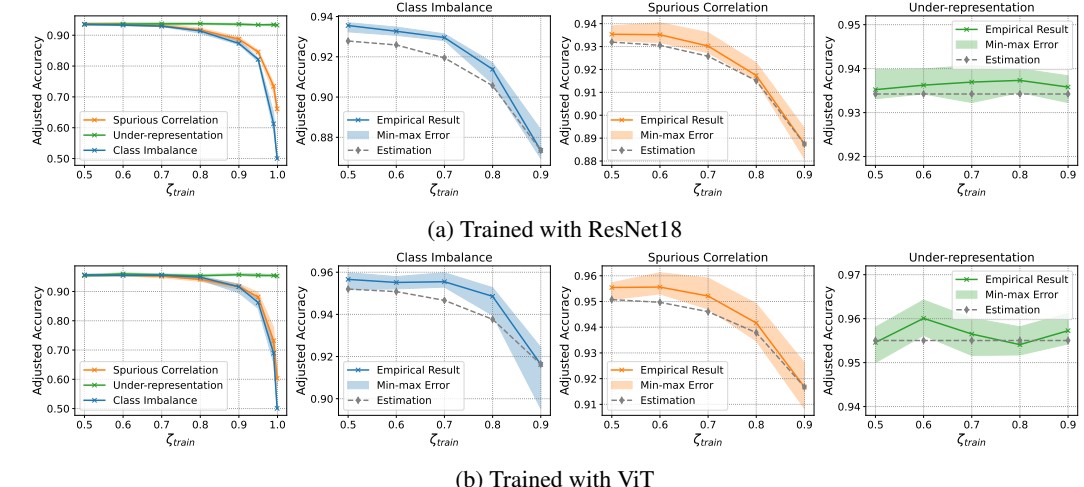

(a) Trained with ResNet18

(b) Trained with ViT

Figure 7: **Adjusted Accuracy with Varying $\zeta_{train}$ on Single-shift Waterbirds-$\zeta$.** Adjusted accuracy is shown as $\zeta_{train}$ increases on Waterbirds-$\zeta$ with a single shift type, using ERM with ResNet18 or ViT. The first subplot compares accuracy across shift types (CI > SC ≫ UR), consistent with Sec. 3.3. The remaining subplots display empirical and estimated accuracy for each shift type, highlighting their alignment.

Table 5: **Adjusted Accuracy of ResNet18 with Various Methods on Waterbirds-$\zeta$ under Single Shifts.** Average adjusted accuracy ($\pm$ standard deviation) is shown for ResNet18 with different methods on Waterbirds-$\zeta$ variants, with $\zeta_{train} = 0.9, 0.7, 0.5$ for each shift type. Results are color-coded by shift type: class imbalance , spurious correlation , and under-representation , with lighter or darker shades indicating lower/higher accuracy within all results with each shift. The best result per $\zeta$ and shift is in **bold**. Adjusted accuracy generally improves with method choice or lower $\zeta$ for CI and SC, with UR showing the least disparity.

| | Class Imbalance | | | Spurious Correlation | | | Under-Representation | | |
|---|---|---|---|---|---|---|---|---|---|
| Method | $\zeta = 0.9$ | $\zeta = 0.7$ | $\zeta = 0.5$ | $\zeta = 0.9$ | $\zeta = 0.7$ | $\zeta = 0.5$ | $\zeta = 0.9$ | $\zeta = 0.7$ | $\zeta = 0.5$ |
| ERM | 0.876±0.006 | 0.930±0.001 | 0.935±0.002 | 0.889±0.004 | 0.930±0.004 | 0.935±0.003 | 0.935±0.002 | 0.937±0.004 | 0.935±0.003 |
| IRM | 0.876±0.005 | 0.933±0.000 | **0.940**±0.002 | 0.895±0.005 | 0.935±0.004 | **0.940**±0.004 | **0.936**±0.002 | **0.940**±0.002 | **0.940**±0.002 |
| CORAL | 0.877±0.005 | 0.931±0.002 | 0.937±0.001 | 0.892±0.004 | 0.934±0.004 | 0.936±0.003 | 0.935±0.003 | 0.937±0.004 | 0.936±0.003 |
| MMD | 0.880±0.005 | 0.933±0.001 | 0.938±0.003 | 0.891±0.003 | 0.932±0.004 | 0.938±0.003 | **0.936**±0.003 | 0.938±0.003 | 0.938±0.002 |
| MMD_b_fair | 0.868±0.006 | 0.928±0.001 | 0.932±0.003 | 0.894±0.004 | 0.930±0.003 | 0.932±0.005 | 0.933±0.003 | 0.933±0.003 | 0.934±0.004 |
| Group DRO | 0.891±0.005 | 0.931±0.002 | 0.935±0.002 | 0.909±0.004 | 0.934±0.004 | 0.935±0.003 | 0.932±0.003 | 0.936±0.003 | 0.935±0.003 |
| Reg | 0.897±0.007 | 0.933±0.002 | 0.936±0.001 | 0.922±0.003 | 0.935±0.002 | 0.936±0.001 | 0.933±0.002 | 0.938±0.002 | 0.936±0.002 |
| NC | **0.914**±0.001 | **0.936**±0.002 | 0.939±0.001 | **0.924**±0.001 | **0.937**±0.002 | 0.938±0.002 | **0.936**±0.002 | **0.940**±0.001 | 0.938±0.002 |

# E    RESULTS ON SYNTHETIC DATASETS

## E.1    ACCURACY UNDER VARYING SHIFT INTENSITIES

In the main paper, we presented the experimental results obtained with ResNet50 trained on Waterbirds-$\zeta$ with the plots of the test accuracy versus shift degrees for each shift type in fig. 4 to compare the empirical and theoretically estimated results. In this section, we provide full results for two datasets (Waterbirds-$\zeta$ and CelebA-$\zeta$) and three image encoders (ResNet18, ResNet50, and ViT) in fig. 7 and fig. 8. The results demonstrate that the estimated accuracy aligns closely with the empirical accuracy trends across datasets and models.

To evaluate the performance of different methods applied to model training, we showed the results of various methods with ResNet50 trained on Waterbirds-$\zeta$ (see section 5.2 ). In addition, we provide the results of ResNet18 and ResNet50 trained on Waterbirds-$\zeta$ or CelebA-$\zeta$ in table 5, table 6, table 7, and table 8 respectively, to further validate that the performance improves with more balanced data (smaller values for $\zeta$), or while trained with methods such as Reg, Group DRO, or NC. The results also indicate that the improvement from $\zeta = 0.9$ to $\zeta = 0.7$ is sufficient and close to results with $\zeta = 0.5$, providing insights for dataset constructions. Compared to the results on Waterbirds-$\zeta$, the accuracy was higher for CelebA-$\zeta$, likely because classifying images in CelebA is less challenging than in Waterbirds.

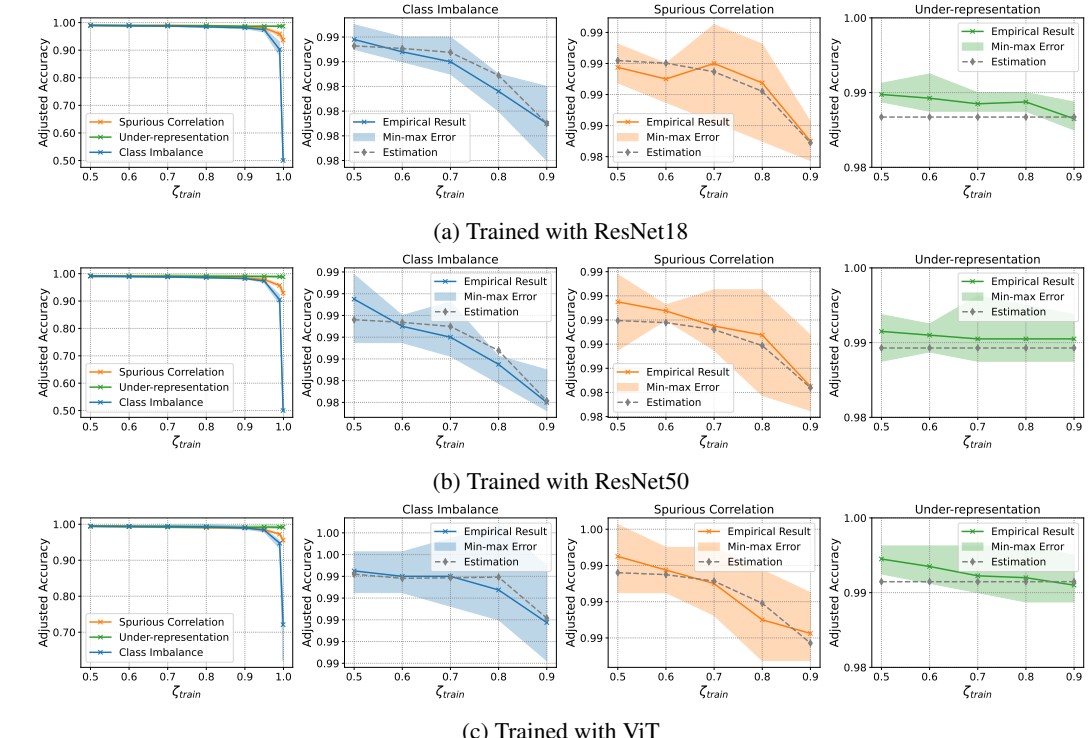

(a) Trained with ResNet18

(b) Trained with ResNet50

(c) Trained with ViT

Figure 8: **Adjusted Accuracy with Varying $\zeta_{\text{train}}$ on Single-shift CelebA-$\zeta$.** Adjusted accuracy is shown as $\zeta_{\text{train}}$ increases on CelebA-$\zeta$ with a single shift type, using ERM with ResNet18, ResNet50, or ViT. The first subplot compares accuracy across shift types (CI > SC ≫ UR), consistent with Sec. 3.3. The remaining subplots display empirical and estimated accuracy for each shift type, highlighting their alignment.

Table 6: **Adjusted Accuracy of ResNet50 with Various Methods on Waterbirds-$\zeta$ under Single Shifts.** Average adjusted accuracy ($\pm$ standard deviation) is shown for ResNet50 with different methods on Waterbirds-$\zeta$ variants, with $\zeta_{\text{train}} = 0.9, 0.7, 0.5$ for each shift type. Results are color-coded by shift type: class imbalance , spurious correlation , and under-representation , with lighter or darker shades indicating lower/higher accuracy within all results with each shift. The best result per $\zeta$ and shift is in **bold**. Adjusted accuracy generally improves with method choice or lower $\zeta$ for CI and SC, with UR showing the least disparity.

| | Class Imbalance | | | Spurious Correlation | | | Under-Representation | | |
|---|---|---|---|---|---|---|---|---|---|
| Method | $\zeta = 0.9$ | $\zeta = 0.7$ | $\zeta = 0.5$ | $\zeta = 0.9$ | $\zeta = 0.7$ | $\zeta = 0.5$ | $\zeta = 0.9$ | $\zeta = 0.7$ | $\zeta = 0.5$ |
| ERM | 0.907±0.002 | 0.948±0.002 | 0.954±0.001 | 0.925±0.002 | 0.951±0.002 | 0.955±0.003 | 0.954±0.003 | 0.955±0.002 | 0.956±0.002 |
| IRM | 0.913±0.003 | 0.952±0.001 | **0.958**±0.001 | 0.932±0.002 | 0.954±0.003 | **0.958**±0.003 | **0.958**±0.001 | **0.959**±0.003 | **0.960**±0.002 |
| CORAL | 0.909±0.002 | 0.948±0.002 | 0.954±0.001 | 0.935±0.002 | 0.954±0.002 | 0.955±0.001 | 0.955±0.002 | 0.955±0.002 | 0.955±0.002 |
| MMD | 0.913±0.002 | 0.950±0.002 | 0.955±0.001 | 0.933±0.003 | 0.946±0.002 | 0.956±0.003 | 0.956±0.002 | 0.957±0.002 | 0.957±0.002 |
| MMD_b_fair | 0.893±0.011 | 0.946±0.002 | 0.953±0.002 | 0.930±0.003 | 0.951±0.002 | 0.954±0.003 | 0.950±0.002 | 0.954±0.001 | 0.955±0.001 |
| Group DRO | 0.915±0.003 | 0.949±0.002 | 0.954±0.001 | 0.937±0.002 | 0.952±0.002 | 0.955±0.002 | 0.955±0.002 | 0.955±0.002 | 0.955±0.003 |
| Reg | 0.931±0.002 | 0.952±0.002 | 0.955±0.002 | 0.945±0.001 | 0.952±0.003 | 0.953±0.002 | 0.951±0.001 | 0.954±0.003 | 0.954±0.002 |
| NC | **0.941**±0.002 | **0.953**±0.002 | 0.957±0.003 | **0.947**±0.002 | **0.955**±0.002 | 0.957±0.002 | 0.956±0.002 | 0.957±0.003 | 0.958±0.002 |

## E.2 PERFORMANCE ESTIMATION FOR MITIGATION METHODS

Table 1 qualitatively shows that the performance of models trained with various mitigation methods approximately follows a similar trend as ERM. To further validate the generalization of our theoretical framework, we evaluate the estimation performance for top-performing mitigation methods. Similar to the setup in Figure 4, we construct the synthetic datasets Waterbirds-$\zeta$ of typical shifts and varying shift intensities. Group DRO, Reg, and NC, the top-performing methods in Table 1, are used to finetune a pretrained ResNet50 on these datasets with 3 random seeds. The separability is

Table 7: **Adjusted Accuracy of ResNet18 with Various Methods on CelebA-$\zeta$ under Single Shifts.** Average adjusted accuracy ($\pm$ standard deviation) is shown for ResNet18 with different methods on CelebA-$\zeta$ variants, with $\zeta_{\text{train}} = 0.9, 0.7, 0.5$ for each shift type. Results are color-coded by shift type: class imbalance , spurious correlation , and under-representation , with lighter or darker shades indicating lower/higher accuracy within all results with each shift. The best result per $\zeta$ and shift is in **bold**. Adjusted accuracy generally improves with method choice or lower $\zeta$ for CI and SC, with UR showing the least disparity.

| | Class Imbalance | | | Spurious Correlation | | | Under-Representation | | |
|---|---|---|---|---|---|---|---|---|---|
| Method | $\zeta = 0.9$ | $\zeta = 0.7$ | $\zeta = 0.5$ | $\zeta = 0.9$ | $\zeta = 0.7$ | $\zeta = 0.5$ | $\zeta = 0.9$ | $\zeta = 0.7$ | $\zeta = 0.5$ |
| ERM | 0.983±0.002 | 0.988±0.002 | 0.990±0.001 | 0.985±0.001 | 0.991±0.001 | 0.990±0.001 | 0.987±0.001 | 0.989±0.001 | 0.990±0.001 |
| IRM | 0.984±0.002 | 0.988±0.002 | 0.990±0.001 | 0.985±0.004 | 0.991±0.000 | 0.990±0.001 | 0.989±0.002 | 0.989±0.001 | 0.990±0.001 |
| CORAL | 0.982±0.000 | 0.987±0.002 | 0.990±0.001 | 0.985±0.002 | 0.988±0.003 | 0.990±0.001 | 0.987±0.003 | 0.990±0.001 | 0.990±0.001 |
| MMD | 0.983±0.001 | 0.986±0.003 | 0.989±0.002 | 0.985±0.001 | 0.987±0.001 | 0.989±0.002 | 0.987±0.002 | 0.989±0.001 | 0.989±0.002 |
| MMD_b_fair | 0.980±0.003 | 0.986±0.001 | 0.987±0.002 | 0.987±0.001 | 0.988±0.002 | 0.987±0.002 | 0.986±0.003 | 0.989±0.002 | 0.987±0.002 |
| Group DRO | 0.985±0.002 | 0.989±0.002 | 0.990±0.001 | 0.990±0.002 | 0.990±0.001 | 0.990±0.001 | 0.988±0.002 | 0.989±0.002 | 0.990±0.001 |
| Reg | 0.987±0.003 | **0.990**±0.002 | **0.992**±0.002 | **0.993**±0.001 | **0.993**±0.001 | **0.992**±0.002 | **0.990**±0.001 | **0.991**±0.002 | **0.992**±0.002 |
| NC | **0.990**±0.002 | **0.990**±0.001 | **0.992**±0.002 | 0.989±0.004 | 0.992±0.002 | **0.992**±0.002 | 0.988±0.002 | 0.990±0.002 | **0.992**±0.002 |

Table 8: **Adjusted Accuracy of ResNet50 with Various Methods on CelebA-$\zeta$ under Single Shifts.** Average adjusted accuracy ($\pm$ standard deviation) is shown for ResNet50 with different methods on CelebA-$\zeta$ variants, with $\zeta_{\text{train}} = 0.9, 0.7, 0.5$ for each shift type. Results are color-coded by shift type: class imbalance , spurious correlation , and under-representation , with lighter or darker shades indicating lower/higher accuracy within all results with each shift. The best result per $\zeta$ and shift is in **bold**. Adjusted accuracy generally improves with method choice or lower $\zeta$ for CI and SC, with UR showing the least disparity.

| | Class Imbalance | | | Spurious Correlation | | | Under-Representation | | |
|---|---|---|---|---|---|---|---|---|---|
| Method | $\zeta = 0.9$ | $\zeta = 0.7$ | $\zeta = 0.5$ | $\zeta = 0.9$ | $\zeta = 0.7$ | $\zeta = 0.5$ | $\zeta = 0.9$ | $\zeta = 0.7$ | $\zeta = 0.5$ |
| ERM | 0.983±0.002 | 0.988±0.003 | **0.993**±0.001 | 0.986±0.002 | 0.990±0.002 | **0.993**±0.001 | **0.992**±0.001 | **0.992**±0.003 | **0.993**±0.001 |
| IRM | 0.983±0.002 | 0.987±0.002 | 0.990±0.001 | 0.985±0.002 | 0.990±0.001 | 0.991±0.001 | 0.990±0.001 | 0.991±0.001 | 0.990±0.001 |
| CORAL | 0.983±0.001 | 0.987±0.003 | 0.991±0.002 | 0.987±0.002 | 0.991±0.001 | 0.991±0.001 | 0.990±0.002 | 0.990±0.002 | 0.991±0.002 |
| MMD | 0.981±0.003 | 0.987±0.003 | 0.992±0.002 | 0.985±0.002 | 0.989±0.002 | 0.991±0.002 | 0.988±0.002 | 0.990±0.003 | 0.992±0.002 |
| MMD_b_fair | 0.979±0.001 | 0.987±0.003 | 0.991±0.001 | 0.989±0.003 | 0.991±0.001 | 0.990±0.001 | 0.988±0.002 | 0.990±0.002 | 0.991±0.001 |
| Group DRO | 0.983±0.001 | 0.987±0.002 | 0.991±0.001 | 0.990±0.001 | **0.992**±0.002 | 0.991±0.001 | 0.990±0.002 | 0.990±0.002 | 0.991±0.001 |
| Reg | **0.989**±0.001 | **0.990**±0.002 | 0.992±0.002 | **0.991**±0.001 | 0.991±0.001 | 0.992±0.002 | 0.991±0.001 | 0.991±0.002 | 0.992±0.002 |
| NC | 0.987±0.001 | **0.990**±0.002 | 0.990±0.004 | **0.991**±0.002 | 0.990±0.003 | 0.991±0.003 | 0.989±0.002 | 0.990±0.003 | 0.990±0.004 |

estimated with two sets of empirical accuracy obtained by models trained with Group DRO/Reg/NC. The results are shown in fig. 9, fig. 10, and fig. 11 for Group DRO, Reg, and NC, respectively. The close alignment between the theoretical and empirical results underlines the effectiveness of our framework, even in more complex scenarios.

### E.3 SEPARABILITY ESTIMATION WITH TWO OR ONE SET OF DATA

We have discussed the feature separability with either two (Appendix A.5) or one (Appendix A.7) set of data. Although the synthetic datasets, Waterbirds-$\zeta$ and CelebA-$\zeta$, provide flexibility to estimate the separability with two sets of variants, we compare the theoretical performance with the separability estimated differently. Based on the ERM results of ResNet50 on Waterbirds-$\zeta$ in Figure 4, we add the theoretical performance with separability estimated from one data variant $\zeta = 0.999$, as solid lines in fig. 12. Despite not being as precise as the dashed line, which is the theoretical result estimated from two sets of data, this new theoretical performance overall aligns well with the empirical one, showing the effectiveness of the alternative approach of estimating separability.

### E.4 EFFECTS OF NUMBER OF SAMPLES FOR ESTIMATION WITH CI

To study how the shift intensities affect the test performance with each shift, we conducted experiments with a large range for $\zeta_{\text{train}}$ from 0.5 to 0.999. However, class imbalance might suffer from insufficient model training when $\zeta_{\text{train}}$ is large, where almost all samples in the training data share the same groundtruth labels. When we constructed Waterbirds-$\zeta$, we upsampled the images of water birds to the same number of land bird images, resulting in 9,410 images of birds in total, which is used in the experiments in Tab 2. However, there will be fewer than 10 samples in the two minority

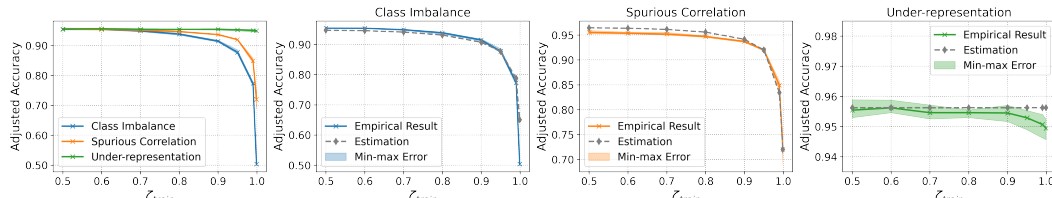

Figure 9: **Adjusted Accuracy with Varying $\zeta_{train}$ on Single-shift Waterbirds-$\zeta$ with Group DRO.** Adjusted accuracy is shown as $\zeta_{train}$ increases on Waterbirds-$\zeta$ with a single shift type, using Group DRO with ResNet50. The first subplot compares accuracy across shift types (CI > SC ≫ UR), consistent with Sec. 3.3. The remaining subplots display empirical and estimated accuracy for each shift type, highlighting their alignment.

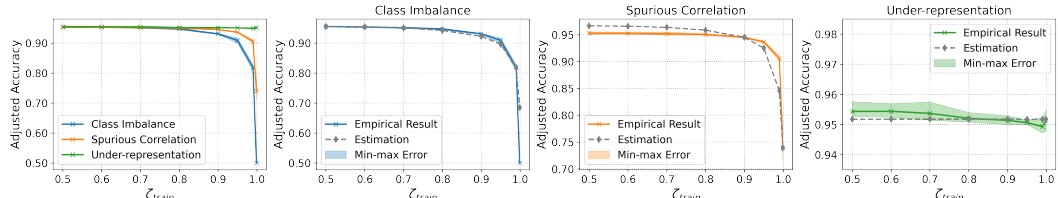

Figure 10: **Adjusted Accuracy with Varying $\zeta_{train}$ on Single-shift Waterbirds-$\zeta$ with Reg.** Adjusted accuracy is shown as $\zeta_{train}$ increases on Waterbirds-$\zeta$ with a single shift type, using Reg with ResNet50. The first subplot compares accuracy across shift types (CI > SC ≫ UR), consistent with Sec. 3.3. The remaining subplots display empirical and estimated accuracy for each shift type, highlighting their alignment.

Table 9: **Worst-Group Accuracy on Benchmark Datasets for Various Methods and Representation Networks.** We report the average adjusted accuracy for 5 random seeds along with the standard deviation, for 3 representation networks applied with various methods on 3 datasets. For each column with the given dataset and representation network, we use gradient colors to visualize the results with lighter colors for smaller values, darker colors for larger values. The largest value uses **bold** text to indicate the best-performing method. Group DRO, Reg, and NC are observed to be top-performing methods across datasets and representation networks.

| | MetaShift | | | Waterbirds | | | CelebA | | |
|---|---|---|---|---|---|---|---|---|---|
| Method | ResNet18 | ResNet50 | Transformer | ResNet18 | ResNet50 | Transformer | ResNet18 | ResNet50 | Transformer |
| ERM | 0.720±0.018 | 0.698±0.008 | 0.735±0.025 | 0.442±0.016 | 0.598±0.013 | 0.550±0.061 | 0.418±0.017 | 0.406±0.018 | 0.432±0.048 |
| IRM | 0.705±0.026 | 0.692±0.022 | 0.667±0.042 | 0.486±0.011 | 0.627±0.007 | 0.575±0.049 | 0.450±0.030 | 0.423±0.056 | 0.393±0.111 |
| CORAL | 0.726±0.018 | 0.702±0.012 | **0.788**±0.025 | 0.431±0.018 | 0.652±0.011 | 0.592±0.023 | 0.418±0.029 | 0.397±0.015 | 0.413±0.034 |
| MMD | 0.726±0.020 | 0.702±0.008 | 0.769±0.035 | 0.434±0.016 | 0.577±0.016 | 0.726±0.033 | 0.429±0.017 | 0.481±0.021 | 0.436±0.075 |
| MMD_b_fair | 0.735±0.018 | 0.702±0.016 | 0.757±0.018 | 0.494±0.010 | 0.647±0.020 | 0.659±0.038 | 0.412±0.047 | 0.452±0.022 | 0.423±0.034 |
| Group DRO | 0.748±0.021 | **0.732**±0.012 | 0.778±0.008 | 0.607±0.016 | 0.726±0.010 | 0.791±0.012 | 0.446±0.010 | 0.446±0.023 | 0.497±0.029 |
| Reg | 0.735±0.033 | 0.729±0.008 | 0.745±0.016 | 0.680±0.023 | 0.813±0.002 | 0.757±0.018 | 0.418±0.018 | 0.472±0.057 | 0.468±0.031 |
| NC | **0.754**±0.000 | 0.708±0.000 | 0.772±0.006 | **0.757**±0.003 | **0.825**±0.002 | **0.794**±0.006 | **0.543**±0.018 | **0.524**±0.039 | **0.550**±0.024 |

groups, which might introduce bias from insufficient model training. By upsampling or downsampling foreground and background images, we created three variants with 5k, 9,410, and 20k samples for CI, and we show the test performance along with the estimated accuracy in fig. 13. To emphasize the difference, we only display the results with $\zeta_{train}$ that are smaller than 0.9. Despite they show the sharpest curve as $\zeta$ varies, the comparison among the results obtained from different training set sizes indicates that a larger number of samples can result in better accuracy estimation for class imbalance.

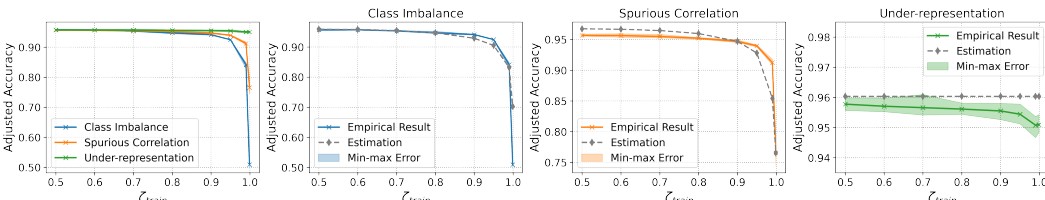

Figure 11: **Adjusted Accuracy with Varying $\zeta_{\text{train}}$ on Single-shift Waterbirds-$\zeta$ with NC.** Adjusted accuracy is shown as $\zeta_{\text{train}}$ increases on Waterbirds-$\zeta$ with a single shift type, using NC with ResNet50. The first subplot compares accuracy across shift types (CI > SC ≫ UR), consistent with Sec. 3.3. The remaining subplots display empirical and estimated accuracy for each shift type, highlighting their alignment.

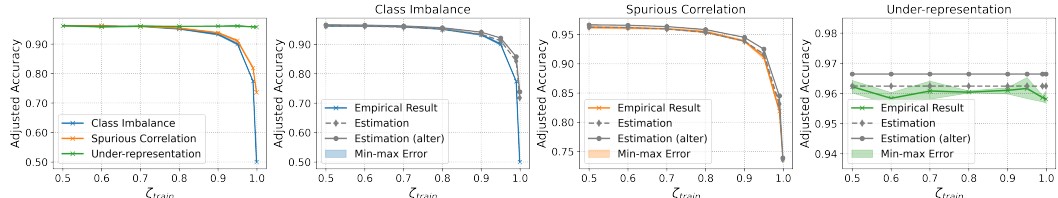

Figure 12: **Adjusted Accuracy with Varying $\zeta_{\text{train}}$ on Single-shift Waterbirds-$\zeta$.** Adjusted accuracy is shown as $\zeta_{\text{train}}$ increases on Waterbirds-$\zeta$ with a single shift type, using ERM with ResNet50. The dashed and solid lines in the right three panels show the alignment in theoretical performance estimated from two or one set of data variants, and the estimation with two sets (dashed line) is slightly better.

Table 10: **Test Accuracy on Benchmark Datasets for Various Methods and Representation Networks.** We report the average adjusted accuracy for 5 random seeds along with the standard deviation, for 3 representation networks applied with various methods on 3 datasets. For each column with the given dataset and representation network, we use gradient colors to visualize the results with lighter colors for smaller values, darker colors for larger values. The largest value uses **bold** text to indicate the best-performing method. Group DRO, Reg, and NC are observed to be top-performing methods across datasets and representation networks.

| | MetaShift | | | Waterbirds | | | CelebA | | |
|---|---|---|---|---|---|---|---|---|---|
| Method | ResNet18 | ResNet50 | Transformer | ResNet18 | ResNet50 | Transformer | ResNet18 | ResNet50 | Transformer |
| ERM | 0.892±0.004 | 0.915±0.001 | 0.880±0.005 | 0.817±0.005 | 0.850±0.003 | 0.830±0.020 | 0.948±0.001 | 0.949±0.001 | 0.956±0.001 |
| IRM | **0.892**±0.006 | 0.912±0.004 | 0.890±0.009 | 0.823±0.002 | 0.866±0.003 | 0.852±0.054 | 0.926±0.059 | **0.955**±0.001 | 0.937±0.020 |
| CORAL | 0.888±0.003 | 0.913±0.002 | 0.890±0.005 | 0.843±0.005 | 0.876±0.005 | 0.884±0.007 | 0.948±0.000 | 0.948±0.001 | **0.957**±0.001 |
| MMD | 0.889±0.005 | 0.909±0.001 | 0.875±0.008 | 0.844±0.004 | 0.880±0.002 | **0.937**±0.004 | 0.946±0.001 | 0.945±0.000 | 0.956±0.002 |
| MMD_b_fair | 0.891±0.005 | 0.911±0.004 | 0.893±0.007 | 0.838±0.006 | 0.869±0.013 | 0.907±0.004 | 0.949±0.002 | 0.950±0.002 | 0.955±0.001 |
| Group DRO | 0.888±0.004 | 0.911±0.004 | **0.909**±0.003 | 0.862±0.003 | 0.900±0.004 | 0.931±0.002 | 0.949±0.001 | 0.950±0.000 | 0.953±0.001 |
| Reg | 0.884±0.003 | 0.919±0.001 | 0.874±0.004 | 0.882±0.005 | 0.917±0.002 | 0.881±0.006 | 0.948±0.002 | 0.952±0.001 | 0.955±0.001 |
| NC | 0.890±0.001 | **0.923**±0.002 | 0.897±0.004 | **0.890**±0.001 | **0.927**±0.001 | 0.899±0.004 | **0.952**±0.001 | 0.952±0.001 | 0.953±0.001 |

# F    RESULTS ON STANDARD DATASETS

## F.1    RESULTS IN OTHER METRICS

In addition to the adjusted accuracy presented in table 2, we also report the WGA and test accuracy on the original test data as reference, in table 9 and table 10 respectively. Among all the methods across these metrics, Reg, Group DRO, and NC are top-performing with top WGA and high test accuracy.

## F.2    THEORETICAL SEPARABILITY VS. PERFORMANCE

As our main text presents the empirical separability versus the performance in Figure 5, this section additionally presents the theoretical separability estimated from the group accuracy on the stan-

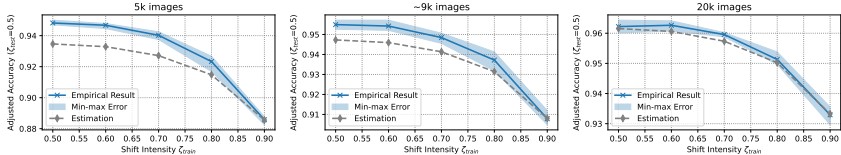

Figure 13: **CI Estimation with Varying Training Set Sizes.** As the number of samples increases, the estimation performance increases and aligns better with the empirical one.

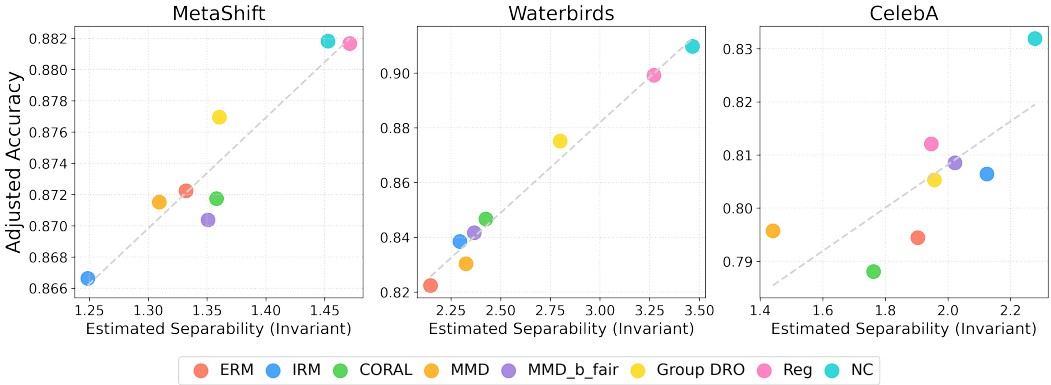

Figure 14: **Theoretically Estimated Separability (Invariant) vs. Adjust Accuracy of Various Methods.** Higher separability of the invariant attribute correlates with better performance.

dard datasets (Appendix A.5) and examines the relation with the performance in fig. 14. We show that the estimated separability for the invariant attribute also shows a positive correlation with the performance, which also validates our claim that competitive shift-mitigation methods consistently achieve higher separability of the invariant attribute.

# G  THE USE OF LARGE LANGUAGE MODELS (LLMS)

We used ChatGPT to improve the clarity of some parts in our manuscript, including grammar, phrasing, and conciseness, and the presentation of our results, including limited assistance in figure plotting. All technical contributions, derivations, and experiments are original work by the authors.

