# OpenReview forum: "Understanding Subpopulation Shifts through a Unified Lens of Separability"
_ICLR.cc/2026/Conference — Submitted to ICLR 2026_

### Official Review · Reviewer_Zda8 · 2025-10-25

**Soundness:** 3
**Presentation:** 3
**Contribution:** 2
**Rating:** 6
**Confidence:** 3

**Summary:**

This paper proposes a unified theoretical framework to analyze subpopulation shifts, positing that phenomena like spurious correlation (SC), under-representation (UR), and class imbalance (CI) can be understood through the single lens of "feature separability". The authors model features as Gaussian mixtures controlled by invariant (label) and non-invariant attributes.

**Strengths:**

The attempt to unify SC, UR, and CI under a single, simple geometric principle (separability) is elegant and provides a novel conceptual lens for the community. The core idea—that robustness relies on making the true feature easy to separate ($m_1$) and the spurious feature hard to separate ($m_2$)—is highly intuitive and explanatorily powerful.

**Weaknesses:**

Despite the paper's strengths, its claims are built on a foundation with several significant and largely unaddressed limitations. The validity of the theoretical derivations is not the primary issue; rather, it is the applicability of their underlying assumptions to real-world deep learning.

The Gaussian Feature Assumption: The entire theoretical framework, from Lemma 1 to Theorem 1, rests on the strong assumption that features $Z_n$ are Gaussian8. The authors state that this analysis "generalizes well to complex models and real-world data"9, but this claim is not sufficiently substantiated. Features extracted from deep networks like ResNets are known to be non-Gaussian. The paper provides no discussion or analysis on why a theory built on this premise holds for complex, high-dimensional, non-Gaussian features. This is a critical omission that undermines the generality of the theoretical claims.

The Linear Classifier Limitation: The theory derives an optimal linear classifier (Lemma 2, Theorem 1). The experiments explicitly adhere to this, training only a linear layer on top of a pre-trained (and presumably frozen) encoder11. This setup does not reflect the dominant paradigm of end-to-end finetuning. In a finetuning scenario, the encoder is dynamic, meaning the feature distributions ($\mu_n, \Sigma_n$) and their separability ($m_n$) are constantly changing. The paper's static framework cannot model this, severely limiting its relevance to how most SOTA models are actually trained for robustness.

The analysis and experiments are strictly confined to binary classification ($Y \in \{\pm1\}$) and binary attributes ($A \in \{\pm1\}^N$). This is a significant simplification. The paper offers no discussion on how this framework (e.g., the definition of $m_n$ and the 3-simplex visualization) would extend to multi-class classification or attributes with more than two values (e.g., 10 different types of backgrounds, not just "water" and "land").

The proposed quantitative tool (Contribution 1) relies on the availability of "data variants". This procedure is only feasible for synthetic datasets (like Waterbirds- $\zeta$) where the authors can control the subpopulation ratios. This tool is not applicable to fixed, "in-the-wild" datasets where such variants cannot be generated. This limitation on the tool's practical utility should be stated far more explicitly.

Comparison to Augmentation-Based Methods: The authors may need cite and contrast their geometric framework with algorithmic approaches based on data interpolation. E.g., Umix: Improving importance weighting for subpopulation shift via uncertainty-aware mixup, Improving out-of-distribution robustness via selective augmentation.

**Questions:**

As shown in Weaknesses

---

> ### Author Response · Authors · 2025-11-25
>
> > ### 1. The Gaussian feature assumption (global response 1).
>
> We appreciate the reviewer’s thoughtful feedback. The Gaussian modeling aims as an abstraction to ensure a tractable theoretical analysis. We show that the derived insights can generalize well in realistic setups, where the features extracted by deep encoders do not follow Gaussian distributions. We have included a detailed discussion on the practical applications of our theoretical framework in the global response 1 with additional results.
>
> > ### 2. The linear classifier limitation (global response 1).
>
> We would like to clarify that our empirical setup does not freeze the encoder. Instead, we adopt a pretrained deep encoder as the feature extractor, followed by a linear classification head, and finetune the entire network on the datasets. The results in Section 5.2 show that our theoretical estimation can match the empirical results well, even in such complex scenarios. Additionally, we also provide the estimation results in the global response 1, where the competitive mitigation methods (Group DRO and NC) are applied during training. The involvement of these methods might further amplify the gaps between theoretical and empirical setups, whereas our method remains predictive.
>
> > ### 3. Framework of binary attributes (global response 2).
>
> The setup of binary attributes serves the aim of deriving the closed-form solution (Lemma 1) considering $sign(\mathbf{w}^T \mathbf{z} + b)$. The multiple classes, involving argmax for the decision rule, make it infeasible for closed-form solutions.
> Except for the invariant attribute for classification, the attributes with multiple discrete values can fit in our framework via one-hot encoding to transform into the binary setup.
> We included a detailed discussion on the analysis of more than two attributes in the global response 2. Regarding the visualization, the 3-simplex plot provides a natural way to represent all possible combinations of the four group sizes, which are non-negative and sum to one. For $N$ attributes, there would be $2^N$ group sizes, still bounded by one condition of the sum being 1. It makes the visualization in 3D spaces infeasible, thus, we leave the analysis of more attributes within the theoretical derivation.
>
> > ### 4. Quantitative tool relying on data variants (global response 5).
>
> We agree that the acquisition of the data variants can be challenging for fixed “in-the-wild” datasets. Our quantitative tool is primarily intended for the dataset construction stage, where such variants are naturally accessible as practitioners might adjust subgroup compositions before finalizing the dataset. In this context, our method is most appropriate when there is a substantial accuracy gap between the two sets of data variants. It's more likely to be available on highly skewed data in the early stage of data construction, when the subgroups have not been calibrated. Our method provides a principled framework to evaluate the datasets during collection and ensure future model training on the dataset.
>
> Additionally, as in the global response 5 and Appendix A.7, we propose an alternative approach to estimate the separability from only one data variant, which makes it possible to apply to standard datasets. We validated that the estimated separability can yield similar performance estimation on synthetic datasets (Appendix E.2). The estimated separability can also be used to validate our model-aspect insights that strong shift mitigation methods tend to have larger separability (Appendix F.2). The supplemented results are provided in the global response with an anonymous link, and in the appendix of the revised paper.
>
> > ### 5. Comparison to augmentation-based methods.
>
> We thank the reviewer for this helpful suggestion. We agree that augmentation-based methods such as Umix and Selective Augmentation are relevant works in terms of subpopulation shift mitigation. These works can be considered as altering the training distribution by importance reweighting or data augmentation, which connects to our study from the data aspect. We have added a discussion and citations in the related work. Following our global response 5, we further added the results from these two methods in the [anonymous link](https://anonymous.4open.science/r/rebuttal-82i5a75qw3/) where the scatters are highlighted with black edge colors. The results from the newly added methods align well with the existing methods and validate our model-aspect claim that the top-performing methods tend to have larger invariant separability.

---

### Official Review · Reviewer_DAn4 · 2025-10-31

**Soundness:** 3
**Presentation:** 3
**Contribution:** 3
**Rating:** 6
**Confidence:** 3

**Summary:**

This paper presents a theoretical framework to describe different type of distributions shifts that arise in data -- including class imbalance, under-representation, and spurious correlations. This framework is built on assumptions that the data can be modeled via Gaussian functions, following advances made by prior work. After presenting how the data is modeled, a series of theoretical results build up to a main theorem given the form of the Bayesian optimal classifier as a function of the dataset parameters. From this, synthetic data can be studied to understand how the effect of data separability affects performance under different domain shifts. Commonly used benchmark datasets from the domain shift literature are used to test the applicability of this theoretical framework to "real" data. To my knowledge, the proposed framework is novel and helpful, especially in that it is general enough to cover different types of distribution shifts with one model. I have some questions/concerns on (i) how realistic this data model is and (ii) connection of the experimental results with the formal results. If the authors could please answer my questions and address my concerns, I will be happy to consider updating my score accordingly.

**Strengths:**

1. To the best of my knowledge, the proposed framework and the theoretical results are novel. This bridges an important gap in modeling and characterizing model performance under different types of distribution shifts.
2. Formal statements are clear, and to large degree well explained in the text surrounding them. Due to time constraints I was not able to fully check the proofs.
3. Empirical results connect the formal results to datasets that are curated from real images to have distribution shift. I found the results in Figure 4 and Figure 5 most compelling in this regard, as they show the trends predicted by the framework tend to hold up in practice (though the correlation in Figure 5 is a bit noisy, it does look like AA is increasing with increasing separability across all three datasets).

**Weaknesses:**

1. The connection between table 1 and 2 and the framework was not clear to me. The authors could make it clearer in the text how the results in those tables contribute to the overall aims of the paper. Specifically,
2. It was not clear to me the degree to which the modeling assumptions make sense for "real" data -- for example, modeling $\Sigma = \textrm{diag}(\Sigma_1, \ldots, \Sigma_N)$, would that imply that the covariance of features corresponding to different attributes $a_n$ have 0 covariances (and why would that make sense in practice)?


Smaller things:
1. Adjectives like "holistic", "comprehensive", etc. do not really need to be used when describing this framework, are vague, and leave the paper's contributions open to scrutiny. Better to be specific about what you mean (this framework allows us to study different types of distribution shift with one data model).
2. Missing a \ for an \in in line 360.

**Questions:**

The data modeling setup in 3.1 to me. To make sure I'm understanding, I ask the following:
1. In addition to the section on Gaussian data modeling in the appendix, could the authors give some intuition for what type of data will/will not be modeled well by the problem setup in section 3.1?
2. Is the Bayes optimal classifier guaranteed to be linear because of the assumptions made in section 3.1? This seems to be taken for granted in Lemma 1 onward, and perhaps it's a direct result of the modeling assumptions, but that could be better explained.

Other questions:
3. In section 5.2, it is stated "Thus, our framework ... is capable of estimating the expected performance to guide the dataset collection for the afterward robust model training." In figure 4 it looks like the optimal training is always around 0.5, a 50/50 split. Is there some other reason the model would prefer a 50/50 split (lots of reasonable models would...)? This experiment alone does not seem enough to really evaluate the claim in the quote, but perhaps I'm missing something.
4. I had some questions that I put in the weaknesses section as well to better contextualize why I am asking them.

---

> ### Author Response · Authors · 2025-11-25
>
> > ### 1. Tables 1 and 2 clarification (global response 4).
>
> We appreciate the reviewer for this comment and have included a discussion in global response 4. We also revised the manuscript accordingly.
>
> > ### 2. Covariance assumption on real data (global response 1).
>
> Our data is modeled as Gaussian mixtures with block-diagonal covariance, which serves as an abstraction for the tractable derivation rather than a statement about real-world feature independence. In our work, the alignment between the theoretical and empirical results supports that the abstraction is effective for modeling and generalization. Our global response 1 provides a discussion on the generalization of our theoretical analysis on realistic setups where the block-diagonal does not hold.
>
>
> > ### 3. Gaussian modeling capacity.
>
> Our model is most appropriate when the shift from training to test distribution focuses on subgroup levels, while variation within subgroups is moderate. This might not apply to covariate shift, where the example-level variation between the training and test data is significant, such as using real pictures as training while using sketch-style images as test data. In addition, we model the attributes as binary, enabling a closed-form accuracy derivation using the decision boundary $sign(\mathbf{w}^T \mathbf{z} + b)$. For multi-class classification, it's unfeasible to derive a closed-form solution with the argmax decision rule. We have added a discussion of these modeling capacities and limitations in the revised manuscript.
>
> > ### 4. Linearity in Bayesian optimal classifier.
>
> Thank you for highlighting this point. We clarify that in our setup the “Bayes-optimal classifier” refers to the Bayes-optimal linear classifier, because our analysis explicitly considers classifiers of the form $sign(\mathbf{w}^T \mathbf{z} + b)$. This setup also matches the common empirical setting, where a linear classification head is used on top of pretrained encoders. Under our Gaussian modeling for data, the linear classifier is also a natural choice. The invariant features correspond to the invariant attribute that aligns with the labels. Therefore, the decision rule reduces to a shared-covariance Gaussian discrimination problem, which produces a linear boundary, similar to classical LDA. We have added clarifying text in the revision to include the above discussion.
>
> > ### 5. Other reasons for models preferring 50/50 (global response 3).
>
> We agree that, in the typical shift setup of Figure 4, it is not surprising that the global optimum occurs near a 50/50 split. We would like to clarify both why this is expected in the experiments, and what “guide dataset collection” means within our framework.
> Figure 4 shows the typical shifts with varying intensities quantified with $\zeta_{train}$, the percentage of two groups with larger sizes. When $\zeta_{train}$ approaches 0.5, it represents a balanced training data where four groups have the same group sizes and the performance is expected to be optimal.
>
> Secondly, our contribution to data collection is a quantitative mapping from the training data distribution to the expected performance, rather than simply stating that balance is the only solution. Figure 4 and the 3-simplex visualization in Figure 2 show that as the training data distribution shifts, the performance shifts to a different extent. For example, the marginal gain from moving $\zeta_{train}=0.9 \rightarrow 0.7$ is much larger than from $0.7 \rightarrow 0.5$. This suggests that, in settings where collecting balanced data is expensive, it may be reasonable to stop at some configuration with considerable performance. We also provide a discussion in the global response 3 to further clarify the usage of our framework in terms of dataset design.
>
> > ### 6. Smaller edits.
>
> Thanks for the helpful feedback from the reviewer. We have updated our descriptions to be more specific and fixed the typo.

---

### Official Review · Reviewer_uHDz · 2025-10-31

**Soundness:** 3
**Presentation:** 2
**Contribution:** 2
**Rating:** 4
**Confidence:** 4

**Summary:**

This paper studies the problem of subpopulation shifts in classification problems, encompassing well-studied areas like spurious correlations, class imbalance, and under-representation. The main framework for the theoretical results is a binary classification setting where training data points are Gaussian parameterized by attributes. Under this model, the authors first provide a characterization of the group accuracy for the Bayes-optimal linear classifier than maximizes the overall test accuracy. This is then used as motivation for two empirical studies. First, on synthetic datasets where the subpopulation shift can be controlled, it is shown that the group performance predicted by the theory (using some estimated parameters) aligns well with the empirical performance. Secondly, on standard subpopulation shift datasets, it is shown that the empirical separability (estimated as the distance between feature clusters) correlates with the group-adjusted accuracy.

**Strengths:**

- The authors provide a characterization of the Bayes-optimal error for linear classifiers in terms of subpopulation shift parameters. The formulation applies to a wide variety of important problems like spurious correlations and class imbalance. It is not altogether surprising to me that the Bayes optimal solution can be derived in this way, but it does require a decent amount of work to go through the details and get the final result.
- From a theoretical perspective, an understanding of the Bayes optimal solution is an important starting point for comparing existing approaches and can serve as a testbed for future work on subpopulation shifts
- In the setting of datasets with "flexible subpopulation configurations" and two attributes, the parameters needed to compute the Bayes optimal error (specifically, the feature separability parameters) can be estimated from two sets of data with different subpopulation configurations. Assuming a well-trained linear model (that is close to the Bayes solution), this allows for prediction of the group errors, which can be useful with the important task of encoder selection

**Weaknesses:**

I outline my main concerns below:
1. Feasibility of using the theory for estimating the expected performance on real problems: From my understanding, this requires first estimating the feature separability for each attribute (side question: this only works for 2 attributes?) based on two sets of data with different subpopulation configurations. Outside of the  synthetic settings considered in the paper, this seems like quite a strong assumption, since if such datasets were available, we could probably do better by just using this extra data to improve performance in the first place. I would appreciate some more discussion about the feasibility of this assumption
2. The main point of Figure 1 seems to be that a well-trained linear classifier is close to Bayes-optimal. This does not seem to be surprising, given the simplicity of the synthetic datasets used in these experiments. In more realistic settings, the empirical and Bayes-optimal solution may differ more significantly, so the theoretical results might have less usefulness overall
3. Writing concerns: I found the writing in several parts to be a bit unclear/vague, especially regarding the terms "data aspect" and "model aspect", which show up in many parts of the paper. For example, "our aim to analyze from the model aspect" (pg. 8). This terminology seems imprecise and I wasn't able to fully understand what is meant by these terms

**Questions:**

- It seemed to me like Tables 1 and 2 have little do with the main argument of the paper, since the theory would only capture the "ERM" method that does not actively try to mitigate shifts. How do these results connect with the insights from the theory?
- I did not fully understand the claim that this framework can aid as "a practical tool for dataset design". Is the idea that you could estimate the group accuracy from a dataset in order to determine what groups to collect more data for?
- What data is used to estimate the empirical separability in Section 5.3?

Small comments/fixes
- Section 3.2 - "randomly parameterized". I assume this result is for a *fixed* w,b, and not a random (i.e., stochastic) choice?
- How is the empirical classifier trained in the experiments (this is in the Appendix, but I think it deserves to be mentioned in the main paper for clarity)

**Details Of Ethics Concerns:**

I have no ethics concerns.

---

> ### Author Response · Authors · 2025-11-25
>
> > ### 1. Usage of two sets of data to improve performance (global response 2).
>
> We would like to clarify that the separability estimation does not require two distinct sets of data variants. Instead, the data variant could be obtained by altering groups in the original data, therefore, the "extra" data could be too limited to improve the performance directly.
> In addition, our estimation method performs best when there is a substantial accuracy gap between the two sets of data variants. It's more likely to be available on highly skewed data, such as when the majority groups take account of 0.9 and 0.999 in the overall data. The supplemented data may still be insufficient to raise the overall performance to an ideal level.
> Our framework also applies to multiple attributes and please refer to global response 2 for a detailed discussion.
>
>
> > ### 2. Gap between empirical and Bayesian optimal classifier in a realistic setting (global response 1).
>
> Comparing classifiers or decision boundaries is meaningful only in the synthetic setting where the data distribution is modeled explicitly with means and covariances. Our Lemma 2 shows that the theoretical weights are colinear with the Gaussian characteristics, while we do not expect such Gaussian distributions for real images. In addition, the estimation of the classifier only serves as an intermediate conclusion, while our goal is to estimate the performance. Therefore, we focus on the comparison between theoretical and empirical performance on realistic images. We have included a discussion regarding the generalization of our framework in the empirical setting in global response 1, which also shows the effectiveness of our framework on non-Gaussian real data.
>
> >  ### 3. Clarification on data and model aspects.
>
> The data-aspect analysis in Sections 4.1 and 5.2 refers to analyzing the influence of subpopulation ratios which are inherent just in data (when $\alpha_+, \alpha_-, \beta$ vary), while the model-aspect analysis in Sections 4.2 and 5.3 refers to analyzing the role of empirical feature separability, which can vary as the encoder or the training strategy varies. We included the information in these sections to ensure clarity.
>
> >  ### 4. Connection of tables 1 and 2 to the main argument (global response 4).
>
> Thank you for raising this point. Tables 1 and 2 connect to our argument from data and model aspects, respectively. The full discussion is provided in global response 4 and we have revised the descriptions in the manuscript accordingly.
>
> >  ### 5. Clarification on "a practical tool for dataset design" (global response 3).
>
> Our framework is designed as a quantitative tool for dataset design by enabling the estimation of expected performance for new configurations. This allows an informed and efficient decision on whether further efforts are needed for the dataset collection. We have provided the clarification and additional results in global response 3 and refer the reviewer to that section for details.
>
> >  ### 6. Empirical separability estimation in Section 5.3 (global response 5).
>
> We only applied separability estimation in Section 5.2 when the data variants are available. In contrast, Section 5.3 studies the role of empirical separability on datasets with fixed group sizes. The empirical separability is evaluated with the distance between feature clusters, as described in Section 4.2. In addition to our validation in the original paper, we introduce an alternative approach to estimate the separability from one set of data that is applicable to standard datasets without other data variants, detailed in the global response 5 and Appendix A.7 in the revised paper. With the solved feature separability, we further validate its relation with the performance, which also indicates that competitive methods are likely to have larger separability.
>
> > ### 7. Clarification on "randomly parameterized".
>
> We agree with the reviewer's understanding. This sentence refers to that we analyze the general performance for a fixed $\mathbf{w}, b$. We have clarified the descriptions in the revised paper.
>
> > ### 8. Training details of the empirical classifier.
>
> We use a pretrained vision encoder (ResNet18, ResNet50, or ViT) along with a one-layer classification head, and we finetune the whole network on these datasets with ERM or mitigation methods. We have added a brief discussion in the main text.

---

### Official Review · Reviewer_tAiR · 2025-11-03

**Soundness:** 3
**Presentation:** 3
**Contribution:** 2
**Rating:** 4
**Confidence:** 4

**Summary:**

The authors theoretically study the problem of learning under subpopulation shift (spurious correlation, under-representation, and class imbalance). They assume a Gaussian features model, and then derive a closed form solution for the overall and per-group accuracies. Of crucial importance is the feature separability of the invariant and spurious features. The authors show that their theory can be used to estimate performance on real-world data by inferring the parameters of their model.

**Strengths:**

1. The paper presents a solid connection between theory and previously observed empirical phenomenon.
2. The authors are able to derived closed-form solutions, and the insights from the paper are compelling.

**Weaknesses:**

1. The novelty of this work over Wang and Wang (2024) is rather limited. In particular, the modelling assumptions and notation are almost identical, with I believe the test-set accuracy from the prior work being the same as the adjusted accuracy from this work. The prior work also notes the importance of feature separation (denoted there as $m_{inv}$ and $m_{spur}$). Though I understand that this work explores the subpopulation shift setting more generally, I am not convinced that the contribution over prior work is significant.

2. The assumptions made in the paper are rather strong. In addition to the mixture of Gaussians assumption, it is further assumed that the covariance has a block-diagonal structure, and that the single invariant attribute is perfectly informative (equals the feature) which rules out any label noise. Further, the most salient analyses are presented for the case of only two features. All of these assumptions reduce the practical applicability of the theory.

3. The connection between adjusted accuracy and WGA is interesting. Can the authors derive a theory showing the relation between these two, e.g. a closed form expression relating the two metrics?

4. In all of the empirical results, under-representation does not seem to affect the adjusted accuracy.

5. The authors should provide more detail in the main paper on how the estimated performance in Figure 4 is calculated. In particular, assuming $\mu$ and $\Sigma$ are estimated from data, are the embeddings actually mixtures of Gaussians? Further, the authors state that they solve Theorem 1 by nonlinear optimization; can the authors comment on the existence and uniqueness of the solution?

**Questions:**

Please address the weaknesses above.

---

> ### Author Response · Authors · 2025-11-25
> **Response (1/2)**
>
> > ### 1. Theoretical contribution compared to prior work [1].
>
> We appreciate the opportunity to clarify how our theoretical contributions differ from the prior work [1]. While both works adopt Gaussian feature modeling for analytical tractability, the scope and contribution differ fundamentally. Our work encompasses the findings in [1] and extends the scope largely, as discussed in our related work and summarized below.
>
> (1) Subpopulation modeling: general subpopulation shifts vs. a single spurious-correlation modeling. We appreciate the reviewer recognizing that our work explores the subpopulation shift setting more generally, and we would like to elaborate more on the modeling and theoretical contributions. The prior work studies only strictly controlled spurious correlation, assuming symmetric group ratios that two majority/minority groups share the same size, and models the data by a single parameter $\zeta$. In contrast, our theoretical analysis greatly extends the modeling with $\mathbf{\alpha}, \beta$, which allows any general subpopulation modeling, encompasses the modeling, and extends the scope largely. Considering the scenario of two attributes, our modeling builds a 3D space ($\alpha_+, \alpha_-, \beta$) to characterize the subpopulations, while [1]'s modeling collapses to a specific case under our modeling by assuming $\zeta=\alpha_+=\alpha_-, \beta=0.5$, which can be considered as a 1-D slice of our general modeling. To summarize, our work greatly extends the modeling from strict spurious correlation to general subpopulation shifts, which allows us to analyze not only spurious correlation but also other subpopulation shifts, such as under-representation and class imbalance.
>
> (2) Theoretical contribution: generalized linear classifier and per-group accuracy. We also would like to note that the derivation of our generalized data modeling is non-trivial compared to the prior work. The assumption of data symmetry in strict spurious correlation can greatly simplify the linear model and the derivation of accuracy by removing the bias term ($b=0$). While under our generalized modeling, we focus on a more general linear classifier with a bias term to capture the nuances of diverse data distributions. Generalizing the accuracy derivation and the solution to the general Bayesian optimal classifier is substantial and matches the realistic settings more. Regarding the objective in the theoretical analysis, the prior work focuses on the testing accuracy, which as noted by the reviewer, matches the concept of the adjusted accuracy in our work. However, our main theoretical analysis focuses on the per-group accuracy, which is more flexible to characterize a series of metrics, including worst-group accuracy, adjusted accuracy, or test accuracy on any test data, which we believe can be more practical. As some of the experiment results are reported with adjusted accuracy, we also include the worst-group accuracy results and the accuracy on the original test data in the appendix.
>
> (3) New insights under the generalized framework: Beyond generalizing the data modeling and the Bayes classifier, our framework leads to qualitatively new insights that cannot be expressed under the symmetric assumptions of [1]. Building on the generalized modeling and derivation, our framework captures how different subpopulations behave differently under shifts, enabling analysis and diagnosis that was impossible in [1]. Besides, our framework reveals previously unknown interactions among spurious correlation, under-representation, and class imbalance. Our theoretical results, along with the 3-simplex visualization, present the quantitative evaluation of how the performance varies when data exhibits a mixture of shifts.
>
> [1] Wang, Yipei, and Xiaoqian Wang. "On the effect of key factors in spurious correlation: A theoretical perspective." International Conference on Artificial Intelligence and Statistics. PMLR, 2024.
>
> > ### 2. Assumptions and practical applicability (global response 1 & 2).
>
> Please refer to global response 1 regarding the theoretical assumptions and empirical generalization, and global response 2 for the analysis of more than two attributes. Below we provide additional clarifications on the modeling of invariant attribute and label noise: Although the invariant attribute $a_1$ is aligned with the label, our model focuses on the features and does not assume that the data directly reveals the invariant attribute. The features are modeled as Gaussians distributed around $+\mathbf{\mu_1}$ or $-\mathbf{\mu_1}$ depending on the value of $a_1$, which therefore incorporates variability in our data modeling.
> This preserves analytical tractability while still allowing for realistic stochasticity.

---

> ### Author Response · Authors · 2025-11-25
> **Response (2/2)**
>
> > ### 3. Connection between adjusted accuracy and worst-group accuracy.
>
> We appreciate the suggestions from the reviewer. As the solution to the Bayesian optimal classifier requires the numerical solution to the equation system, the closed-form relation between the adjusted accuracy and worst-group accuracy is challenging to provide. However, we included a discussion on their relations in Appendix B, presenting an interesting phenomenon that adjusted accuracy and worst-group accuracy sometimes show a negative correlation in certain continuous shifts. We have mentioned the discussion briefly in the main text.
>
> > ### 4. The effects of under-representation on adjusted accuracy.
>
> The under-representation we studied excludes other shifts by keeping the invariant attribute balanced and only varying the distribution of the non-invariant attribute.
> The results of under-representation are expected both theoretically and empirically. We found that when the shift in the training data consists only of under-representation, the classifier can learn very well without regard to the distribution of the other attribute. It showcases the important role of studying mixed shifts, and the 3-simplex visualization further shows that the under-representation itself might not harm the performance, but could lead to a drop while mixing with other shifts.
>
> > ### 5. Details on performance estimation.
>
> We clarify that our estimation does not assume the empirical features are Gaussian mixtures, thus not requiring direct estimation of their means and covariances.
> Our Lemma 3 suggested that the Bayesian optimal accuracy can be derived with the modeling of feature separability $m = \mathbf{\mu}^T\Sigma^{-1}\mathbf{\mu}$ without the explicit usage of $\mathbf{\mu}$ or $\Sigma$. While the feature separability can be considered as a constant evaluating how easy an attribute can be classified, we can solve them numerically by building an equation system, as detailed in Appendix A.5. Denoting the empirical accuracy as $acc_1, acc_2$, the parameters for the Bayesian classifiers as $(\tau_1, b_1), (\tau_2, b_2)$, we can solve the six parameters $\hat{m}_1, \hat{m}_2, \tau_1, b_1, \tau_2, b_2$ from the following equation system:
>
> \\begin{align}
>     \\left\\{
>     \\begin{aligned}
>     F_1 &= \\tau_1 - \\frac{\\gamma_{1,1}(\\tau_1,b_1) + \\gamma_{-1,-1}(\\tau_1,b_1) - \\gamma_{1,-1}(\\tau_1,b_1) - \\gamma_{-1,1}(\\tau,b)}{\\gamma_{1,1}(\\tau_1,b_1) + \\gamma_{-1,-1}(\\tau_1,b_1) + \\gamma_{1,-1}(\\tau_1,b_1) + \\gamma_{-1,1}(\\tau_1,b_1)} = 0 \\\\
>     F_2 &= \\gamma_{1,1}(\\tau_1,b_1) + \\gamma_{1,-1}(\\tau_1,b_1) - \\gamma_{-1,-1}(\\tau_1,b_1) - \\gamma_{-1,1}(\\tau_1,b_1) = 0\\\\
>     F_3 &= \\tau_2 - \\frac{\\gamma_{1,1}(\\tau_2,b_2) + \\gamma_{-1,-1}(\\tau_2,b_2) - \\gamma_{1,-1}(\\tau_2,b_2) - \\gamma_{-1,1}(\\tau_2,b_2)}{\\gamma_{1,1}(\\tau_2,b_2) + \\gamma_{-1,-1}(\\tau_2,b_2) + \\gamma_{1,-1}(\\tau_2,b_2) + \\gamma_{-1,1}(\\tau_2,b_2)} = 0\\\\
>     F_4 &= \\gamma_{1,1}(\\tau_2,b_2) + \\gamma_{1,-1}(\\tau_2,b_2) - \\gamma_{-1,-1}(\\tau_2,b_2) - \\gamma_{-1,1}(\\tau_2,b_2) = 0\\\\
>     F_5 &= acc_1 - \\sum_{\\mathbf{a}\\in\\{\\pm1\\}^2} \\frac{1}{2}\\Big(1+\\operatorname{erf}\\big(\\frac{a_1 m_1 + \\tau_1 a_2 m_2 + a_1 b_1}{\\sqrt{2(m_1 ^2 + \\tau_1^2 m_2^2)}}\\big)\\Big) = 0\\\\
>     F_6 &= acc_2 - \\sum_{\\mathbf{a}\\in\\{\\pm1\\}^2} \\frac{1}{2}\\Big(1+\\operatorname{erf}\\big(\\frac{a_1 m_1 + \\tau_2 a_2 m_2 + a_1 b_2}{\\sqrt{2(m_1 ^2 + \\tau_2^2 m_2^2)}}\\big)\\Big) = 0\\\\
>     \\end{aligned}
>     \\right.
> \\end{align}
>
> The existence and uniqueness of the solution in Theorem 1: The equation system is obtained by determining the Bayesian optimal classifier with the stationary points to maximize the performance on the training distribution. Therefore, given a training distribution, there exists at least one pair $(\tau^\*, b^\*)$ as the maximizer of the performance. Considering the nonlinear optimization for the solution, the uniqueness might not be guaranteed. Since our analysis focuses on the performance rather than the specific numerical values of $(\tau, b)$, any stationary solution achieving Bayes-optimal accuracy suffices for the theoretical development.

---

### Author Response · Authors · 2025-11-25
**Global response (3/3)**

> ### 4. Tables 1 and 2 clarification.

Tables 1 and 2 connect to our argument from different aspects. Table 1 and Section 5.2 study the data aspect, that how the performance varies when the subpopulations in the training data change. Table 1 shows the results obtained on datasets with typical shifts and varying shift intensities, which provides prospects to examine the performance trend across methods and shows the framework's potential in terms of characterizing the performance changes. The theoretical estimation is primarily conducted on ERM results, but Table 1 shows a similar trend across methods. In addition, Table 1 also shows the framework for guiding the usage of the mitigation methods, which when the performance is close to the best achievable performance, the improvement from the mitigation methods becomes marginal. We also supplemented the estimation results for Group DRO and NC in the [anonymous link](https://anonymous.4open.science/r/rebuttal-82i5a75qw3/) and Appendix E, as discussed in the global response 1.

Table 2 and Section 5.3 validate the role of empirical separability of the invariant features by analyzing the performance on a fixed setup of training data. Following the observation from Figure 5 that higher empirical invariant separability helps a better performance, Table 2 shows the methods with higher separability achieved top performance across datasets and feature extractors. As the performance disparity could be limited for line plots, we primarily show the results on ResNet50 in Figure 5. We also updated the display of Table 2 in the revised paper by using the value of empirical invariant separability to color the brightness. The new Table 2 better illustrates the role of empirical invariant separability in the top performance.

> ### 5. Separability estimation relying on two sets of data variants.

To address reviewer concerns about evaluating separability on standard datasets that lack another variant, we introduce an alternative method for estimating feature separability directly from group accuracies from one dataset setup. This method is detailed in Appendix A.7, with corresponding analyses in Appendices E.3 and F.2. While this estimation is less precise than using results obtained from two sets of dataset variants, it still produces (1) accurate performance predictions and (2) yields meaningful insights for model behavior.

(1) We further compare empirical results to theoretical estimates obtained using separability derived from one (alternative) versus two dataset variants (discussed and utilized in the original paper), reported as “Estimation” and “Estimation (alter).” These comparisons (see Global response 5/Alter_Figure_4.jpg in the [anonymous link](https://anonymous.4open.science/r/rebuttal-82i5a75qw3/) and Appendix E.3) show that the alternative approach remains reliable.

(2) Using this alternative separability estimation, we are able to further validate our Section 5.3 claim: competitive shift-mitigation methods consistently achieve higher separability of the invariant attribute, in line with our theoretical analysis in Section 4.2. While our original paper presents the empirical separability versus the performance, we show that the theoretically estimated separability shows a similar positive correlation with the performance. The corresponding figure (Alter_Figure_5.jpg) is available in the [anonymous link](https://anonymous.4open.science/r/rebuttal-82i5a75qw3/) and included in Appendix F.2.

---

### Author Response · Authors · 2025-11-25
**Global response (2/3)**

> ### 1. Theoretical assumptions and the practical application.

We adopted simplified data modeling with Gaussian mixtures to enable a tractable characterization of the Bayes-optimal classifier. Gaussian assumptions have been used in prior work on subpopulation shifts or out-of-distribution [1,2,3], and these studies show that derivations can generalize well beyond their idealized settings. Consistently, our empirical results (Figures 2c, 4, 7, and 8) closely match the theoretical predictions. The practical setup might violate the assumptions, including that features extracted by deep encoders deviate from Gaussian and covariances are not block-diagonal. **This suggests that our framework captures the essentials in analyzing subpopulation shift rather than relying on the exact theoretical assumptions.**

To further assess practical applicability, we added new evaluations for theoretical performance estimation on models trained with top-performing mitigation methods (Group DRO, Reg, and NC). Compared to ERM, these mitigation methods further amplify the theoretical and empirical gap, underscoring that our theory remains effective under realistic settings. The results follow the same protocol as Figure 4 and have been included in Appendix E (Figures 9, 10, and 11). The figures are also provided in the [anonymous link](https://anonymous.4open.science/r/rebuttal-82i5a75qw3/). **These results show that the theoretical alignment persists even in more complex scenarios.**

[1] Nagarajan, Vaishnavh, Anders Andreassen, and Behnam Neyshabur. "Understanding the failure modes of out-of-distribution generalization." arXiv preprint arXiv:2010.15775 (2020).

[2] Liu, Hong, et al. "Self-supervised learning is more robust to dataset imbalance." arXiv preprint arXiv:2110.05025 (2021).

[3] Ming, Yifei, Hang Yin, and Yixuan Li. "On the impact of spurious correlation for out-of-distribution detection." Proceedings of the AAAI conference on artificial intelligence. Vol. 36. No. 9. 2022.


> ### 2. Analysis of more than two attributes.

**Our framework is capable of handling more than two attributes, as Lemmas 1–3 establish the general structure of the Bayesian optimal classifier and group-wise accuracies.** Our Theorem 1 focuses on two attributes to make the analysis and follow-up visualization clear and consistent with the structure of most benchmark datasets, which contain two primary attributes. Extending Theorem 1 to $N$ attributes introduces $2^N$ $\gamma$-functions and $N$ equations, but requires no new conceptual framework and remains computationally feasible. Moreover, the redundant attributes with multiple discrete values can be equivalently represented as several binary attributes via one-hot encoding, and thus fit directly into our formulation. For clarity, we present the three-attribute extension of Theorem 1 in Theorem 2 as follows. We have included its full proof in Appendix A.6 and also emphasized the utility of multiple attributes in the revised paper.

Theorem 2. (Bayesian Classifier) When $N = 3$, the Bayesian classifier is $\\hat{y} = \\mathbf{\\mu}^T_1\\Sigma^{-1}_1\\mathbf{z}_1 + \\tau_1\\mathbf{\\mu}^T_2\\Sigma^{-1}_2\\mathbf{z}_2  + \\tau_2\\mathbf{\\mu}^T_3\\Sigma^{-1}_3\\mathbf{z}_3 + b$ where $\\tau_1, \\tau_2, b\\in\\mathbb{R}$ are the solutions to the equation system:

\\begin{align}
    \\left\\{
    \\begin{aligned}
    &\\tau_1 = \\frac{\\gamma_{1,1,1} + \\gamma_{1,1,-1} + \\gamma_{-1,-1,-1} + \\gamma_{-1,-1,1} - \\gamma_{1,-1,1} - \\gamma_{1,-1,-1} - \\gamma_{-1,1,-1} - \\gamma_{-1,1,1}}{\\gamma_{1,1,1} + \\gamma_{1,1,-1} + \\gamma_{1,-1,1} + \\gamma_{1,-1,-1} + \\gamma_{-1,-1,-1} + \\gamma_{-1,1,-1} + \\gamma_{-1,-1,1} + \\gamma_{-1,1,1}} \\\\
    &\\tau_2 = \\frac{\\gamma_{1,1,1} + \\gamma_{-1,-1,-1} + \\gamma_{-1,1,-1} + \\gamma_{1,-1,1} - \\gamma_{1,1,-1} - \\gamma_{1,-1,-1} - \\gamma_{-1,-1,1} - \\gamma_{-1,1,1}}{\\gamma_{1,1,1} + \\gamma_{1,1,-1} + \\gamma_{1,-1,1} + \\gamma_{1,-1,-1} + \\gamma_{-1,-1,-1} + \\gamma_{-1,1,-1} + \\gamma_{-1,-1,1} + \\gamma_{-1,1,1}} \\\\
    &\\gamma_{1,1,1} + \\gamma_{1,1,-1} + \\gamma_{1,-1,1} + \\gamma_{1,-1,-1} = \\gamma_{-1,-1,-1} + \\gamma_{-1,1,-1} + \\gamma_{-1,-1,1} + \\gamma_{-1,1,1}
    \\end{aligned}
    \\right.
\\end{align}
where $\\gamma$s are functions of $\\tau_1,\\tau_2,b$.

*continued in the next reply*

---

### Author Response · Authors · 2025-11-30
**Global response (1/3)**

We sincerely thank all reviewers for their time and effort in providing insightful and constructive feedback for our work. We appreciate the positive remarks from reviewers, including (tAiR) "solid connection between theory and previously observed empirical phenomenon", "insights are compelling", (uHDz) "the formulation applies to a wide variety of important problems", "an important starting point", "useful with the important task of encoder selection", (DAn4) "bridges an important gap", "empirical results connect the formal results", (Zda8) "a novel conceptual lens", and "highly intuitive and explanatorily powerful". We are also grateful that reviewers provided positive ratings on the soundness, presentations, and contributions of our work.

We summarize our contribution, the rebuttal, and the global responses to common concerns below. A revised version of the paper, with changes highlighted in blue, has been uploaded accordingly. We stored our supplemented results for the rebuttal in the [anonymous link](https://anonymous.4open.science/r/rebuttal-82i5a75qw3/), detailed in the following global responses.

### **Contribution Summary**

We introduce a unified theoretical framework for modeling random subpopulation shifts that encompasses spurious correlations, under-representation, and class imbalance. Our contributions span both data and model perspectives. From the data-side contribution, our framework provides a theoretical characterization of model performance under arbitrary subpopulation shifts in the training data (Section 4.1). This enables systematic analysis of how performance varies as the training distribution changes. Experiments on controlled datasets (Section 5.2) show that our theoretical estimates closely match empirical results with ERM in the complex setting of realistic images and deep encoders.
On the other hand, from the model side, our analysis highlights the separability of the invariant attribute as a key factor governing performance under subpopulation shifts (Section 4.2). We empirically validate this insight on benchmark datasets by examining various methods of shift mitigation (Section 5.3), showing strong alignment between empirical separability and observed performance. **To conclude, we contribute a framework that allows performance examination under various subpopulation shifts and provides insights from both data and model aspects, which has potential in aiding future dataset construction and shift mitigation methods.**

### **Rebuttal Remark**

We appreciate the time and effort of reviewers dedicated to providing feedback for our paper. We summarize and address five common concerns mentioned across reviewers in the Global Response. All reviewers appreciate the technical contribution of our theoretical framework while expressing concerns about the practical usage under theoretical assumptions. As answered in Global Response 1, we clarified that the assumptions are introduced for tractable derivation and our paper shows that the resulting insights generalize well in realistic experimental settings. We also provided additional results under even more complex cases. Reviewers tAiR, uHDz, and Zda8 are concerned about the framework's applicability to more than two attributes. Global Response 2 explained that our framework already encompasses the general cases and demonstrated the capacity by presenting the three-attribute Bayesian classifier. Reviewers uHDz and DAn4 shared the confusion in the broader impact of dataset design, as well as the clarification on Tables 1/2. These points were addressed in Global Responses 3 and 4, respectively, where we also included a supplementary example. Reviewers uHDz and Zda8 noted the reliance of our theoretical estimation on two data variants. Global Response 5 showed that the single-dataset estimation remains reliable. In addition to the global responses, we provided reviewer-specific clarifications where needed.

According to the reviewers’ feedback, we have added clarifications in the main text and supplemented both theoretical and empirical results in the appendix, as in the updated revision. We believe that the revisions and detailed responses have successfully addressed all major concerns, further strengthening the clarity of our paper. We hope the AC and reviewers will find our response satisfactory.

*continued in the next reply*

---

### Meta-Review · Area_Chair_7Jky · 2026-01-05

**Summary:**

This paper theoretically studies the problem of learning under subpopulation shift (spurious correlation, under-representation, and class imbalance). Under the  Gaussian features assumptions, this paper derives a closed-form  Bayesian classifier for the overall and per-group accuracies for the binary classification case.  Bayesian classifier for binary classification is widely studied in the literature. This paper seems to be an incremental work.

**Reviewer Concerns:**

The Reviewer tAiR is concerned about the incremental novelty compared with prior work and the strong Gaussian assumptions further requiring a block-diagonal covariance structure.

Both the reviewer  uHDz and the reviewer Zda8 are concerned about the strong Gaussian Feature Assumption and the Linear Classifier Limitation, which limit the feasibility of using the theory for estimating the expected performance on real problems.


It seems that these concerns have not been well addressed.

**Reviewer Scores:**

The reviewer tAiR and the reviewer uHDz gave scores below the acceptance threshold.

After the rebuttal, some of their concerns seem not been well addressed.

---

### Decision · Program_Chairs · 2026-01-26

Reject